# Eph-ephrin signaling couples endothelial cell sorting and arterial specification

Jonas Stewen [1], Kai Kruse [1,2], Anca T. Godoi-Filip [1], Zenia [1], Hyun-Woo Jeong [1,3], Susanne Adams[1], Frank Berkenfeld[1], Martin Stehling[4], Kristy Red-Horse [5,6,7], Ralf H. Adams [1] ✉ & Mara E. Pitulescu [1] ✉

Cell segregation allows the compartmentalization of cells with similar fates during morphogenesis, which can be enhanced by cell fate plasticity in response to local molecular and biomechanical cues. Endothelial tip cells in the growing retina, which lead vessel sprouts, give rise to arterial endothelial cells and thereby mediate arterial growth. Here, we have combined cell type-specific and inducible mouse genetics, flow experiments in vitro, single-cell RNA sequencing and biochemistry to show that the balance between ephrin-B2 and its receptor EphB4 is critical for arterial specification, cell sorting and arteriovenous patterning. At the molecular level, elevated ephrin-B2 function after loss of EphB4 enhances signaling responses by the Notch pathway, VEGF and the transcription factor Dach1, which is influenced by endothelial shear stress. Our findings reveal how Eph-ephrin interactions integrate cell segregation and arteriovenous specification in the vasculature, which has potential relevance for human vascular malformations caused by *EPHB4* mutations.

Cell sorting and segregation processes are critical for establishing tissue boundaries and compartmentalizing cells with similar fates in the developing organism[1–3]. Previous studies have shown that in the embryonic vertebrate hindbrain, patterning processes are enhanced by dynamic and context-dependent alterations in cell identity, leading to sharper tissue boundaries[4]. Cell segregation[5] and specification[6] are also critical in the developing vascular system, where endothelial cells (ECs) acquire distinct identities, giving rise to separated arterial and venous branches interconnected by capillaries.

Defective arteriovenous (AV) specification is a cause of vascular malformations, which can be fatal and have a significant impact on health[7]. Likewise, given the vital role of adequate arterial blood flow[8,9] in providing oxygen and nutrients to tissues, it is not surprising that acute or chronic impairment of arterial transport is associated with a wide range of human diseases including stroke, cardiac infarction, diabetic ischemia, or avascular necrosis in bone[10–13]. Thus, a better understanding of the cellular and molecular mechanisms controlling arterial growth and function has great medical relevance.

Previous work has shown that the Notch signaling pathway is a major regulator of AV specification. In the embryo, Notch promotes the development of the first artery, the dorsal aorta, and antagonizes venous differentiation[14,15]. Notch function in the endothelium remains important throughout life. Ectopic expression of active Notch4 in ECs, for example, is sufficient to trigger the formation of AV malformations in the adult brain through the enlargement of capillaries[16].

Our own work has shown that Notch signaling plays a key role during artery formation in the postnatal retina[17]. Genetic fate tracking has revealed that endothelial tip cells, which are located at the distal (leading) end of endothelial sprouts and mediate vascular growth[18], give rise to arterial but not venous endothelial cells[17,19]. This process depends not only on the Notch pathway but also on vascular endothelial growth factor (VEGF) signaling, which is another important mediator of arterial specification[17,20–22].

[1]Department of Tissue Morphogenesis, Max Planck Institute for Molecular Biomedicine, D-48149 Münster, Germany. [2]Bioinformatics Service Unit, Max Planck Institute for Molecular Biomedicine, D-48149 Münster, Germany. [3]Sequencing Core Facility, Max Planck Institute for Molecular Biomedicine, D-48149 Münster, Germany. [4]Flow Cytometry Unit, Max Planck Institute for Molecular Biomedicine, D-48149 Münster, Germany. [5]Department of Biology, Stanford University, Stanford, CA, USA. [6]Institute for Stem Cell Biology and Regenerative Medicine, Stanford University School of Medicine, Stanford, CA, USA. [7]Howard Hughes Medical Institute, Stanford, CA, USA. ✉e-mail: ralf.adams@mpi-muenster.mpg.de; mara.pitulescu@mpi-muenster.mpg.de

To gain deeper insight into tip cell progeny-mediated artery formation, we have turned our attention to Eph-ephrin signaling, which is a major cell contact-mediated system participating in a wide range of morphogenetic processes[23]. While Eph receptors possess a cytoplasmic kinase domain enabling downstream (forward) signaling, ephrin ligands are also membrane-anchored and can mediate so-called reverse signaling into ligand-expressing cells, resulting in bidirectional signal transduction[24,25].

In the vasculature, expression of ephrin-B2, a B-type ephrin with a transmembrane region and short cytoplasmic tail, is high in arteries, whereas expression of EphB4, one of receptors for ephrin-B2, dominates in venous ECs[26,27]. Ephrin-B2 and EphB4 are indispensable in ECs during angiogenic blood vessel growth, similar to key Notch pathway components such as the receptor Notch1[28–30] and the ligand Delta-like 4 (Dll4)[31–33], which complicates functional studies addressing AV specification.

Here, we have used Esm1-CreERT2 transgenic mice[34], which allow genetic manipulations in retinal tip cells, to study the role of ephrin-B2 and EphB4 in artery formation. Our findings in mutant mice and cultured ECs show that expression of the ligand-receptor pair is reciprocally balanced, which involves the SoxF family transcription factors Sox7, Sox17 and Sox18[35]. We find that Eph-ephrin interactions intersect with VEGF-mediated signaling and act both upstream and downstream of the Notch pathway. Ephrin-B2, which is strongly upregulated after loss of EphB4, promotes major signaling responses regulating cell sorting, migration under flow and the specification of arterial identity.

As mutations in the human EPHB4 gene are the cause of vascular malformations[36–38], insights into regulation of AV patterning by ephrin-B2 and EphB4 might well have clinical relevance.

## Results

### Ephrin-B2 and EphB4 control arteriovenous patterning

Eph-ephrin interactions are known to control cell sorting processes in many different organs and biological settings[39–41]. CellTracker-labeled human umbilical venous endothelial cells (HUVECs) or human umbilical arterial ECs (HUAECs) in vitro show extensive intermingling at the interface in homotypic confrontation assays (Fig. 1a). By contrast, a sharp border is formed between venous and arterial cells in heterotypic cultures. EphB4 and ephrin-B2 are expressed in both cell types, albeit at different levels (Fig. 1b). Notably, silencing RNA (siRNA)-mediated knockdown (KD) of EPHB4 expression in HUVECs (Supplementary Fig. 1a) enables intermingling with HUAECs, indicating that segregation of arterial and venous cells is mediated by EphB4 (Fig. 1a). Reciprocally, although to a lesser extent, HUVECs mix with HUAECs when EFNB2 is silenced (Fig. 1a, Supplementary Fig. 1a), confirming that EphB4 and its sole ligand ephrin-B2 promote arteriovenous segregation (Fig. 1j).

The postnatal vascularization of the murine retina is a well-established model system for the analysis of vascular growth and AV differentiation[42]. Angiogenesis in this model is impaired by postnatal pan-endothelial inactivation of conditional (loxP-flanked) alleles of Ephb4 or Efnb2, respectively, with the tamoxifen-inducible Cdh5-CreERT2 transgenic allele[43]. The resulting pan-endothelial Efnb2[iΔEC] or Ephb4[iΔEC] mutants show reductions in radial vessel outgrowth, vessel branching, vascular density, EC proliferation, sprouting, and the number of filopodia at postnatal day (P) 6 (Supplementary Fig. 1b–j).

Despite the phenotypic similarities in the two mutant models, artery formation is differentially affected. Efnb2 inactivation leads to decreased arterial extension into the periphery and reduction of the arterial area (Supplementary Fig. 1d), whereas loss of Ephb4 increases the area covered by arteries relative to the total EC area without altering arterial extension (Supplementary Fig. 1e). However, malformed arteries are also more abundant in pan-endothelial Ephb4[iΔEC] mutants relative to control littermates (Supplementary Fig. 1k). Moreover, 25% of Ephb4[iΔEC] mutants manifest pathological

arteriovenous crossings, a malformation described in human retinopathies[44], which can lead to arteriovenous nicking or branch retinal vein occlusion (BRVO)[45–47].

Based on these results and our previous work demonstrating that the progeny of Esm1+ tip cells in the postnatal retina is incorporated into growing arteries[17,19], we reasoned that EphB4 and ephrin-B2 might mediate the guidance of future arterial ECs into their correct domain. Consistent with this model, EphB4 not only marks veins and ephrin-B2 not only arteries, but both molecules are co-expressed by sprouting tip cells and by ECs in the adjacent vessel plexus[43].

To directly address the roles of ephrin-B2 and EphB4 in sprouting ECs, we combined loxP-flanked alleles of these genes with Esm1-CreERT2 transgenic mice[34] and the R26-mTmG reporter[48]. This strategy revealed that postnatal Ephb4 inactivation in Esm1+ cells (Ephb4[iΔTC]) leads to a significantly increased incorporation of recombined (GFP+) ECs into arteries at P6, without significantly modifying the total contribution of GFP+ cells to the EC area (Fig. 1c, Supplementary Fig. 2a). This is confirmed by quantitation of GFP+ cell distribution into the angiogenic front, plexus and artery regions relative to the total GFP+ area (Fig. 1e). Furthermore, Ephb4[iΔTC] retinas show a larger number of arterial branches and increased artery extension relative to control samples (Fig. 1g).

Conversely, inactivation of Efnb2 results in a strongly decreased proportion of GFP+ area per EC area and impaired incorporation of tip cell-derived progeny into mutant (Efnb2[iΔTC]) arteries (Fig. 1d, f, Supplementary Fig. 2b). This decrease in arterial GFP+ cells is confirmed by quantitation of the percentual GFP+ cell distribution inside the angiogenic front, plexus and artery (Fig. 1f). Phenotypically, Efnb2[iΔTC] retinas show fewer arterial branches and decreased extension compared to control littermate retinas (Fig. 1h). While apoptosis is not detectably increased in Efnb2[iΔTC] GFP+ retinal ECs, proliferation (EdU incorporation) is reduced relative to control GFP+ cells (Supplementary Fig. 2c, d).

Taken together, these data indicate that EphB4 restrains the incorporation of tip cell progeny into arteries, whereas ephrin-B2 promotes this process. Interestingly, similar to pan-endothelial Ephb4 mutants, 25% of Ephb4[iΔTC] mice display crossing of arteries and veins. Moreover, Ephb4[iΔTC] mutant retinas have overextended retinal arteries with long side branches relative to control littermates (Fig. 1i, Supplementary Fig. 2e). These phenotypes indicate that EphB4 in tip cell progeny controls normal AV patterning, which also links defective interactions between EphB4 and ephrin-B2 to vascular malformations.

### Reciprocal regulation of ephrin-B2 and EphB4

Apart from finding that ephrin-B2 and EphB4 play opposite roles in artery formation, we found that the expression of the two molecules is regulated reciprocally. siRNA-mediated EPHB4 knockdown (siEPHB4) in HUVECs results in a marked increase of ephrin-B2 transcript and protein, whereas EphB4 is upregulated after knockdown of EFNB2 (Fig. 2a–c). In vivo, pan-endothelial inactivation of Efnb2 results in increased EphB4 protein levels in the front of the P6 retinal vasculature but not in the plexus (Fig. 2d, e, Supplementary Fig. 3a). As ephrin-B2 immunostaining with the available antibodies is unreliable, we used a fusion protein consisting of the EphB4 extracellular domain and human alkaline phosphatase (EphB4/AP) for the detection of the ligand. This approach shows increased EphB4/AP binding to pan-endothelial mutant Ephb4[iΔEC] arteries relative to littermate controls (Fig. 2e, f, Supplementary Fig. 3b). Furthermore, analysis of FACS-sorted brain ECs, which were chosen because they are much more abundant than retinal ECs, confirms a statistically significant upregulation of Efnb2 transcripts in Ephb4[iΔEC] mutants (Supplementary Fig. 3c).

Notably, siEPHB4-treated HUVECs show increased protein expression of known arterial markers, such as Notch1, Dll4, Sox17 and VEGFR2[49] (Fig. 2a, b). Conversely, the knockdown of EFNB2 strongly

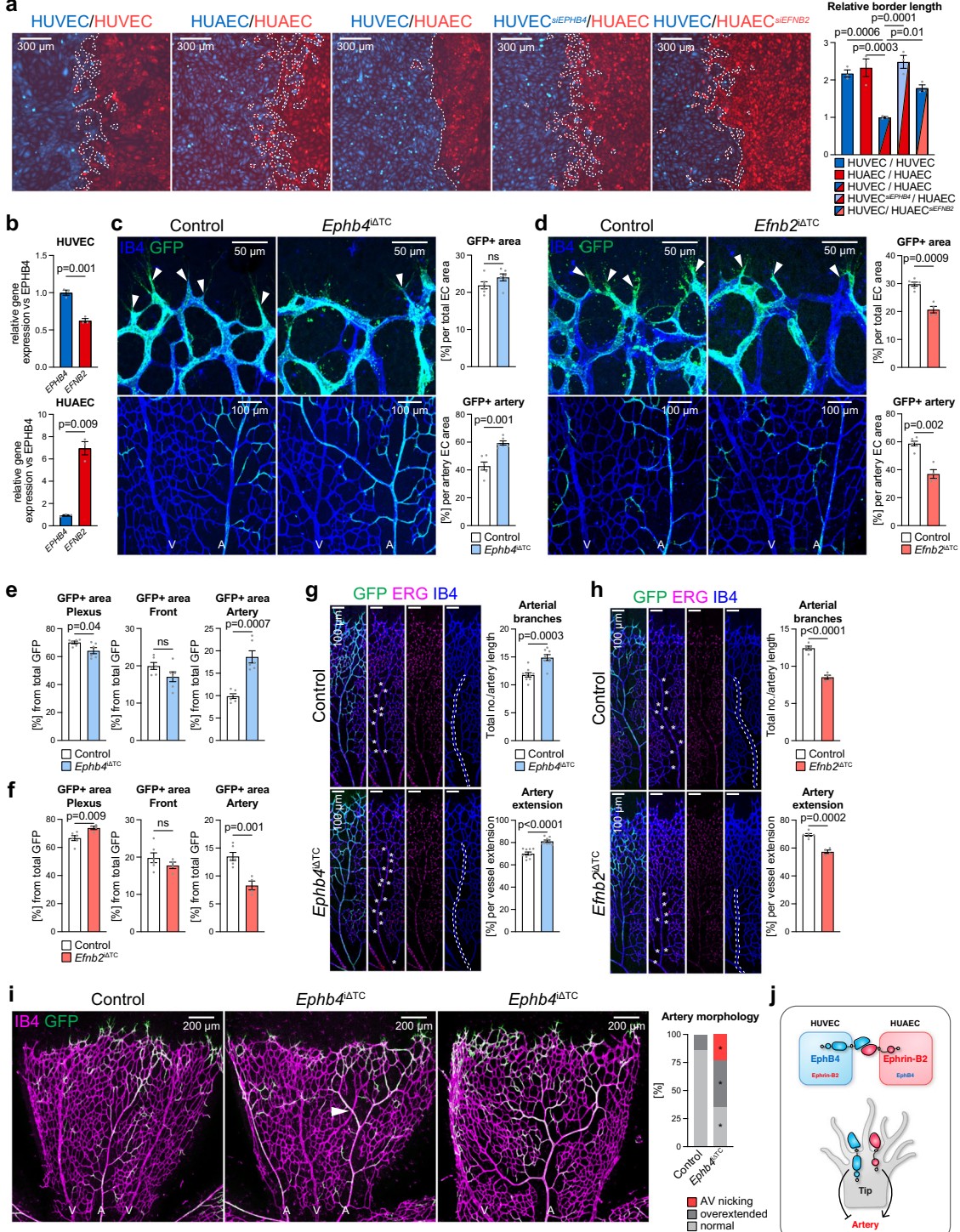

**Fig. 1 | Ephrin-B2 and EphB4 regulate EC segregation and artery formation.**
**a** Homotypic or heterotypic co-culture of HUVECs and HUAECs, as indicated. Final snapshots of CellTracker-labeled cells imaged for 48 h after removal of Ibidi insert. Quantitation graph for relative border length ($n = 3$ experiments). **b** RT-qPCR for *EPHB4* and *EFNB2* in HUVECs and HUAECs ($n = 3$ experiments). **c, d** EphB4 restrains while ephrin-B2 promotes artery growth. High-magnification confocal images of IB4 and GFP stained P6 retinal vasculature ($n = 6$ control and *Ephb4*[iΔTC] mice (**c**) and $n = 6$ control and 4 *Efnb2*[ΔTC] mice (**d**)). **e, f** Quantitation of the GFP+ area (Esm1-derived progeny) in capillary plexus, front and artery per total GFP positive area ($n = 6$ control and *Ephb4*[iΔTC] mice (**e**) and $n = 6$ control and 4 *Efnb2*[iΔTC] mice (**f**)). **g, h** Arteries in *Ephb4*[iΔTC] mice are longer and have more branches, while both

parameters are reduced in *Efnb2*[iΔTC] mice. Graphs show number of arterial branches and artery extension relative to vascular plexus extension ($n = 9$ control and 8 *Ephb4*[iΔTC] mice (**g**) and $n = 6$ control and 4 *Efnb2*[ΔTC] mice (**h**)). **i** Representative images of *Ephb4*[iΔTC] retinas with arterial alterations compared to control. Graph shows quantitation of AV retinal crossings, overextended arterial side branches and normal vascular morphology ($n = 17$ control and 17 *Ephb4*[iΔTC]). **j** Scheme depicting that ephrin-B2-EphB4 interaction and correct levels are required for proper artery formation. *P* values were calculated by one-way ANOVA (**a**), two-tailed unpaired *t* test (**b**–**h**) and Chi-square test (**i**). In vivo experiments were performed with tamoxifen injections from P1-P3 with analysis at P6 (**c, d, g**–**i**). Data are presented as mean values ± SEM. Source data are provided as a Source Data file.

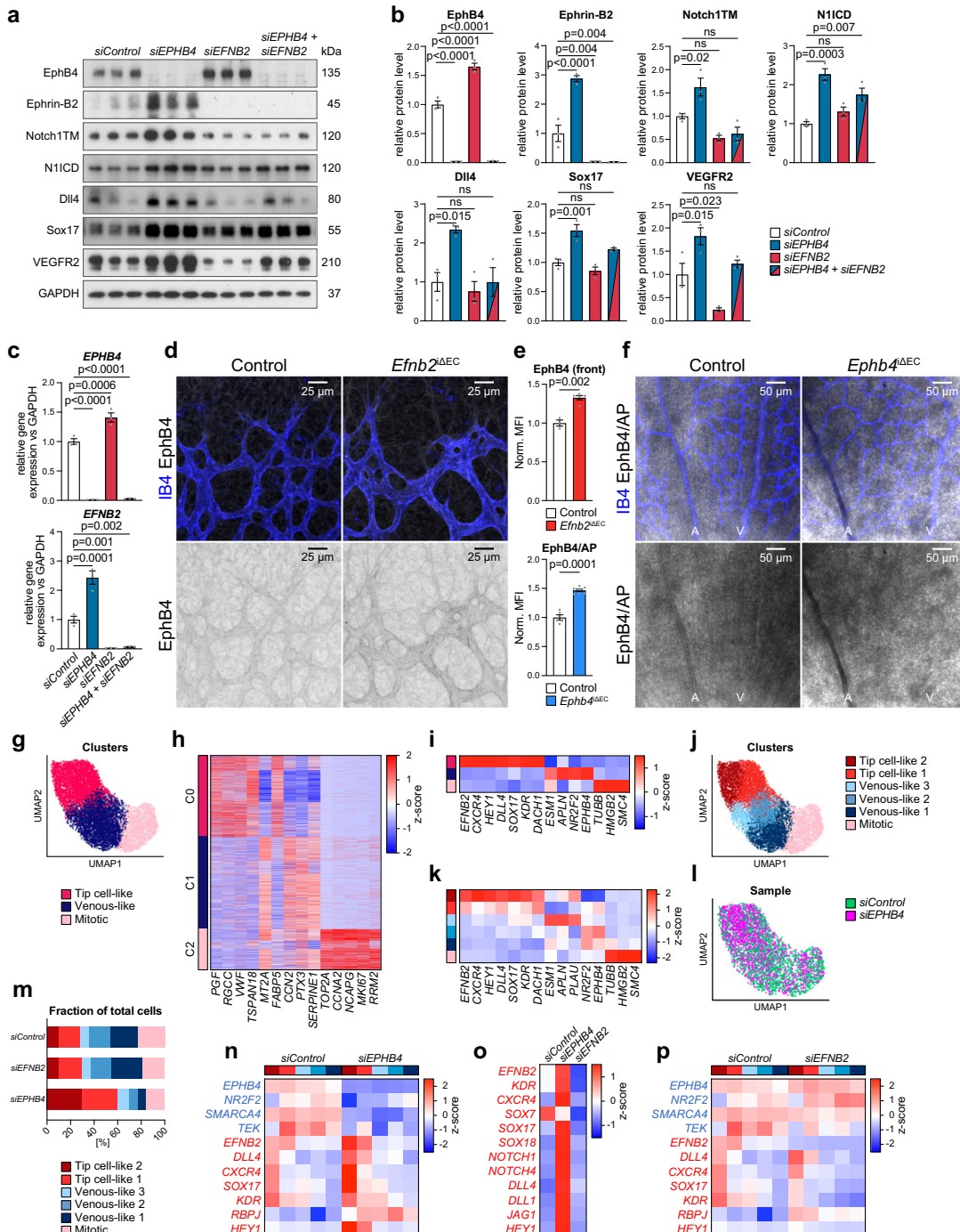

**Fig. 2 | EphB4 and ephrin-B2 modulate HUVEC heterogeneity and cell fate plasticity. a, b** Reciprocal regulation of EphB4 and ephrin-B2. Immunoblot analysis of EphB4, ephrin-B2, transmembrane Notch1 (NOTCH1TM), cleaved Notch1 intracellular domain (N1ICD), Dll4, Sox17, VEGFR2 and GAPDH in *siControl*, *siEPHB4*, *siEFNB2* and *siEPHB4 + siEFNB2* HUVEC lysates (*n* = 3 experiments). **c** RT-qPCR for *EPHB4* and *EFNB2* gene expression after *EPHB4* and *EFNB2* KD in HUVECs (*n* = 3 experiments). **d** EphB4 upregulation in *Efnb2*^iΔEC retinal vascular front. **e** EphB4 immunosignal quantitation (*n* = 3 control and 5 *Efnb2*^ΔEC mice) and quantitation of EphB4/AP signal (*n* = 5 control and 7 *Ephb4*^iΔEC mice). **f** Ephrin-B2 increase in *Ephb4*^iΔEC P6 retina. Indirect ephrin-B2 detection using EphB4/AP protein. **g** 2D UMAP representation of unbiased Leiden clustering of merged *siControl*, *siEPHB4* and *siEFNB2* HUVEC scRNA-seq. **h** Z-normalized gene expression of top 5 marker genes per Leiden cluster. **i** Z-normalized gene expression of key cell type markers in Mitotic, Venous-like and Tip cell-like clusters. **j** 2D UMAP representation of

unbiased Leiden sub-clustering. **k** Z-normalized gene expression of key cell type markers in Mitotic, Venous-like 1-3 and Tip cell-like 1-2 subclusters. **l** UMAP representation showing increased proportion of *siEPHB4* cells in the Tip cell-like clusters relative to *siControl*. **m** Fraction distribution of *siControl*, *siEPHB4* and *siEFNB2* cells in HUVEC subclusters. **n** Upregulation of arterial and downregulation of venous genes in *siEPHB4* cells. Arterial and venous markers selected from pseudobulk DGE analysis comparing *siControl* and *siEPHB4* subpopulations. **o** Z-normalized gene expression comparing *siControl*, *siEPHB4* and *siEFNB2* HUVECs. **p** Arterial and venous markers selected from pseudobulk DGE analysis comparing *siControl* and *siEFNB2* subpopulations. *P* values were calculated using one-way ANOVA (**b, c**) and two-tailed unpaired *t* test (**e**). In vivo experiments were performed with tamoxifen injections at P1-P3 with analysis at P6 (**d, f**). Data are presented as mean values ± SEM. Source data are provided as a Source Data file.

lowers VEGFR2 protein levels without substantially altering the other markers. Double knockdown of both *EFNB2* and *EPHB4* partially restores NICD levels and normalizes Dll4, Sox17 and VEGFR2 protein levels (Fig. 2a–c), arguing that the higher expression of arterial markers in *siEPHB4* HUVECs requires upregulation of ephrin-B2.

Altogether, these results indicate an unexpected reciprocal balance controlling ephrin-B2 ligand and EphB4 receptor expression with implications for arterial fate specification. This reciprocal regulation is also observed transcriptionally in cultured human retinal endothelial cells (HRECs) (Supplementary Fig. 3d), indicating the conservation of this mechanism in ECs from different vascular beds. On the other hand, silencing of *EFNB2* in HUAECs, which express much higher levels of the ligand than HUVECs (Supplementary Fig. 3e), does not lead to upregulation of *EPHB4* (Supplementary Fig. 3f). This indicates that reciprocal regulation of ephrin-B2 and EphB4 levels is not uniformly seen in all EC subtypes.

### Regulation of EC heterogeneity and cell fate plasticity

To gain deeper insight into how EphB4 and ephrin-B2 regulate EC function and AV fate specification in vitro, we employed a single-cell RNA sequencing (scRNA-seq) approach. Integrated analysis of the transcriptome of HUVECs treated with control siRNA (*siControl*), *siEPHB4* or *siEFNB2* siRNAs yielded 3 clusters (C0, C1 and C2) with distinct expression signatures of marker enrichment in two-dimensional (2D) Uniform Manifold Approximation and Projection (UMAP) representation (Fig. 2g, h). The C0 cluster presents cells with increased expression markers that are characteristic for arteries and sprouting tip ECs (*EFNB2, CXCR4, HEY1, DLL4, SOX17, KDR* and *DACH1*) (Fig. 2i). The cluster C1 contains cells with higher expression of venous markers (*NR2F2* and *EPHB4*) and C2 contains proliferating endothelial cells. Breaking down these initial clusters into smaller and more homogenous subpopulations identifies two subclusters with different levels of tip cell marker expression and three venous-like subclusters (Fig. 2j). Among the latter, one subpopulation shows increased expression of known tip cell genes such as *ESM1, APLN* and *PLAU*, placing it closer to the tip cell-like subclusters (Fig. 2k).

*EPHB4* knockdown diminishes the mitotic and venous-like subclusters while doubling the fraction of cells in the tip cell-like subpopulation (Fig. 2l, m). Furthermore, pseudobulk differential gene expression (DE) analysis on the subpopulation level shows elevated expression of tip cell and arterial marker genes, namely *KDR, EFNB2, CXCR4, SOX17* and *DLL4*, in the *siEPHB4* tip cell-like population (Fig. 2n). Conversely, expression levels of *NR2F2, TEK* and *SMARCA4*, key regulators of venous cell fate[50,51], are reduced (Fig. 2n). EphB4 role in repressing arterialization is seen not only in HUVECs but also in HRECs, as *EPHB4* silencing increases expression of arterial markers detected by bulk qPCR (Supplementary Fig. 3g, h).

On the other hand, knockdown of *EFNB2* in HUVECs marginally increases the venous-like population at the expense of mitotic cells, without considerably altering the proportion of tip cell-like cells and with limited changes in venous marker expression (Fig. 2m, o, p). Expression of most arterial genes interrogated in tip cell-like populations and in pooled HUVECs or HRECs is not markedly altered by *EFNB2* silencing. However, expression of *GJA5*, a gene coding for the Connexin 40 (CX40) tight-junction protein and marker of differentiated arterial cells, increases (Fig. 2o, p, Supplementary Fig. 3g, h). Similarly, the knockdown of *EFNB2* in HUAECs results in increased expression of *GJA4* and the related *GJA5* connexin gene (Supplementary Fig. 3i).

To address the fate of *siEFNB2* and *siEPHB4* KD cells in a more unbiased fashion, we investigated the expression of arterial markers from two previously published scRNA-seq datasets[52,53] in our tip cell-like cluster. In this analysis, *EPHB4* KD HUVECs resemble arterial ECs of the angiogenic (and therefore immature) mouse retina (Supplementary Fig. 3j, k) and less those of quiescent arteries in the adult mouse brain (Supplementary Fig. 3l). Conversely, *EFNB2* KD HUVECs show lower expression of arterial markers, which suggests these cells have more venous-like features. Indeed, *EFNB2* KD HUVECs segregate from arterial cells in a heterotypic confrontation assay, similarly to co-cultured *siControl*-treated HUVECs and HUAECs (Supplementary Fig. 3m). Furthermore, *EFNB2* KD downregulates the expression of *VEGFR2, ESM1* and various transcripts for regulators of angiogenesis in HUVECs (Fig. 2o, p, Supplementary Fig. 3n, o).

It also should be noted that KD of *EPHB4* in HUVECs leads not only to upregulation of transcripts for ephrin-B2 but also for the related ligand ephrin-B1 (Supplementary Fig. 3p). Likewise, *siEFNB2* transfection leads to elevated expression of EphB4 but also of the B-class receptor EphB2 (Supplementary Fig. 3p), which raises the possibility that additional Eph/ephrin molecules might contribute to the regulation of EC behavior in these settings.

Altogether, these data indicate that ephrin-B2 and EphB4 preserve the equilibrium of AV markers and angiogenic regulators in ECs in vitro.

### Interplay between EphB4-ephrin-B2 and SoxF transcription factors

Consistent with increased arterial specification in the postnatal *Ephb4*^iΔEC retina, Sox17 immunostaining is elevated in tip cells and throughout the arterial network (Fig. 3a, b). In contrast, Sox17 is not appreciably affected in established *Efnb2*^iΔEC arteries (Fig. 3c, d). However, the proportion of ECs with high Sox17 immunostaining (Sox17hi) is diminished in EC-specific *Efnb2* mutants and elevated in *Ephb4* loss-of-function retinas (Fig. 3b, d). *EPHB4* silencing in cultured HUVECs not only upregulates *SOX17* but also *SOX18* expression (Fig. 3e), which, together with *SOX7*, comprise the SoxF family of transcription factors that can function redundantly in the cardiovascular system[35,54].

Confirming that Sox17 is also an important regulator of arterial specification[49,54], knockdown of *SOX17* together with *SOX7* and *SOX18* leads to reduced expression of the arterial marker genes *DLL4, HEY1*, and *VEGFR2* in HUVECs (Supplementary Fig. 4a). At the same time, transcripts for the venous transcription factor *NR2F2* are increased. Importantly, combined knockdown of the three *SOXF* genes leads to a marked decrease of ephrin-B2 and upregulation of EphB4 both at the transcript and protein level (Fig. 3f–h).

In HUVECs but also in mouse retinal tip cells, *SOX18* transcripts show higher expression relative to *SOX7* and *SOX17* (Fig. 3e, Supplementary Fig. 4b)[52]. We therefore searched for putative Sox18 binding motifs in the promoter and intron regions of the *EPHB4* and *EFNB2* genes (Fig. 3i). Potential candidate binding motifs with approximately 1 kb of surrounding genomic DNA were inserted into luciferase reporter vectors and tested for activity in HUVECs. Whereas two putative Sox18 binding regions from the *EPHB4* gene generate no luciferase activity (Fig. 3j), two regions from the *EFNB2* gene substantially increase luciferase expression, which is further augmented by *EPHB4* silencing (Fig. 3j, k). These findings indicate that *EFNB2* might be directly regulated by SoxF proteins. Moreover, in retinal vasculature of *Efnb2-GFP* reporter mice[55], which carry an insertion of a cassette encoding nuclear EGFP in exon 1 of the *Efnb2* gene, nuclear EGFP signal and Sox17 expression strongly correlate in endothelial sprouts (Fig. 3l, m).

We also addressed whether beta-catenin or Smad4, two important regulators of AV differentiation[56–58], might reciprocally control the expression of *EPHB4* and *EFNB2*. However, the knockdown of beta-catenin (*CTNNB1*) lowers *EFNB2* transcript levels without affecting *EPHB4*, whereas *SMAD4* KD strongly reduces *EPHB4* expression without altering *EFNB2* (Supplementary Fig. 4c, d). Taken together, these results indicate that SoxF factors contribute to the balanced expression of ephrin-B2 and EphB4, but other pathways also act upstream of the Eph-ephrin pair in ECs (Fig. 3n).

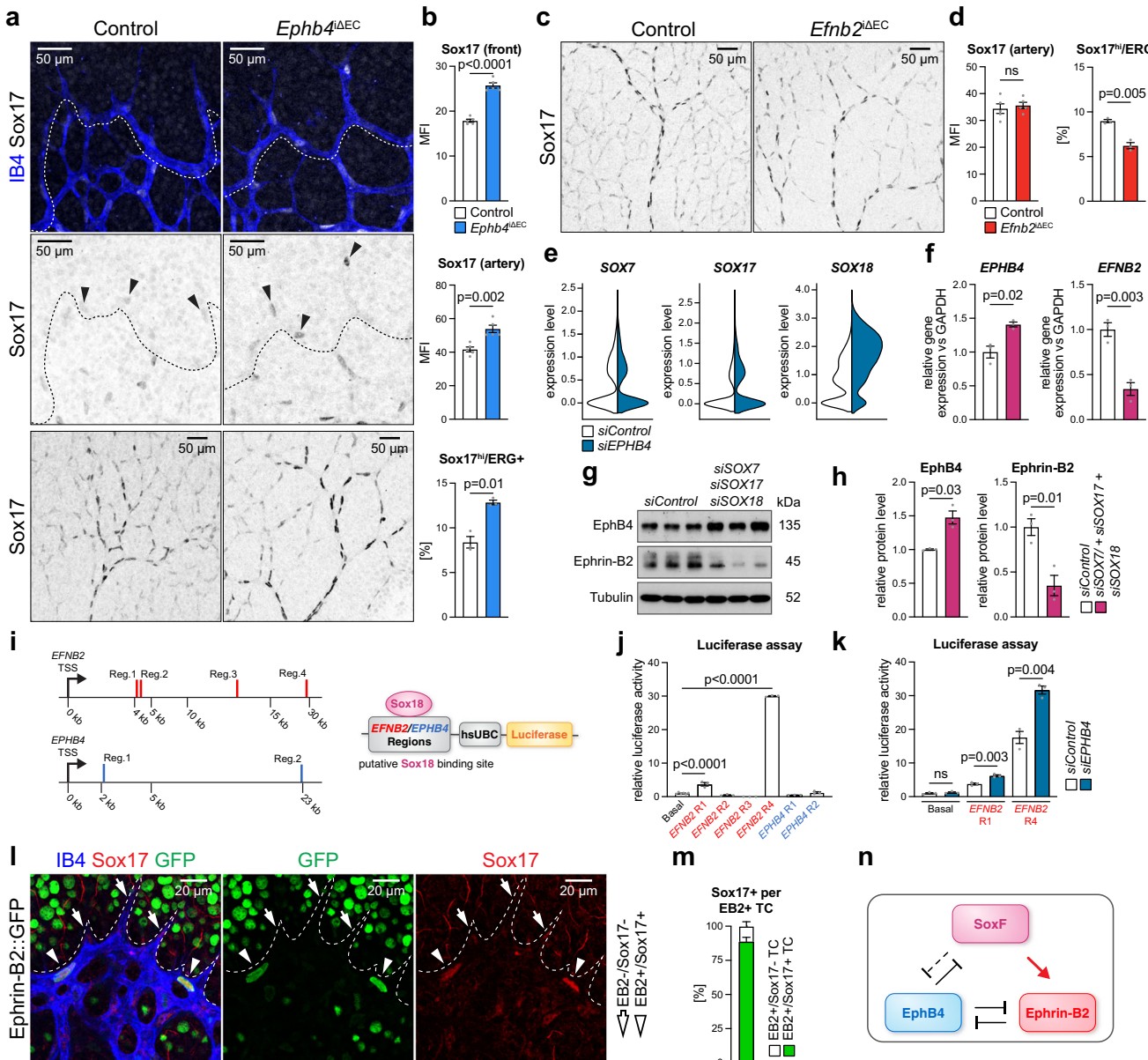

**Fig. 3 | Interplay of EphB4, ephrin-B2 and SoxF factors. a–d** Confocal images and quantitation of Sox17 immunostaining in retinal sprouts (**b**) and artery (**b**, **d**) ($n = 5$ control and 5 $Ephb4^{iΔEC}$ and $n = 5$ control and 5 $Efnb2^{iΔEC}$) and percentage of cells with high Sox17 immunosignal vs total ECs ($n = 3$ control and 3 $Ephb4^{iΔEC}$ and $n = 3$ control and 3 $Efnb2^{iΔEC}$) (**b**, **d**). **e** Violin plots of *SOX7*, *SOX17* and *SOX18* gene expression from scRNA-seq experiment comparing *siControl* and *siEPHB4* cells. **f** SoxF factors reduce *EPHB4* and increase *EFNB2* transcripts ($n = 3$ experiments). **g**, **h** Immunoblot analysis and quantitation of EphB4 and ephrin-B2 protein upon *SOX7* + *SOX17* + *SOX18* knockdown ($n = 3$ experiments). **i** Genome localization of Sox18 putative motifs in *EFNB2* and *EPHB4* human genes selected for further investigation. **j** Quantitation graphs for luciferase activity of putative binding regions of Sox18 on *EFNB2* and *EPHB4* showing direct regulation of *EFNB2* expression via Sox18 in two regions ($n = 3$ experiments). **k** Increased luciferase activity upon *EPHB4* KD compared to *siControl* for Sox18 binding regions on *EFNB2* ($n = 3$ experiments). **l** Sox17 expression in ephrin-B2+ sprouts (single confocal z-plane). IB4, GFP and Sox17 staining in P6 *Efnb2-H2B-GFP* knock-in reporter retinal vasculature. **m** Graph indicating proportion of Sox17+ cells in ephrin-B2+ tip cells ($n = 3$ *Efnb2-GFP* mice). **n** Schematic representation of interactions between Sox17, ephrin-B2 and EphB4. *P* values were calculated using two-tailed unpaired *t* test (**b**, **d**, **f**, **h**) and one-way ANOVA (**j**, **k**). In vivo experiments were performed with tamoxifen injections at P1-P3 and analysis at P6 (**a**, **c**). Data are presented as mean values ± SEM. Source data are provided as a Source Data file.

### Regulation of arterial specification through VEGFR2

VEGFR2 activation and downstream signaling are known to play a central role in the regulation of endothelial AV fate[59–62]. Arterial specification is promoted by VEGF-A-induced activation of extracellular signal-regulated kinase 1/2 (ERK1/2) and phospholipase C γ1 (PLCγ) downstream of the tyrosine phosphorylation (Y) site 1175 in VEGFR2, whereas Akt signaling, promoted by phosphorylation of the Y951 site in VEGFR2, favors the venous fate[63–66]. In *siEPHB4* HUVECs, levels of total VEGFR2 and phospho-Y1175 as well as phospho-PLCγ and phospho-ERK1/2 are considerably increased (Fig. 4a, b). While the ratio of phospho-Y1175 to total VEGFR2 is comparable in *siEPHB4* and *siControl* ECs, a similar analysis for Y951 shows that relative phosphorylation of this site is reduced, which is also reflected by lower VEGF-induced activation of Akt (Fig. 4a, b) and reduced expression of venous markers (Fig. 2n).

Confirming these in vitro findings, increased phospho-ERK1/2 can be seen in endothelial sprouts in the retinal vasculature after pan-endothelial or tip cell-specific inactivation of *Ephb4* (Fig. 4c, Supplementary Fig. 5a). These results are consistent with phospho-ERK1/2 upregulation in the embryonic vasculature after pan-endothelial

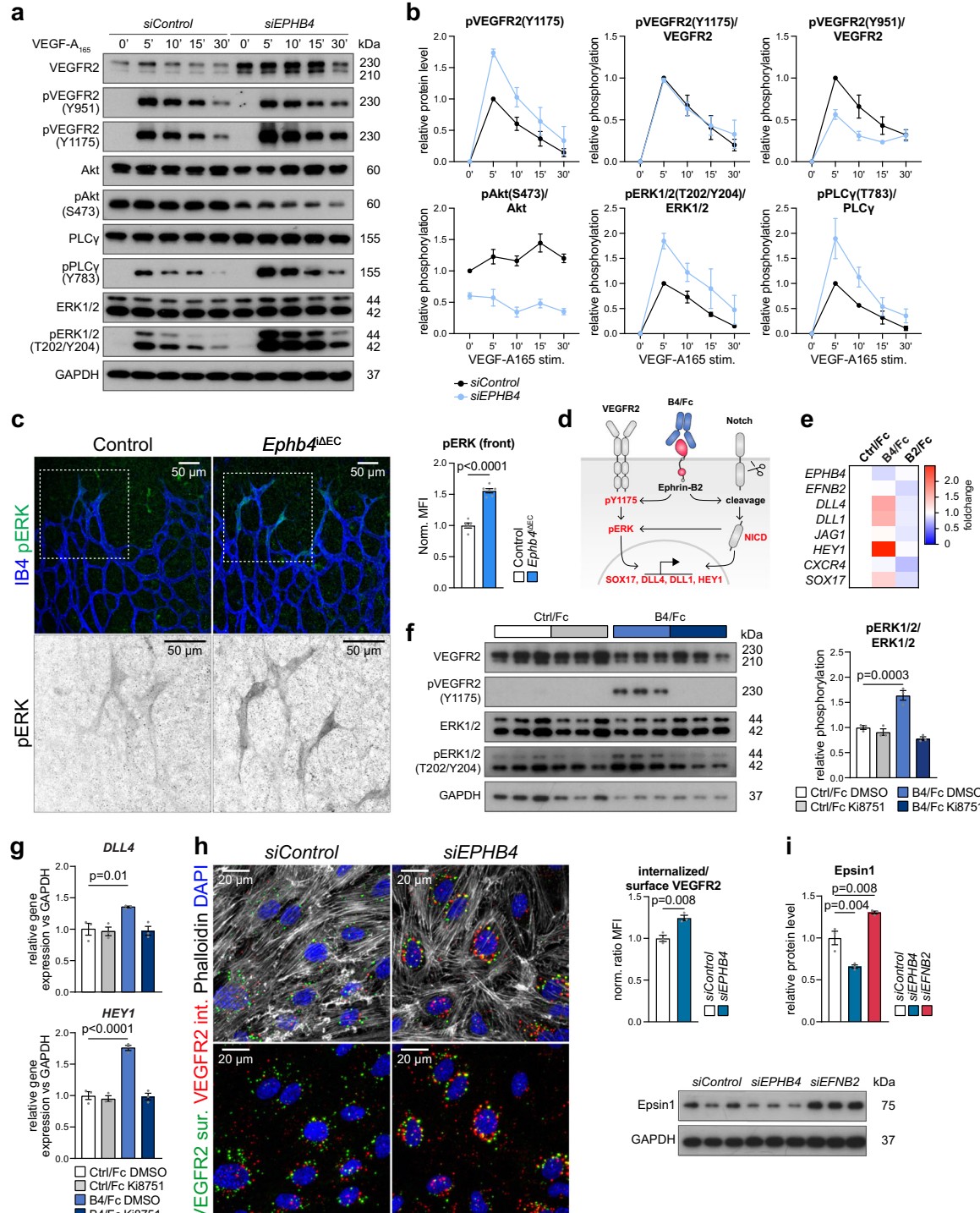

**Fig. 4 | EphB4 limits VEGFR2 activation, signaling and turnover.**
**a**, **b** Immunoblot detection and quantitation of VEGFR2 phosphorylation (Y951, Y1175) and downstream signaling molecules, as indicated (n = 3 experiments). **c** pERK1/2 and IB4 staining of P6 retinal vascular front. Lower panels show higher magnification of inverted pERK1/2 channel from depicted insets. Quantitation graph for pERK1/2 immunosignal (n = 3 control and 3 Ephb4$^{iΔEC}$ mice). **d** Scheme represents effects downstream of EphB4/Fc (B4/Fc) stimulation. **e** B4/Fc stimulation promotes HUVEC arterialization, while ephrin-B2/Fc (B2/Fc) inhibits it. Average of mRNA expression fold change for investigated genes upon 30 min stimulation of HUVECs with Ctrl/Fc, B4/Fc or B2/Fc (n = 3 experiments). **f** VEGFR2 and ERK1/2 activation upon B4/Fc stimulation. Immunoblotting and quantitation of relative pVEGFR2 (Y1175) and pERK1/2 levels in stimulated HUVECs treated with VEGFR2

inhibitor (Ki8751) or vehicle (n = 3 experiments). **g** VEGFR2 signaling is required downstream of ephrin-B2 stimulation to activate DLL4 and HEY1 expression. RT-qPCR for HEY1 and DLL4 (n = 3 experiments). **h** Increased internalized VEGFR2 in EPHB4 KD cells. Immunolabeling and quantitation of surface (green) and internalized (red) VEGFR2 in siControl and siEPHB4-treated HUVECs (n = 3 experiments). **i** EphB4 and ephrin-B2 control Epsin1 levels. Immunoblotting and quantitation of Epsin1 protein levels in siControl, siEPHB4 and siEFNB2 HUVECs (n = 3 experiments). P values were calculated by two-way ANOVA (**b**), two-tailed unpaired t test (**c**, **h**) and one-way ANOVA (**e**–**g**, **i**). In vivo experiments were performed with tamoxifen injections at P1-P3 with analysis at P6 (**c**). Data are presented as mean values ± SEM. Source data are provided as a Source Data file.

deletion of *Ephb4*[67]. In contrast, tip cell-specific inactivation of *Efnb2* or treatment with the mitogen-activated protein kinase kinase (MEK) inhibitor SL327 strongly diminishes phospho-ERK1/2 immunostaining in vascular sprouts in the retina (Supplementary Fig. 5b, c). These data argue that EphB4 normally limits VEGF-dependent activation of signaling processes that promote arterial specification, whereas ephrin-B2 plays the opposite role.

Given that ephrin-B2 and EphB4 are a ligand-receptor pair, we investigated the role of their signaling interactions. Stimulation with recombinant EphB4/Fc fusion protein is known to activate ephrin-B2 reverse signaling, while stimulation with ephrin-B2/Fc leads to activation of the EphB4 kinase and downstream signaling cascades[68,69]. Notably, 30 min stimulation of HUVECs with EphB4/Fc reduces *EPHB4* transcription but also increases expression of arterial genes including *DLL1*, *DLL4*, *HEY1* and *SOX17* (Fig. 4d, e). Conversely, stimulation with ephrin-B2/Fc decreases the abundance of *EFNB2, SOX17* and of Notch pathway gene transcripts (Fig. 4e).

Further arguing for a role of VEGF signaling downstream of ephrin-B2, EphB4/Fc enhances phosphorylation of VEGFR2 (Y1175) and ERK1/2 (Fig. 4d, f). Pharmacological inhibition of VEGFR2 with Ki8751 (Fig. 4f) or ERK inhibition with U0126 (Supplementary Fig. 5d) eliminate the increase in *DLL4* and *HEY1* transcripts after EphB4/Fc stimulation (Fig. 4g, Supplementary Fig. 5e). Likewise, *DLL4* and *HEY1* gene expression is normalized when *siEPHB4* cells are treated with the ERK inhibitor U0126 (Supplementary Fig. 5f). Together, the data above indicate that EphB4 represses and ephrin-B2 promotes arterial fate specification via VEGFR2 activation and downstream ERK signaling.

As VEGF receptor signaling takes place not only at the plasma membrane but also in endosomes[43,70,71], we investigated whether loss of EphB4 affects VEGFR2 internalization. Analysis of VEGFR2 internalization by antibody feeding assay (see Methods) shows an increased fraction of internalized VEGFR2 in *siEPHB4* cells compared to control HUVECs (Fig. 4h). While endocytosis of receptor tyrosine kinases is typically coupled to lysosomal trafficking and degradation, VEGF-A stimulation in the presence of cycloheximide (CHX) indicates strongly increased VEGFR2 stability in *siEPHB4* HUVECs (Supplementary Fig. 5g).

Consistent with increased VEGFR2 stability, the levels of Epsin1, an E3 ubiquitin ligase involved in VEGFR2 degradation[72], are decreased in *siEPHB4* cells but increased in *siEFNB2* HUVECs (Fig. 4i). Taken together, these data indicate that EphB4 promotes VEGFR2 degradation, potentially through Epsin1.

## Notch and ephrin-B2 direct the progeny of Esm1+ cells into growing arteries

Based on the finding that *Ephb4* loss-of-function leads to upregulation of ephrin-B2, we next addressed whether overexpression of the B-class ephrin alone is sufficient to enhance arterial specification. To this end, we generated conditional, Cre-controlled *Efnb2* gain-of-function mice containing an internal ribosome entry site (IRES)-coupled EGFP reporter cassette (Supplementary Fig. 6a). In combination with *Esm1-CreERT2*, ephrin-B2 overexpression (*R26-Efnb2*[iOTC]) in Esm1+ ECs increases arterial incorporation of tip cell progeny relative to heterozygous littermates (Fig. 5a, Supplementary Fig. 6b).

Ephrin-B2 expression is enhanced by Notch[73] and our previous work has shown that expression of active Notch1 (*NICD*[iOTC]) in Esm1+ ECs directs the progeny of these cells into growing arteries[17]. We therefore investigated the interplay between ephrin-B2 expression and Notch during arterial specification of tip cell progeny. While *NICD*[iOTC] gain-of-function tip cell progeny is efficiently directed into retinal arteries, a large fraction of these cells is retained inside the capillary plexus after simultaneous loss of ephrin-B2 in *NICD*[iOTC] *Efnb2*[iΔTC] compound mutants (Fig. 5b, Supplementary Fig. 6c). This indicates that ephrin-B2 mediates the migration of tip cells downstream of Notch.

An attractive hypothesis is that ephrin-B2 contributes to this process by controlling VEGF signaling. Indeed, anti-VEGF-A treatment not only impairs the specification of Esm1+ cells but also decreases the number of arterial branches (Supplementary Fig. 6d, e). Moreover, VEGF-A blockade after activation of NICD in Esm1+ cells slows down the migration of *NICD*[iOTC] tip cell progeny into arteries and reduces the number of retinal arterial branches (Fig. 5c, d, Supplementary Fig. 6f). Interestingly, Esm1 expression is increased at the *Ephb4*[iΔEC] angiogenic front but also in ECs entering arteries, indicating increased VEGF signaling (Fig. 5d). Altogether, these results indicate that ephrin-B2 is necessary downstream of Notch for the proper migration of pre-arterial cells, potentially by modulating VEGF-A signaling (Fig. 5e).

## Interplay of Eph-ephrin, Notch and ERK signaling

Notch acts upstream of ephrin-B2[60,74–76], but key components of the Notch pathway are also upregulated in *siEPHB4* HUVECs and upon stimulation with EphB4/Fc (Fig. 2a, b, n, o and Fig. 4f). Inhibition of Notch with the γ-secretase inhibitor dibenzazepine (DBZ) normalizes ephrin-B2 and decreases Dll4 protein levels, however, without significantly changing VEGFR2 levels in *siEPHB4* HUVECs (Fig. 6a). Similarly, DBZ treatment also attenuates the increased expression of *DLL4* or *HEY1* after EphB4/Fc stimulation (Supplementary Fig. 7a).

These in vitro results, which indicate that Notch signaling is also acting downstream of ephrin-B2, are further supported by findings in vivo. Genetic inactivation of *Ephb4* increases Dll4 expression in the retinal angiogenic front vasculature, artery and peri-arterial plexus (Fig. 6b, Supplementary Fig. 7b, c), whereas loss of *Efnb2* results in decreased Dll4 immunostaining in the vessel plexus near the angiogenic front (Supplementary Fig. 7d). In agreement with the previously established role of Notch in arterial specification of tip cell progeny[17], *Esm1-CreERT2*-controlled inhibition of Notch-mediated gene transcription by expression of a dominant negative version of Mastermind-like 1 (dnMaml1)[77] impairs arterial incorporation of recombined ECs (Supplementary Fig. 7e). Moreover, dnMaml1 also strongly reduces arterial specification of GFP+ EphB4-deficient ECs in *Ephb4*[iΔTC] *R26-dnMaml1*[iOTC/+] compound mutants (Fig. 6c).

ERK1/2 signaling controls sprouting angiogenesis, EC proliferation and migration. Accordingly, sprouting tip cells have high ERK activity[78–80]. Moreover, studies have shown that ERK activation downstream of VEGF controls Dll4 and Notch expression[66,79] and inhibition of Notch signaling leads to ectopic ERK activation in stalk and more distant quiescent retinal ECs[80]. In addition, ERK1/2 signaling also controls arterialization, and ECs with high ERK signaling are more likely to form arterial vessels[63,81,82].

Our data in cultured HUVECs show that EphB4/Fc treatment increases ERK1/2 activation downstream of VEGFR2 and controls transcription of genes relevant for arterial specification, such as *DLL4* and *HEY1* (Fig. 4g, h, Supplementary Fig. 5d, e). The EphB4/Fc-induced increase in phospo-ERK1/2 is eliminated by Notch inhibition (Fig. 6d).

Likewise, phospo-ERK1/2 immunostaining in retinal vascular sprouts is reduced by expression of dnMaml1 (Fig. 6e) or by acute pharmacological inhibition of Notch (DAPT, 14 h) (Fig. 6f, Supplementary Fig. 7f). Consistent with previously published findings[80], robust Notch inhibition (two DAPT injections within 24 h) induces a broader upregulation of phospo-ERK1/2 in the peripheral vascular plexus (Supplementary Fig. 7g), which is likely a response to hypoxia, strong VEGF-A upregulation, and thereby increased VEGFR2 signaling in this region[17]. The sum of these data indicates that the function of EphB4 and ephrin-B2 in pre-arterial tip cell progeny is closely linked to VEGFR2, ERK and Notch activity.

Notch not only controls arterial cell specification but also cell migration downstream of Eph-ephrin signaling. Acute inactivation of *Ephb4* in Esm1+ ECs, analyzed at 36 h after a single dose of 4-hydroxy

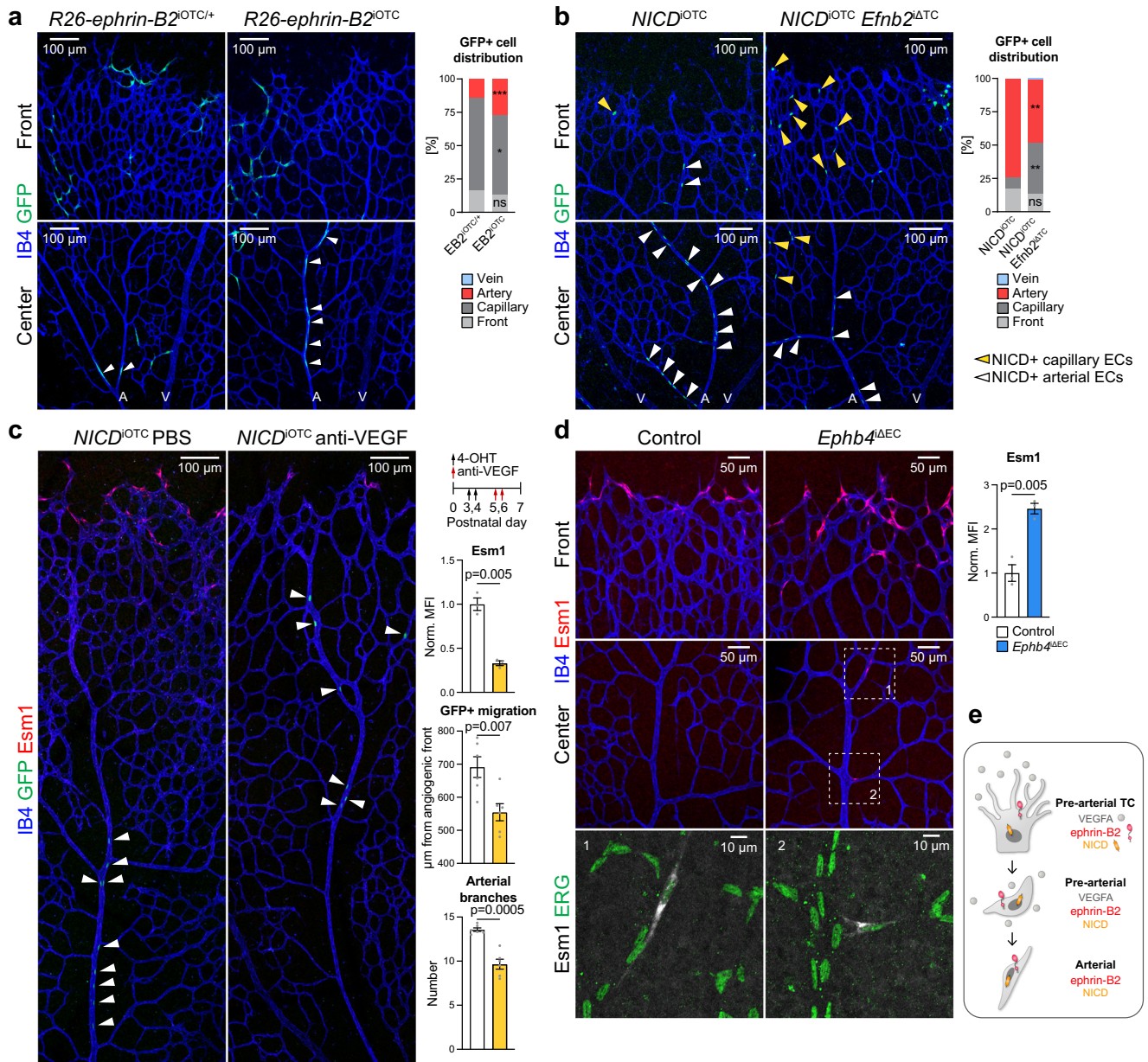

**Fig. 5 | Interplay of ephrin-B2, VEGF-A and Notch in pre-arterial ECs.**
**a** Distribution of ephrin-B2 overexpressing cells (GFP+) within the retinal vasculature ($n = 10$ *R26-ephrin-B2*[iOTC] heterozygotes and 12 homozygotes). **b** Confocal images of IB4+ and GFP+ (NICD+) ECs in the P6 retinal vasculature. Graph shows distribution of GFP+ cells ($n = 3$ control/ *NICD*[iOTC] and 3 *NICD*[iOTC]; *Efnb2*[iΔTC] mice). **c** Confocal images of vehicle or anti-VEGF treated *NICD*[iOTC] animals showing IB4+, Esm1+ and GFP+ (NICD+) ECs. Graphs show quantitation of Esm1 mean fluorescence intensity (MFI), migration distance of GFP+ cells from the vascular front towards the optic disc, and number of arterial branches ($n = 3$ *NICD*[iOTC] vehicle and *NICD*[iOTC] treated with anti-VEGF for Esm1 immunostaining, $n = 6$ *NICD*[iOTC] vehicle and *NICD*[iOTC]

treated with anti-VEGF for migration and arterial branch quantitation). **d** Esm1 upregulation in the *Ephb4*[iΔEC] retinal front vasculature and plexus. Quantitation of Esm1 staining and immunosignal at the vascular front ($n = 3$ control and 3 *Ephb4*[iΔEC] mice). **e** Scheme depicting the roles of VEGF-A, NICD and ephrin-B2 in arterialization and migration of arterial cells. *P* values were calculated using two-tailed unpaired *t* test (**a**–**d**). In vivo experiments were performed with tamoxifen injections at P1-P3 with analysis at P6 (**a**, **b**, **d**) or 4-OHT injections at P3-P4 followed by anti-VEGF administration at P5-P6 and analysis at P7 (**c**). Data are presented as mean values ± SEM. Source data are provided as a Source Data file.

tamoxifen (4-OHT), results in markedly increased arterial incorporation of GFP-labeled tip cell progeny without altering the total GFP + EC area (Fig. 6g, Supplementary Fig. 7h). Furthermore, the enhanced migration of *Ephb4*[iΔEC] GFP+ ECs into arteries is abrogated by pharmacological Notch inhibition with the γ-secretase inhibitor DAPT (Fig. 6h). Taken together, our data show that Notch and ERK signaling are required for arterial specification and pre-arterial cell migration downstream of ephrin-B2.

## EphB4-controlled arterial specification initiates cell cycle exit
Endothelial cell cycle has a strong impact on AV fate decisions, whereby early G1 state enhances venous specification and late G1 state promotes arterial specification[52]. In agreement with this finding, previous studies have indicated that cell cycle arrest precedes arterialization, whereas the venous transcription factor COUP-TF2/NR2F2 promotes the expression of cell cycle regulators and of EphB4, together with suppression of ephrin-B2 and Notch signaling[82–85]. 5-Ethynyl-

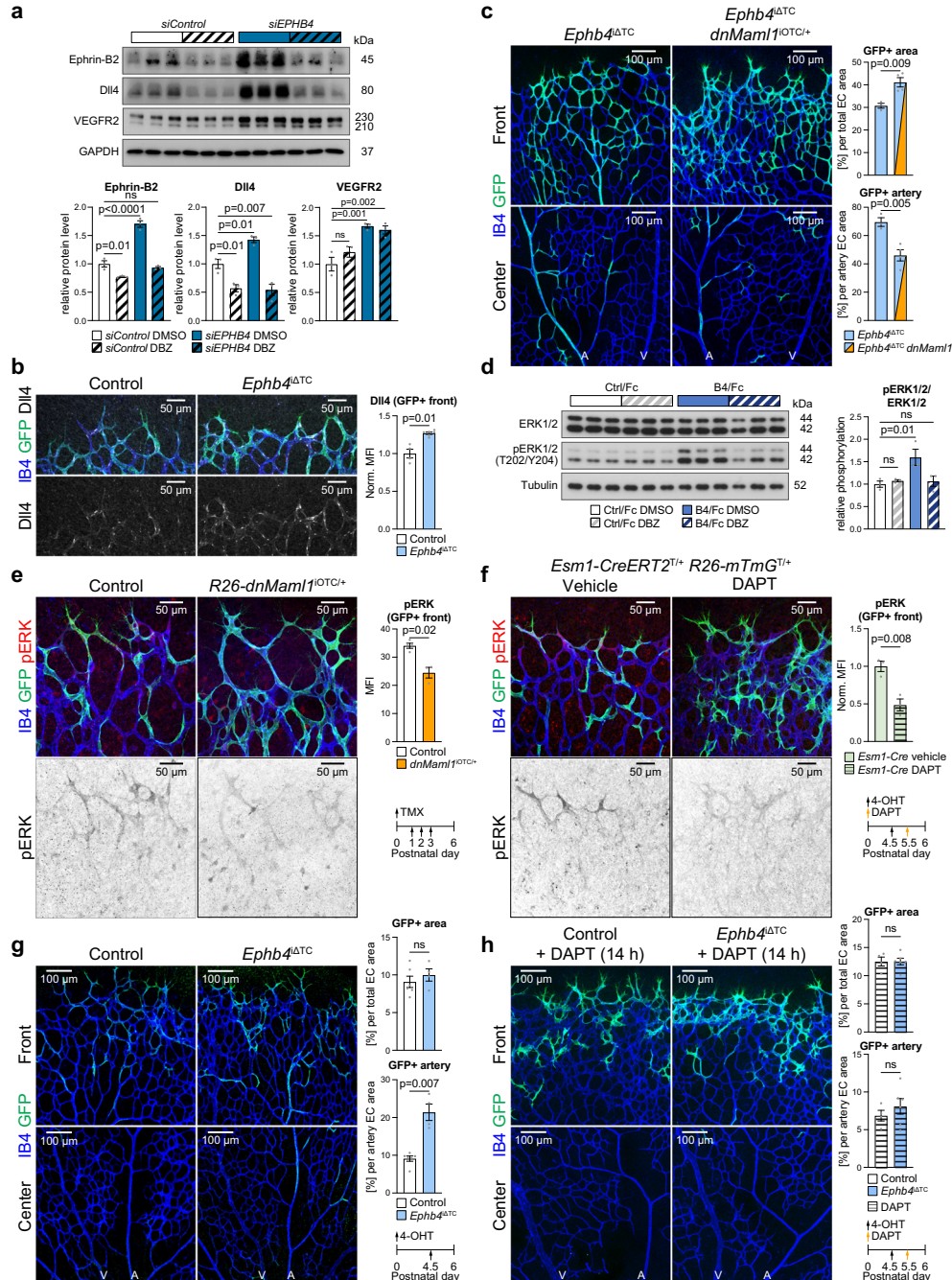

**Fig. 6 | EphB4 and Notch are linked to ERK1/2 activation. a** Notch inhibition (DBZ) impairs increase in ephrin-B2 and Dll4 in *siEPHB4* HUVECs. Immunoblotting and quantitation of ephrin-B2, VEGFR2, and Dll4 in *siControl* and *siEPHB4* cells treated with DBZ or vehicle (*n* = 3 experiments). **b** Increased Dll4 expression upon tip cell-specific *Ephb4* inactivation. Quantitation of Dll4 immunosignal in the P6 retinal angiogenic front (*n* = 4 control and 6 *Ephb4*^iΔTC). **c** Blocking Notch signaling via dominant-negative Mastermind Like Transcriptional Coactivator 1 (Maml1) reduces arterial incorporation of *Ephb4*^iΔTC cells. Graphs show quantitation of total and arterial GFP+ area relative to EC area (*n* = 3 *Ephb4*^iΔTC and 4 *Ephb4*^iΔTC; *R26-dnMaml1*^iOTC/+). **d** Reduced ERK1/2 activation upon EphB4/Fc stimulation combined with Notch inhibition. Immunoblot and quantitation of pERK1/2 in stimulated HUVECs treated with Notch inhibitor (DBZ) or vehicle (*n* = 3 experiments). **e** Tip cell-specific genetic inhibition of Notch signaling decreases ERK1/2 activation. Confocal images and quantitation of pERK in GFP+ cells at the vascular front of

*R26-dnMaml1*^iOTC/+ and control mice (*n* = 4 control and 3 *R26-dnMaml1*^iOTC/+). **f** Acute Notch inhibition (DAPT) for 14 h reduces pERK. Graph shows quantitation of pERK MFI in GFP+ cells (*n* = 3 vehicle and *n* = 3 DAPT injected *Esm1-CreERT2 R26-mTmG*^+/T animals). **g** Acute *Ephb4* inactivation (single injection with 4-OHT) promotes arterial incorporation of tip cell progeny. Quantitation of total GFP+ area and GFP+ arterial area (*n* = 6 control and 4 *Ephb4*^iΔTC mice). **h** Acute Notch inhibition (DAPT) impairs arterial incorporation of *Ephb4*-depleted ECs. Quantitation of total GFP+ area and of GFP+ area incorporated into artery per total EC area (*n* = 4 control + DAPT and 6 *Ephb4*^iΔTC + DAPT mice). *P* values were calculated using two-tailed unpaired *t* test (**b, c, e–h**) and one-way ANOVA (**a, d**). In vivo experiments were performed with tamoxifen injections at P1-P3 with analysis at P6 (**b, c, e**) or 4-OHT injections at P4.5 following analysis at P6 (**f–h**) following analysis at P6. Data are presented as mean values ± SEM. Source data are provided as a Source Data file.

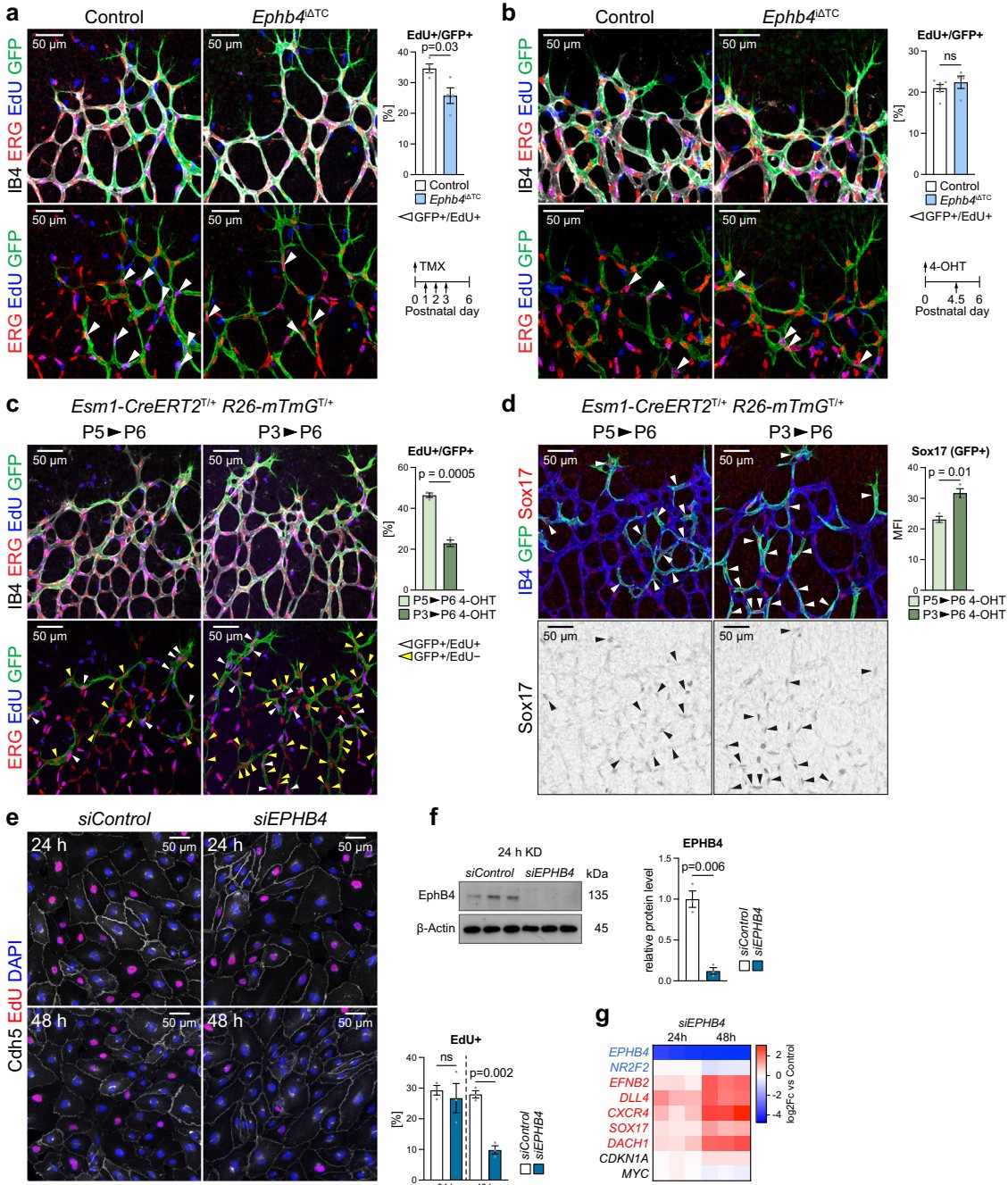

**Fig. 7 | First steps of arterial specification precede cell cycle changes. a** Reduced proliferation of tip cell progeny after prolonged *Ephb4* inactivation. Quantitation of EdU+/GFP+ progeny in the P6 retinal front area after P1-P3 tamoxifen administration (*n* = 4 control and 4 *Ephb4*^iΔTC^ mice). **b** Proliferation of *Ephb4*^iΔTC^ ECs is unaffected shortly after gene deletion. Quantitation of EdU+/GFP+ ECs in P6 retinal front area after 4-OHT administration at P4.5 (*n* = 6 control and 4 *Ephb4*^iΔTC^ mice). **c**, **d** Lineage-tracing of Esm1-derived (GFP+) cells for 24 h and 72 h. Proliferation of GFP+ cells is reduced at 72 h relative to 24 h (**c**), whereas Sox17 expression is increased (**d**) (*n* = 3 for 24 h and 72 h tracing). **e** Proliferation of *siEPHB4* and *siControl* cells 24 h and 48 h after KD. *EPHB4*-depleted cells show significantly

reduced proliferation only after 48 h of KD (*n* = 3 experiments). **f** Validation of *EPHB4* knockdown after 24 h by immunoblotting of EphB4 protein and corresponding quantitation (*n* = 3 experiments). **g** RT-qPCR of arterial, venous and cell cycle markers in *siControl* and *siEPHB4* HUVECs at 24 h and 48 h after KD. Cell cycle markers are not changed at 24 h, whereas arterial markers are already increased but lower than at 48 h (*n* = 3 experiments). *P* values were calculated using two-tailed unpaired *t* test (**a**–**d**, **f**) and one-way ANOVA (**e**, **g**). In vivo experiments were performed with tamoxifen injections at P1-P3 (**a**), 4-OHT injections at P4.5 (**b**) or 4-OHT injections at P3/P5 (**c**, **d**) following analysis at P6. Data are presented as mean values ± SEM. Source data are provided as a Source Data file.

2′-deoxyuridine (EdU) incorporation in the P6 *Ephb4*^iΔTC^ retina (after tamoxifen treatment at P1-P3) shows that *Esm1-CreERT2*-labeled (recombined) ECs at the growth front are less proliferative compared to GFP+ ECs in control littermates (Fig. 7a).

Likewise, *siEPHB4* HUVECs in culture show decreased levels of Cyclin D1, an important regulator of cell cycle progression, and

increased p21 cyclin-dependent kinase (CDK) inhibitor at 48 h of knockdown compared to *siControl* cells (Supplementary Fig. 8a–c). These changes are consistent with cell cycle state shifts seen in our scRNA-seq data, which indicate that a larger proportion of *siEPHB4* cells are found in the G1 phase, whereas the fraction of cells in the G2/ M and S phase is drastically reduced (Supplementary Fig. 8d, e).

In contrast, no significant alteration in EdU incorporation is seen after acute *Ephb4* inactivation (4-OHT given 36 h prior to analysis) (Fig. 7b) even though the migration of tip cell progeny towards arteries is already enhanced (Fig. 6g). Together, these data argue for a sequential acquisition of arterial features in which the earliest steps of arterial specification and migration precede cell cycle changes in tip cell progeny. Indeed, genetic tracking shows that *Esm1-CreERT2*-labeled ECs at 1 day after administration of a single dose of 4-OHT exhibit higher proliferation but lower Sox17 expression compared to GFP+ cells at 3 days after 4-OHT (Fig. 7c, d). Sequential acquisition of arterial features can be also observed in vitro. *EPHB4* KD for 24 h does not impair HUVEC proliferation but amplifies arterial gene expression, whereas reduced proliferation and further enhancement of arterial gene expression are evident after 48 h (Fig. 7e–g). Consistent with the role of ephrin-B2 in angiogenic blood vessel growth, *siEFNB2* HUVECs show significantly reduced EdU incorporation (Supplementary Fig. 8f).

The sum of the data above suggests stepwise arterial fate acquisition by tip cell progeny where first changes in gene expression emerge before changes in EdU incorporation, whereas cell cycle arrest precedes the further progression of arterial differentiation.

## Eph-ephrin function is linked to the transcription factor Dach1

Flow sensing is thought to be an important mediator of AV specification and is necessary for the maintenance of arterial identity[9,86–88]. Accordingly, loss of blood flow induces defects that can lead to AV malformations[89]. Unlike their venous counterpart, arterial ECs and their nuclei are elongated and align along the flow axis in the postnatal retinal vasculature (Supplementary Fig. 9a). Consistent with the upregulation of arterial markers, *siEPHB4* HUVECs elongate and align even in the absence of flow (Fig. 8a), which is accompanied by increased expression of mechanoresponsive genes (Fig. 8b). Further exposure of *siEPHB4* cells to arterial flow (15 dyn/cm²) for 12 h strongly and significantly increases alignment relative to control ECs (Fig. 8c).

Previous work has shown that arterially-committed tip cell progeny changes polarity, turns away from the vascular growth front and migrates towards arteries against the direction of blood flow[17,19,90,91]. To mimic this process, we performed live imaging of HUVECs under flow over a period of 24 h (Fig. 8d, Supplementary Fig. 9b, Supplementary Movie 1-3). The majority of *siControl* cells migrates in the flow direction under these conditions, but a fraction of 25% spontaneously reverses and moves against the flow. This fraction of turning cells is doubled for *siEPHB4* HUVECs (Fig. 8d, e). Dach1, an important transcriptional regulator of arterialization, was previously shown to promote shear stress-guided EC migration against the direction of flow[92,93]. Remarkably, nuclear Dach1 is elevated after 24 h of *EPHB4* silencing and is further increased after 48 h (Fig. 8f), which suggests that EphB4 expression restrains nuclear translocation of Dach1. In agreement, increased turning of *siEPHB4* HUVECs under flow is no longer observed after *DACH1* knockdown (Fig. 8d, e).

Cell tracking by live imaging followed by Dach1 immunostaining reveals that expression of the transcription factor in HUVECs under flow is strongly associated with turning behavior (Fig. 8g). Cells that have migrated in the direction of flow without turning show lower nuclear Dach1 immunostaining than HUVECs that have undergone turning (Fig. 8g, h). Furthermore, analysis of turning *siEPHB4* cells shows that nuclear Dach1 is highest shortly after they start moving against the flow direction and gradually declines afterwards (Fig. 8i, j). This decrease of Dach1 is analogous to retinal immunostaining where Dach1 is downregulated in the ECs of larger and therefore more mature arteries, which are exposed to higher flow[92] (Supplementary Fig. 9c). Altogether, these findings indicate that Dach1 is functionally linked to signaling by EphB4 and ephrin-B2 in the regulation of EC migratory behavior and artery formation.

*DACH1* transcript levels are strongly decreased in vitro after combined knockdown of *SOX7*, *SOX17* and *SOX18* (Fig. 9a), indicating that similar transcriptional programs control Dach1 expression together with other key regulators of arterial development. Strikingly, knockdown of *DACH1* significantly increases *EPHB4* and decreases *EFNB2* transcript levels, thereby controlling the balance between the ligand and its receptor (Fig. 9b). Next, we analyzed Dach1 expression and function particularly in retinal tip cells. Ephrin-B2 expression (nuclear EGFP signal) and nuclear Dach1 immunostaining overlap strongly in morphologically identifiable tip cells in *Efnb2-GFP* reporter mice (Fig. 9c, d). Furthermore, genetic fate tracking of *Esm1-CreERT2*-derived tip cell progeny after a very short labeling period (12 h) shows that 90% of these cells are ephrin-B2 positive (Fig. 9e, f).

As shown above, EphB4 limits arterial specification in tip cell progeny but also limits the migration of these cells into retinal arteries (Fig. 1c, e and Fig. 6f, g). Dach1 was previously shown to facilitate arterial growth in the developing heart[93]. Following *Esm1-CreERT2*-mediated acute inactivation of *Ephb4* (4-OHT at P4.5), nuclear Dach1 is increased in retinal tip cells (Fig. 9g, h). Furthermore, a genetic approach enabling Dach1 overexpression in Esm1+ cells accelerates the migration of GFP+ ECs into arteries without altering the total proportion of GFP+ cells (Fig. 9i, j). Thus, Dach1 promotes the migration of tip cell progeny towards the artery and thereby arterial growth.

Taken together, the results above indicate that sprouting tip cells in the retina contain a population of pre-arterial ECs marked by ephrin-B2, active Notch, Sox17, Dach1, and ERK activation (Fig. 9k). We propose that the balance between ephrin-B2 and its receptor EphB4 helps to determine whether tip cell progeny undergoes arterial specification and migrates into the arterial domain.

## Discussion

Our work shows how cell sorting and fate determination are closely integrated to control developmental artery formation in the postnatal murine retina, one of the most established model systems for angiogenic blood vessel growth. Ephrin-B2 directs tip cell-derived progeny into growing arteries, the domain with high levels of ephrin-B2 expression, whereas the receptor EphB4, which is highly expressed by venous ECs, opposes this process.

Apart from simply mediating cell segregation, however, the balance between ephrin-B2 and EphB4 in ECs also directly affects arterial specification. This involves the modulation of VEGF and Notch signaling, which are known key regulators of arterial fate acquisition[94]. While previous work has shown that ephrin-B2 expression is increased downstream of active Notch[74–76,95], we show here that the ephrin is also acting upstream of Notch signaling. This involves the regulation of VEGF signaling, which has been previously shown to induce the expression of Dll4, the most critical Notch ligand in ECs[95,96]. Ephrin-B2 enhances the internalization of VEGF receptors and downstream activation of MAPK signaling[43,70]. We now show that VEGFR2 internalization and MAPK-controlled arterial specification are increased after loss of EphB4. Dll4 expression is also controlled by Notch signaling and SoxF transcription factors[49,97], which, in turn, regulate the expression of Notch1 and VEGFR2[49,98,99]. Given that VEGF signaling can upregulate the expression of Sox7 and Sox17[99], it becomes clear that artery formation depends on a network of integrated feedback loops involving multiple molecular pathways.

These molecular processes mentioned above are probably influenced by hemodynamic factors and higher shear stress in the arterial branch of the vasculature. Shear stress has been shown to promote Notch activation and Notch itself acts as a mechanosensor in arterial endothelium[76,84,100]. In contrast, Dach1, which has been identified as an important modulator of coronary artery growth in the heart, is downregulated by high laminar flow and, accordingly, its expression is absent in mature, larger caliber arteries[92]. It is therefore likely that Dach1 promotes an early step of arterial specification[93], which is

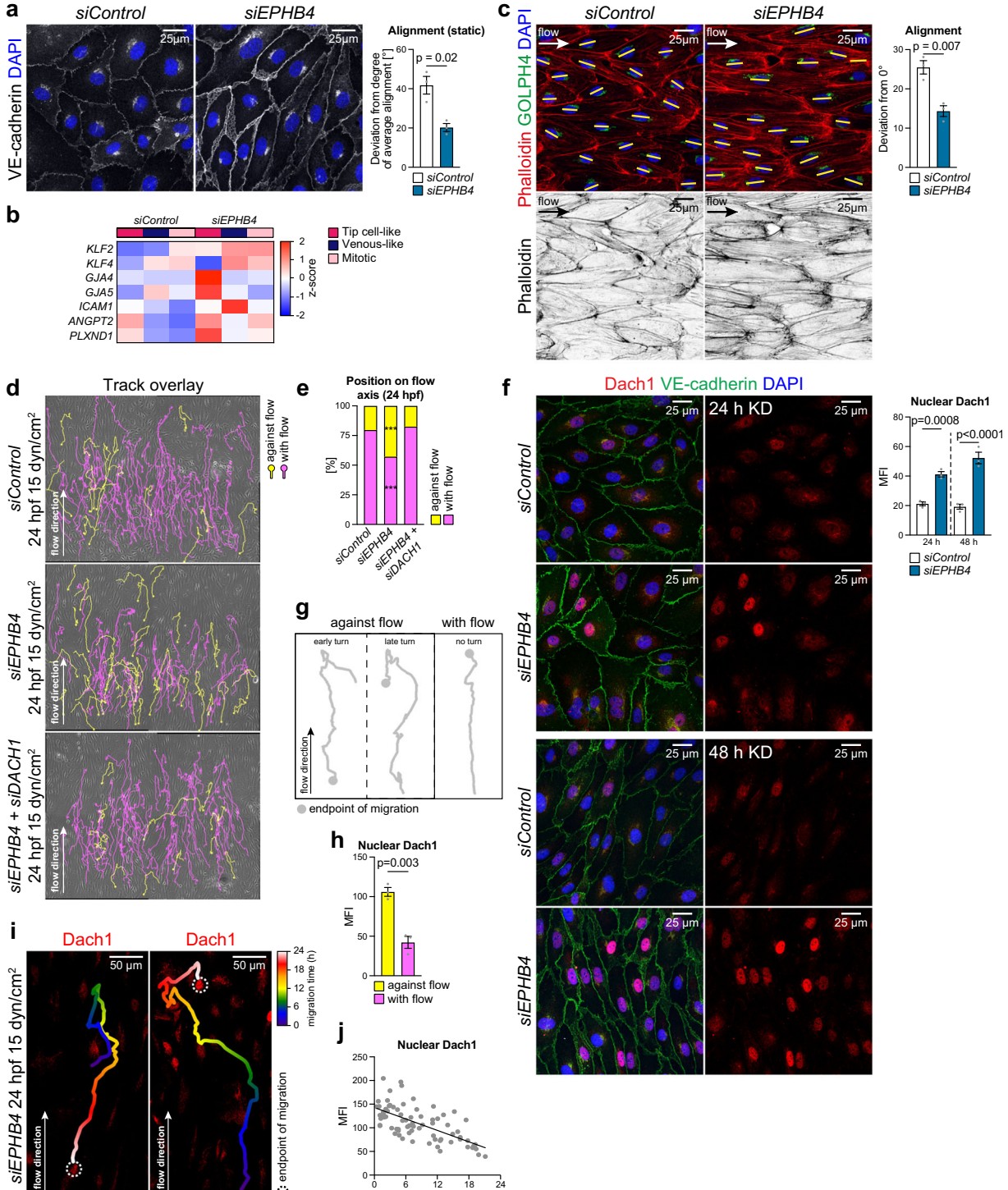

**Fig. 8 | EphB4 regulates EC turning under flow through Dach1. a** *siEPHB4* HUVECs show partial alignment already in static culture. VE-cadherin and DAPI staining of *siControl* and *siEPHB4* cells at 48 h post-transfection. Graph quantifies deviation (degree) from the average alignment direction per field (*n* = 3 experiments). **b** Expression of flow-responsive genes is increased upon *EPHB4* KD compared to control. Data selected from pseudobulk DGE analysis comparing *siControl* and *siEPHB4* cells. **c** Enhanced alignment of *siEPHB4* HUVECs under arterial flow (15 dyn/cm²). Quantitation of deviation from 0° angle calculated between the long axis of each cell and flow direction (*n* = 3 experiments). **d, e** *siEPHB4* promotes EC turning and migration against flow, which requires *DACH1* expression. Still images from 24 h movies acquired for *siControl*, *siEPHB4* and *siEPHB4/DACH1* cells exposed to arterial flow. Graph quantifies proportion of cells migrating with (purple) or

against flow (yellow) (*n* = 3 experiments, 80 tracks/experiment selected randomly). **f** Nuclear Dach1 is increased in *siEPHB4* ECs (*n* = 3 experiments). **g, h** Dach1 levels are increased during migration against flow. Scheme depicts three different cellular trajectories under flow (*n* = 3 experiments, 80 randomly selected tracks/experiment). **i** Representative examples of Dach1 immunostaining of *siEPHB4* cells showing high nuclear Dach1 shortly after turning against flow. **j** Graph indicates strong correlation between the time spent migrating against flow (after turning) and Dach1 nuclear immunosignal (*n* = 3 experiments, 74 tracks in total). *P* values were calculated using two-tailed unpaired *t* test (**a, c, h**) and one-way ANOVA (**e–g**). Data are presented as mean values ± SEM. Source data are provided as a Source Data file.

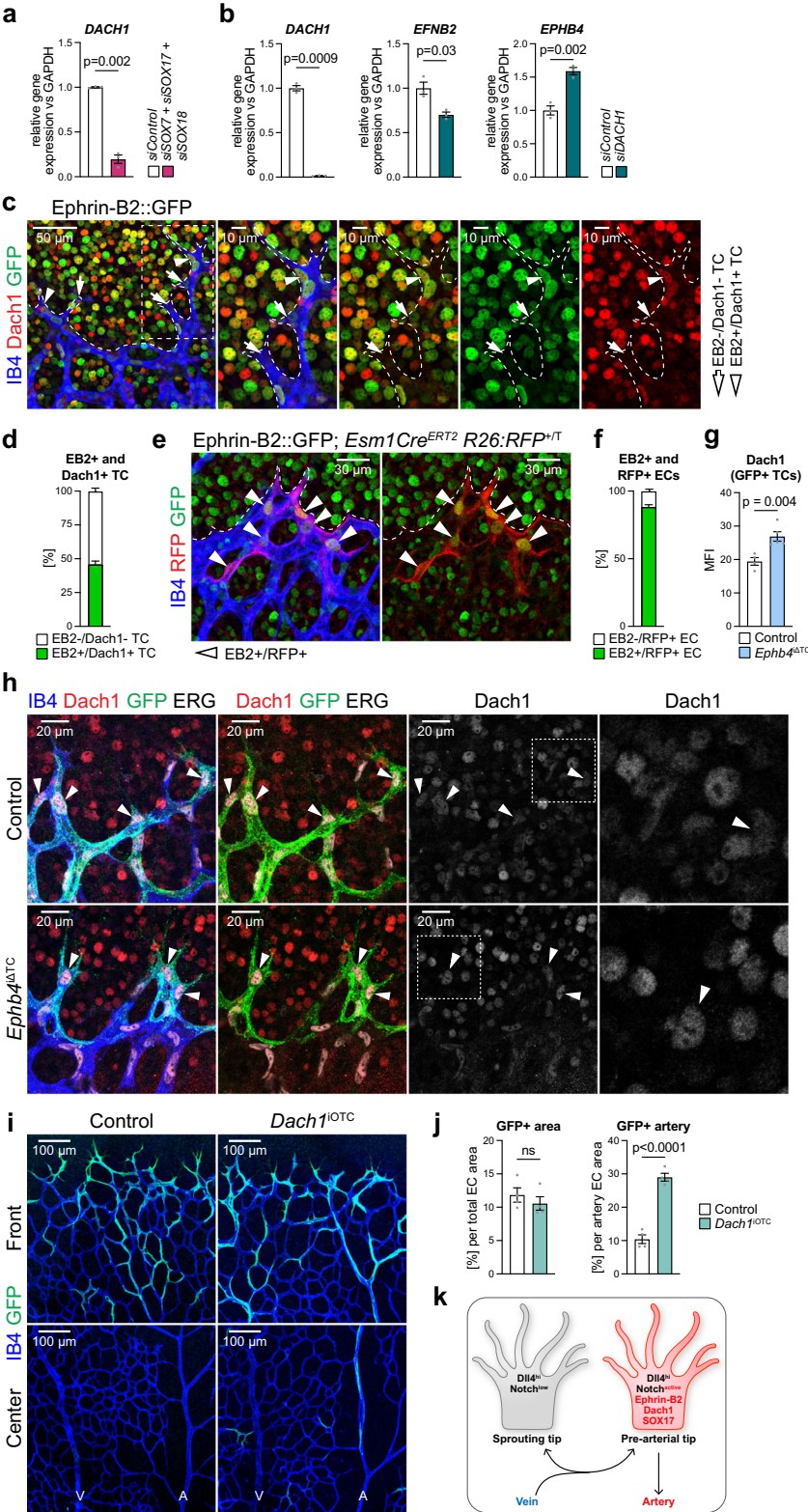

**Fig. 9 | Ephrin-B2 and Dach1 label pre-arterial tip cells in the retina. a** RT-qPCR for *DACH1* showing that SoxF factors positively regulate *DACH1* expression (*n* = 3 experiments). **b** Dach1 regulates *EPHB4* and *EFNB2* expression in opposite directions (*n* = 3 experiments). **c, d** 50% of retinal sprouts are positive for ephrin-B2 and Dach1. IB4, GFP and Dach1 staining of the P6 *Efnb2-H2B-GFP* knock-in reporter retinal vasculature. Graph indicating proportion of ephrin-B2+ and Dach1+ tip cells (*n* = 6 mice). All Dach1+ tip cells are ephrin-B2+. **e, f** 90% of Esm1-derived progeny (RFP+) are ephrin-B2+ (GFP+). High magnification images of *Efnb2-H2B-GFP* knock-in mice combined with *Esm1-CreERT2 R26-RFP* reporter. Graph shows proportion of RFP+ cells being positive or negative for GFP (*n* = 4 mice). **g, h** Dach1 expression is increased in *Ephb4*iΔTC tip cells. IB4, GFP, ERG and Dach1 staining in retinal angiogenic front areas of P6 pups (*n* = 4 control and 5 *Ephb4*iΔTC mice). **i** Acute overexpression of Dach1 in tip cells increases their arterial incorporation. **j** Quantitation of total GFP+ area and of GFP+ area inside arteries per total EC area (*n* = 4 control and 4 *Dach1*iOTC mice). **k** Schematic depiction of features defining pre-arterial tip cells. *P* values were calculated using two-tailed unpaired *t* test (**a**, **b**, **g**, **j**). In vivo experiments were performed with tamoxifen injections at P1-P3 (**e**) or 4-OHT injections at P4.5 (**h**, **i**) following analysis at P6. Data are presented as mean values ± SEM. Source data are provided as a Source Data file.

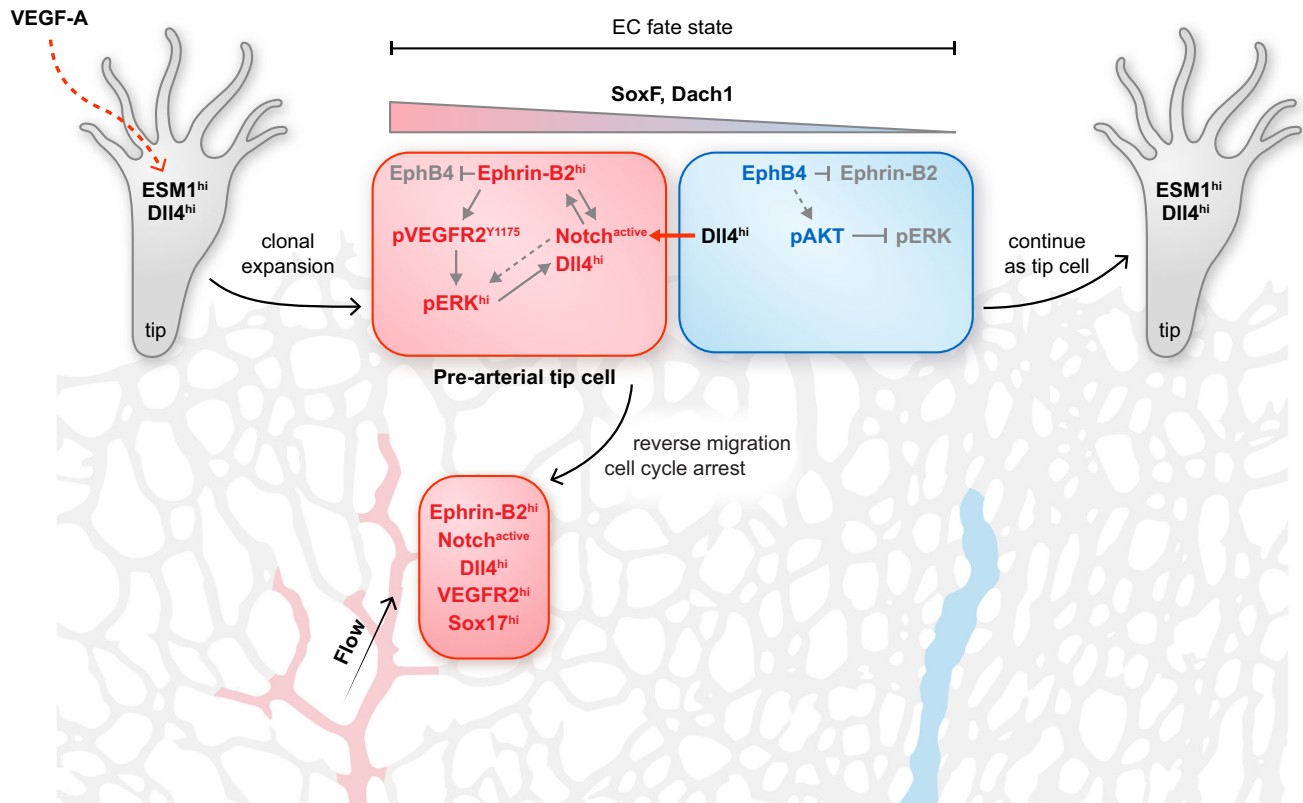

**Fig. 10 | Schematic depiction of key findings.** VEGF-A induces Esm1 expression in endothelial tip cells. High ephrin-B2 expression, which is linked to low EphB4, in tip cell progeny promotes arterial specification, which involves Notch activation, signaling by VEGFR2 and ERK1/2 together with the expression of the transcription factors SoxF and Dach1.

consistent with our own findings showing Dach1 expression in ephrin-B2+ ECs within vascular sprouts.

While the exact function of Dach1 deserves further investigation, the transcription factor promotes EC migration against the direction of flow[92] and, as our data indicate, can promote directional changes during cell migration under flow, a behavior that is also facilitated by loss of EphB4. Moreover, *Dach1* gain-of-function mice[93], like *Ephb4* loss-of-function mutants, show an increased frequency of arteriovenous crosses in retinal vasculature, which supports that these molecules cooperate to control the segregation of arterial and venous territories.

SoxF transcription factors control the expression of Dach1, either directly or indirectly through other SoxF-controlled modulators, but Dach1 also influences the balance between ephrin-B2 and EphB4 in ECs. Loss of EphB4 and the concomitant upregulation of ephrin-B2 or the genetically controlled overexpression of the ligand in *R26-ephrin-B2*[iOTC] tip cells are sufficient to enhance arterial specification. Thus, the balance between ephrin-B2 and EphB4 functions as a potent amplifying system controlling AV fate, in which arteries (high ephrin-B2) and veins (high EphB4) represent the extreme poles, whereas sprouting ECs show mixed expression of the two molecules[43,70]. We propose that pre-arterial tip cells display high levels of ephrin-B2 and Notch, active MAPK signaling together with elevated expression of nuclear Dach1 and SoxF transcription factors (Fig. 10).

Reciprocal regulation of EphB receptors and ephrin-B1, another B-class ligand that is closely related to ephrin-B2, is highly relevant for craniofacial development and other patterning processes in the growing skeletal system. Mosaic inactivation of the X-linked *Efnb1* gene in female heterozygous mutant mice leads to strong upregulation of EphB receptors in ephrin-B1-deficient cells, whereas cell clusters utilizing the *Efnb1* wild-type copy upregulate the ligand and suppress receptor expression. The resulting complementary pattern of ligand and receptor expression prevents cell mixing processes and leads to the generation of ectopic tissue boundaries, interfering with normal skeletal morphogenesis[101,102]. Consistent with these striking findings, mutations in the human *EFNB1* gene have been identified as the cause of Craniofrontonasal syndrome (CFNS), a human X-linked disease that has been very puzzling for geneticists because it involves patterning defects in heterozygous female carriers but not their hemizygous fathers[103,104]. Our new results indicate that reciprocal regulation of Eph receptors and ephrin ligands occurs in different organ systems to couple cell sorting and differentiation processes.

The interdependent function of ephrin-B2 and EphB4 in the murine vascular system and the AV crossing phenotype detected in *Ephb4* mutant retinas might be relevant for human diseases. Investigations of retinal vessels in human patients using non-invasive retinal imaging technology created the opportunity to indirectly assess the general health status, in particular of the cerebral vasculature[47]. Retinal AV nicking, a vascular abnormality characterized by reduced vein caliber due to a crossing artery[45], appears to be a strong predictor of cardiovascular diseases, such as hypertension or stroke[105]. In addition, branch retinal vein occlusion (BRVO), a common retinal vascular disease, can further develop as a focal occlusion at the AV crossing point[46,106,107].

Whole-exome sequencing has shown that loss-of-function mutations in the human *EPHB4* gene cause a rare disease termed Capillary Malformation-Arteriovenous Malformation (CM-AVM2)[36,108], characterized by cutaneous capillary malformations and AV defects such as arteriovenous fistulas. Mutations in human *EPHB4* are also associated with venous valve aplasia, a disease leading to venous reflux, pain and edema formation, and venous aneurysmal malformations[37,38]. Furthermore, in vitro experiments indicate that alterations in Dll4-Notch and VEGF signaling but also increased EphB4 kinase activation are

associated with loss of CCM3 function, one of the causes of a group of diseases termed cerebral cavernous malformations (CCMs)[109,110]. These results raise the possibility that AV malformations involve alterations compromising the interplay between ephrin-B2 and EphB4 or the balanced expression of these two molecules. Our findings might therefore enable mechanistic insights into the etiology of certain vascular diseases and could also facilitate the development of new therapeutic strategies.

## Methods

All animal experiments were performed according to the institutional guidelines and laws, approved by local animal ethical committee and were conducted at the Max Planck Institute for Molecular Biomedicine with necessary permissions (Az 81-02.04.2019.A114, Az 85-02.04.2015.A185, Az 81-02.04.2022.A390) granted by the Landesamt für Natur, Umwelt und Verbraucherschutz (LANUV) of North Rhine-Westphalia, Germany.

### Genetic mouse models and pharmacological treatments

A summary of all genetic mouse models used in this study can be found in supplementary table S1. To inactivate *Ephb4* in the postnatal endothelium, *Ephb4*[lox/lox] mice[111] and *Cdh5(PAC)-CreERT2*[+/T] transgenic mice[43] were interbred. Male *Ephb4*[lox/lox], *Cdh5-CreERT2*[+/T] offspring were mated with *Ephb4*[lox/lox] female mice to generate progeny for analysis. For Cre activation, pups were injected from P1 to P3 with 50 µg tamoxifen (Sigma, T5648), followed by analysis at P6.

For TC-specific *Ephb4* loss-of-function experiments, mice homozygous for the *R26-mTmG* transgene[48] were interbred with the *Ephb4*[lox/lox] line to generate *Ephb4*[+/lox], *R26-mTmG*[T/T] offspring. Additionally, *Ephb4*[lox/lox] mice were interbred with *Esm1-CreERT2*[+/T] transgenic animals[34] to generate *Ephb4*[+/lox], *Esm1-CreERT2*[+/T] offspring. These mice were interbred with *Ephb4*[+/lox], *R26-mTmG*[T/T] animals to obtain *R26-mTmG*[+/T], *Esm1-CreERT2*[+/T] controls and *Ephb4*[lox/lox], *R26-mTmG*[+/T], *Esm1-CreERT2*[+/T] mutants. Pups were injected daily from P1 to P3 with 50 µg tamoxifen and analyzed at P6. For acute deletion experiments, pups were injected with a single dose of 50 µg 4-hydroxy-tamoxifen (4-OHT; Sigma, H7904) at P4.5 and analyzed at P6.

To achieve pan-endothelial and TC-specific deletion of *Efnb2*[112], similar mating strategies as described above were used. Pups were injected with 50 µg tamoxifen from P1 to P3 and retinas analyzed at P6.

For TC-specific dnMaml1 overexpression experiments, we mated *R26-dnMaml1*[+/T] (ref. 77) and *R26-mTmG*[T/T], *Esm1-CreERT2*[+/T] mice. Offspring was daily injected from P1 to P3 with 50 µg tamoxifen and analyzed at P6. For combined inactivation of *Ephb4* and overexpression of *dnMaml1* in TCs, *Ephb4*[lox/lox], *R26-dnMaml1*[+/T] females were bred with *Ephb4*[lox/lox], *R26-mTmG*[T/T], *Esm1-CreERT2*[+/T] males. Cre activation was induced by daily injection of 50 µg tamoxifen from P1 to P3. Retinas were analyzed at P6.

For combined TC-specific overexpression of active Notch and deletion of *Efnb2*, *Gt(ROSA)*[26Sor tm1(Notch1)dam] homozygous mice (*NICD*[lox/lox])[113] were interbred with *Efnb2*[lox/lox], *Esm1-CreERT2*[+/T]. Further, *NICD*[lox/lox], *Efnb2*[+/lox], *Esm1-CreERT2*[+/T] male mice were crossbred with *NICD*[lox/lox], *Efnb2*[+/lox] females, to obtain *NICD*[lox/lox], *Efnb2*[lox/lox], *Esm1Cre-ERT2*[+/T] double mutants and *NICD*[lox/lox], *Esm1-CreERT2*[+/T] single mutants. Pups were injected daily from P1 to P3 with 50 µg tamoxifen and retinas were analyzed at P6.

For overexpression of ephrin-B2 in TCs, *Esm1-CreERT2*[+/T] transgenics were combined with *R26-ephrin-B2-GFP*[T/T] mice. These mice carry a transcriptional stop cassette flanked by *loxP* sites, which is followed by the wild-type sequence of ephrin-B2 and an *frt* flanked IRES enhanced GFP (EGFP) construct. A map of the transgenic construct is shown in Supplementary Fig. 6a. *R26-ephrin-B2-GFP*[+/T], *Esm1-CreERT2*[+/T] offspring were further bred with *R26-ephrin-B2-GFP*[T/T], to obtain heterozygous and homozygous mutants for the *R26-ephrin-B2-GFP* allele in combination with *Esm1-CreERT2*. Pups were injected with 50 µg

tamoxifen from P1 to P3. Heterozygous and homozygous mutant littermates were analyzed at P6.

TC-specific overexpression of Dach1 was achieved by interbreeding of *R26-Dach1*[+/T] (ref. 93) and *R26-mTmG*[T/T], *Esm1-CreERT2*[+/T] mice. Pups were injected daily with 50 µg tamoxifen from P1 to P3 and retinas were analyzed at P6.

For visualization of endogenous ephrin-B2 expression, *Efnb2-GFP* knock-in reporter mice[55] were used. These mice carry a complementary DNA encoding a fusion construct of histone H2B and enhanced GFP, which has been inserted into the endogenous *Efnb2* locus. As homozygous *Efnb2-GFP* mice are not viable, *Efnb2-GFP*[+/T] animals were interbred with C57BL/6 J wild-type mice to obtain heterozygous offspring, which was analyzed at P6.

Lineage tracing experiments of Esm1-derived cells were performed using *R26-mTmG*[+/T], *Esm1-CreERT2*[+/T] offspring generated by mating female mice homozygous for the *R26-mTmG* transgene with *Esm1-CreERT2*[+/T] males. Cre recombination was achieved by injection of single dose 4-OHT at P3 or P5, followed by analysis at P6.

Immunostaining of wild-type retinas was performed using C57BL/6J P6 pups.

For Notch inhibition, DAPT (Merck Millipore, 565770) was dissolved in 10% ethanol, 90% peanut oil and 0.1 mg/g body weight were intraperitoneally (IP) injected at P5.5 and pups were analyzed 14 h later at P6. For robust Notch inactivation mice were injected at P5 twice, 24 h and 14 h prior analysis at P6. For vehicle treatment 10% ethanol, 90% peanut oil of similar volumes was used.

To inhibit ERK phosphorylation, SL327 (Selleckchem, S1066) was dissolved in 10% DMSO, 90% corn coil and IP injected into pups at a dosage of 120 mg/kg body weight 36 h and 12 h prior to dissection at P6[80].

To block VEGF, pups were IP injected with VEGF blocking antibody (Creative Biolabs, HPAB-0330CQ-F(E), clone mAb G6-31) using a dosage of 5 mg/kg body weight.

To assess EC proliferation, P6 pups were injected IP with EdU (100 µg) 90 min before dissection[17]. EdU-incorporated cells in the retina were detected using the Click-iT EdU Imaging Kit (Thermo Fisher Scientific, C10340).

All animal procedures were performed in compliance with the relevant laws and institutional guidelines, were approved by local animal ethics committees and were conducted at the MPI for Molecular Biomedicine with permissions granted by the Landesamt für Natur, Umwelt und Verbraucherschutz (LANUV) of North Rhine-Westphalia. Animals were combined in groups for experiments irrespective of their sex.

### Whole-mount immunohistochemistry of mouse retinas

Retina immunostaining was performed as previously described[114] with minor modifications. Briefly, eyes were collected and immediately fixed in 4% PFA in PBS. Dissected retinas were incubated in blocking buffer (1% BSA, 0.3% Triton X-100 in PBS) for 30 min at room temperature (RT), rinsed with modified Pblec buffer (1 mM CaCl2, 1 mM MgCl2, 1 mM MnCl2, 0.4% Triton X-100 in PBS), and subsequently incubated with primary antibodies diluted in modified Pblec buffer overnight at 4 °C. After washing in blocking buffer diluted 1:1 with PBS and three washes in PBS, retinas were incubated for 1 h at RT with secondary antibodies diluted in blocking buffer. After similar washes as described above, retinas were refixed with 4% PFA for 20 min at RT and washed twice with PBS prior to mounting with Fluoromount-G (SouthernBiotech, 0100-01).

Immunostaining of pERK1/2 was performed as described above, with some modifications. Eyes were fixed in 4% PFA for 20 min at RT. Dissected retinas were fixed in 4% PFA for 1 h at 4 °C, washed three times with PBS and further fixed in 100% methanol at −20 °C for 30 min. After washing with PBS, retinas were permeabilized in 0.3%

PBST for 30 min at RT and blocked for 1 h at RT in blocking buffer (0.3% PBST, 10% donkey serum). Primary antibody incubation was performed in staining buffer (0.3% PBST, 10% donkey serum, 0.2% BSA) overnight at 4 °C. Next, retinas were washed once with PBST and twice with PBS, followed by incubation with secondary antibodies diluted in staining buffer for 2 h at RT. After similar washes as described above, retinas were mounted with Fluoromount-G. Primary and secondary antibodies that were used are listed in supplementary table S2.

### Alkaline phosphatase construct transfection, quantitation and staining

Transfection of HEK 293 cells (DSMZ, ACC305) with the EphB4-AP fusion construct was performed using the CalPhos Mammalian Transfection Kit (Clontech, 631312) according to the manufacturer's protocol. Briefly, cells were transfected with 18 µg plasmid DNA per 10 cm dish diluted in the transfection solution and incubated for 20 h at 37 °C in a 5% $CO_2$ incubator. Cells treated without plasmid DNA served as Mock-transfected control. Next, medium was changed for Opti-MEM and supernatants were harvested after 72 h. Supernatants were centrifuged at 3000 g for 15 min at 4 °C to remove cell debris, supplemented with 20 mM HEPES pH 7 and 0.1% $NaN_3$ before sterile filtering. Alkaline phosphatase activity was quantified using the Alkaline Phosphatase Assay Kit (Colorimetric) (Abcam, ab83369) following the manufacturer's guidelines.

In vivo alkaline phosphatase detection was performed on P6 retinas. Pups were prewarmed at 37 °C for at least 30 min. Enucleated eyes were prefixed in 4% PFA for 6 min at room temperature, rinsed three times with ice-cold PBS before retina dissection. Next, dissected retinas were further fixed in Dent's fixative (20% DMSO, 80% methanol) for 30 min at room temperature. After several PBS washes, retinas were incubated with 30 nM AP fusion protein + 10% FCS overnight at 4 °C. Following additional PBS washes, endogenous alkaline phosphatase activity was quenched by incubating retinas at 65 °C for 30 min. Following multiple washes in AP staining buffer (100 mM NaCl, 5 mM $MgCl_2$, 100 mM TrisHCl pH 9.5), retinas were incubated with BCIP/NBT substrate solution (Sigma Aldrich, B5655) for 3 h at RT. Color-reaction was stopped by washing with PBS + 30 mM ETDA. Subsequently, retinas were incubated with IB4-488 antibodies diluted in modified Pblec buffer without Triton X-100 (1 mM $CaCl_2$, 1 mM $MgCl_2$, 1 mM $MnCl_2$ in PBS), for 2 h at room temperature. Retinas were washed with PBS + 30 mM EDTA before mounting with Fluoromount-G.

### FACS of brain endothelial cells

P6 mouse brain cortices were dissected and digestion was performed in enzymatic digestion buffer (DMEM, 100 µg/ml Liberase TM (Sigma, 5401119001), 100 U/ml DNase I (Worthington, LK003170)) at 33 °C. Digestion was stopped with FACS buffer. Dissociated cells were centrifuged for 5 min at 4 °C and the pellet was resuspended in 1 ml cold FACS buffer. Myelin was removed by addition of 1.7x volume 22% BSA in DMEM (Carl Roth, 8076.2) and cells were centrifuged at 1000 g for 12 min at 4 °C. Cells were subsequently filtered through a 50 µm Cell Trics filter (Sysmex, 04-0042-2317). After centrifugation, cells were stained with antibodies in FACS buffer for 30 min on ice. While antibodies anti-CD45, anti-Ter119, anti-CD140a, anti-CD140b, anti-CD326, anti-podoplanin and anti-CD208 were used to deplete the blood cells, mural cells, epithelial cells, antibodies anti-CD31 were employed to positively select endothelial cells (supplementary table S5). DAPI labeling was used for the exclusion of dead or damaged cells by FACS gating. Endothelial cells were sorted using FACS Canto flow cytometer and analysed using FACS Diva software (BD Bioscience, Vers 6.0). Cells were lysed in Monarch DNA/RNA Protection Reagent (NEB, T2010).

### FACS of retinal endothelial cells

P8 mouse retinas were dissected in DMEM (Thermo Fisher Scientific, 31053028) supplemented with 2% FCS. Digestion was performed in enzymatic digestion buffer comprised of Earle's Balanced Salt Solution (EBSS, Wothington, LK003188), 20 U/ml Papain (Worthington, LK003176) and 80 U/ml DNase I (Worthington, LK003170) at 37 °C. After further mechanical dissociation, cells were centrifuged for 5 min at 4 °C. Digestion was completely stopped by resuspension of the pellet in EBSS supplemented with Albumin-Ovomucoid inhibitor solution (Worthington, LK003182) and 100 U/ml DNase I. Cells were subsequently filtered through a 50 µm Cell Trics filter (Sysmex, 04-0042-2317) in FACS buffer (2% FCS in DMEM) and centrifuged for 5 min at 4 °C. Pellet was resuspended in FACS buffer.

For *Esm1-CreERT2*$^{+/T}$ lines with *R26-mTmG* transgene, GFP+ cells were FACS-sorted based on endogenous GFP fluorescence.

For transgenic lines without reporter allele, cells were further stained with antibodies in FACS buffer for 20 min on ice. While antibodies anti-CD45, anti-Ter119, anti-CD140a, anti- CD140b were used to deplete blood cells, mural cells and other non-vascular cells, antibody anti-CD31 was employed to positively select ECs. Cells were washed with FACS buffer and resuspended in FACS buffer. DAPI was added before sorting for the exclusion of dead or damaged cells by FACS gating. Endothelial cells were sorted using FACS Canto flow cytometer and analyzed using FACS Diva software (BD Bioscience, Vers 6.0). Cells were lysed in 1x DNA/RNA Shield Reagent (Zymo, R1110) and RNA extraction was performed using the Quick-RNA MicroPrep (Zymo, R1050).

### Cell culture

Human umbilical venous endothelial cells (HUVECs, Thermo Fisher, C0035C or Provitro, 1210111) and human umbilical arterial endothelial cells (HUAECs, Provitro, 1210112) were cultured in ECPM (endothelial cell proliferation medium) (Provitro, 211 0001) supplemented with 7% FCS, 10 µg/ml heparin, 10 ng/ml EGF, 5 ng/ml bFGF, 5 ng/ml R3-IGF-1, 0.5 ng/ml VEGF, 1 µg/ml ascorbic acid and 0.2 µg/ml hydrocortisone, and kept in a humidified incubator at 37 °C, 5% $CO_2$. Cells were seeded into six-well plates coated with 0.1% gelatin for protein extraction, into 12 well plates for RNA extraction or into µ-Slide 8 well (ibidi, 80826) for immunostaining.

Human retinal endothelial cells (HRECs, Innoprot, P10880) were cultured in ECM (endothelial cell medium) (Innoprot, P60104) supplemented with 5% FBS and endothelial cell growth supplement mix at 37 °C and 5% $CO_2$. Fibronectin-coated plates were used for their culture. Passages three to five were used for this study.

### RNA interference in HUVECs and HUAECs

Cells were transfected with siRNA using Lipofectamine RNAiMAX (Invitrogen, 13778) according to the manufacturer's protocol. siRNAs: Negative control (Ambion, 4390844), *EPHB4* s244 (Ambion, 4390824), *EFNB2* s4514 (4392420), *SOX7* s38028 (4392420), *SOX17* 30105 (AM16708), *SOX18* s28927 (4392420), *CTNNB1* s437 (4392420), *SMAD4* s8403 (4392420) and *DACH1* s3899 (4392420). After 4 h the medium was changed to fresh culture medium and analysis was performed 48 h post-transfection.

### Cell culture stimulations and inhibitor treatments

For VEGF-A stimulation experiments, cells were starved for 3.5 h in basal medium and then incubated with basal medium containing 50 ng/ml recombinant human VEGF-A$_{165}$ (RELIATech, 300-076) at 37 °C for the indicated time points. Following stimulation, cells were processed for protein isolation.

For Fc stimulation, cells were starved for 3.5 h in basal medium and subsequently incubated with basal medium containing 4 µg/ml pre-clustered control human IgG1 Fc (R&D Systems, 110-HG) or mouse EphB4/Fc (R&D Systems, 466-B4) proteins for 30 min. Pre-clustering of proteins was performed immediately before treatment by incubation with 0.2 µg anti-human IgG (Fc specific) (Jackson Immuno Research, 109-005-098) per µg of Fc protein at 37 °C at a final concentration of

10 μg/ml. Inhibitor treatment was performed simultaneously with Fc stimulation. Inhibitors were dissolved in DMSO and used at the following final concentrations: U0126 (10 μM, Promega, V1121), Ki8751 (25 nM, Selleckchem, S1363) and DBZ (0.08 μM, Sellekchem, S2711). DMSO was used as vehicle treatment.

Notch inhibition in siRNA-treated HUVECs was performed using DBZ (Sellekchem, S2711), which was dissolved in DMSO and added to the culture medium 16 h before analysis at a concentration of 0.08 μM. DMSO was used as a control treatment. A similar experimental design was used for ERK inhibition in siRNA-treated HUVECs, using U0126 at a concentration of 10 μM for 16 h.

### In vitro EdU incorporation assay
To assess EC proliferation in vitro, cells were incubated with EdU (20 μM) in a complete medium for 60 min at 37 °C. EdU-incorporated cells were detected using the Click-iT EdU Imaging Kit (Thermo Fisher Scientific, C10340).

### RNA isolation and RT-qPCR
RNA extraction was performed using the Monarch Total RNA Miniprep Kit (NEB, T2010S) according to the manufacturer's protocol. Prior to quantitative Real-Time (qRT) PCR analysis, cDNA was synthesized using the LunaScript RT SuperMix Kit (NEB, E3010). Obtained cDNAs were mixed with primers and Luna Universal qPCR Master Mix (NEB, M3003) and analyzed using the CFX96 Touch™ Real-Time PCR System (Biorad). Gene expression levels were normalized to the VIC-conjugated GAPDH probe and quantified using the 2-ΔΔCT method. All Taqman probes used are listed in supplementary table S3.

### Protein isolation and western blotting
For immunoblotting, cells were washed twice with ice-cold PBS containing 1 mM PMSF and lysed on ice in modified RIPA buffer (20 mM Tris HCl, pH 8.0, 150 mM NaCl, 0.5% Triton X-100, 0.1% SDS, 0.1% Na-DOC, 2 mM EDTA, Halt Protease Inhibitor Cocktail (Thermo Scientific, 78429) and Phosphatase Inhibitor Cocktail Set V (EMD Millipore, 524629) for 20 min at 4 °C. Cell lysates were centrifuged for 10 min at 4 °C and protein concentrations of the supernatants were quantified using the BCA Protein Assay Kit (Pierce, 23225). After adding sample loading buffer to the lysates, samples were boiled at 95 °C for 5 min. Proteins were separated by SDS-polyacrylamide gel electrophoresis and blotted onto Immobilon-P Polyvinylidene fluoride (PVDF) membranes (Millipore, IPVH00010). Membranes were probed with specific primary antibodies followed by peroxidase-conjugated secondary antibodies. Antibodies that were used for western blotting are listed in supplementary table S4. Blots were quantified using Fiji[115].

Coomassie staining of proteins was performed using SimplyBlue SafeStain (Thermo Scientific, LC6060) according to the manufacturer's protocol.

### Endothelial cell immunostaining
Immunostaining of HUVECs was performed as previously described[116]. Briefly, cells were fixed with 4% PFA for 10 min. Following additional incubations with 4% sucrose/PBS and 50 mM NH4Cl/PBS, cells were permeabilized in ice-cold 0.1% Triton X-100/PBS for 5 min at 4 °C. After washing with PBS, cells were blocked with blocking buffer (4% donkey serum, 2% BSA in PBS) for 30 min. Primary antibodies were diluted in blocking buffer and incubated with the cells for 1 h at room temperature. Following three PBS washes, cells were incubated with secondary antibodies diluted in blocking buffer for 30 min, washed with PBS and Fluoromount-G was applied in each μ-Slide well. Primary antibodies that were used to stain cultured ECs are listed in supplementary table S5.

### Antibody feeding assay
For the analysis of VEGFR2 internalization, HUVECs were starved for 2.5 h in a basal medium. Afterwards, cells were incubated with ice-cold

blocking buffer (1% BSA in basal medium) for 30 min at 4 °C and subsequently incubated with VEGFR2 antibody diluted in blocking buffer for 1 h at 4 °C. Next, cells were washed three times with ice-cold blocking buffer and stimulated with prewarmed basal medium containing 50 ng/ml VEGF-A at 37 °C for 15 min. After the incubation, the unbound ligand was removed by rinsing cells three times with a basal medium. Next, cells were fixed in 4% PFA for 10 min at room temperature, incubated with 4% sucrose for 15 min and additionally incubated with 50 mM NH4Cl for 10 min at room temperature. After blocking with blocking buffer (2% BSA, 4% donkey serum in PBS) for 30 min, the surface receptor was detected by incubation with Alexa-Fluor 488-coupled secondary antibodies diluted in blocking buffer for 30 min at room temperature. Following three PBS washes, cells were permeabilized using ice-cold 0.1% Triton X-100 in PBS for 2 min at 4 °C. After several PBS washes, cells were further incubated in a blocking buffer. Subsequently, internalized receptor was detected by incubation with Alexa-Fluor 546-coupled secondary antibodies. DAPI and Phalloidin staining were added in this step. After several PBS washes, Fluoromount-G was added to each μ-Slide well.

### Confrontation assay of HUVECs and HUAECs
To investigate the adhesive and repulsion properties of HUVECs and HUAECs, a confrontation assay was performed. To distinguish between cell populations, cells were labeled overnight using CellTracker Green CMFDA Dye (Thermo Fisher, C7025) or CellTracker Orange CMRA Dye (Thermo Fisher, C34551) at final concentrations of 0.5 μM diluted in ECPM. Next, cells were trypsinized and seeded into Culture-Insert 4 well in 35 mm μ-Dishes (ibidi, 80466) and incubated for 2 h at 37 °C. Subsequently, the insert was removed, fresh medium was applied and cells were incubated for 48 h at 37 °C in a 5% CO2 incubator before 24 h imaging.

### VEGFR2 degradation assay
For the analysis of VEGFR2 degradation, HUVECs were starved for 3.5 h in basal medium. While 1 dish was treated with vehicle (DMSO), the other dishes were pre-treated with basal medium containing 10 μM cycloheximide (CHX, Sigma Aldrich, 1810) and 50 ng/ml VEGF-A for 30 min at 37 °C. After stimulation, cells were washed three times with basal medium. Next, cells were left in basal medium for 1 h at 37 °C to allow VEGFR2 recovery or continued to be incubated in basal medium containing CHX to block protein synthesis. Finally, cells were lysed and samples prepared for immunoblotting as described above.

### Endothelial cell culture under laminar flow conditions
For migration experiments under flow, HUVECs were seeded into 0.4 mm μ-Slide I Luer (ibidi, 80176) coated with 0.1 % gelatin 24 h after siRNA knockdown and exposed to 15 dyn/cm² shear stress for 24 h using an ibidi Pump System (ibidi, 10902). Live imaging was performed with a 10 min interval.

To analyze the early flow response and alignment of endothelial cells under laminar flow, HUVECs were exposed to 15 dyn/cm² shear stress for 12 h and were immediately processed for immunostaining as described above.

### Luciferase assay
For luciferase assays, 1 kilobase predicted sequence via JASPAR was cloned upstream to the minimal promoter of pGL4-UBCmin[20,117]. The regions were amplified from HUVECs using the following primers:

*EFNB2*_reg. 1-fwd: 5'-tctggcctaactggccggtacCTGGAAATGTTGG AGTCATCTGCATG-3';

*EFNB2*_reg. 1-rev: 5'-atcggcgttcaccgccccatccTTTTTTTAATTCAG-CAGAGAACAGTAATGCCTAGC-3';

*EFNB2*_reg. 2-fwd: 5'-tctggcctaactggccggtacTACAGACAAGAGGTT TTAATATAAATCCACCAATTG TTC-3';

*EFNB2*_reg. 2-rev: 5′-atcggcgttcaccgccccatccATCTGTATTGGTATG GTGTTCTGCAAT-3′;

*EFNB2*_reg. 3-fwd: 5′-tctggcctaactggccggtacTAGCTAGATAAG AAAAAAATGTGGAGATTTGAAGAG TAAGG-3′;

*EFNB2*_reg. 3-rev: 5′-atcggcgttcaccgccccatccCCACAGCAGACCTA GCCCT-3′;

*EFNB2*_reg. 4-fwd: 5′-tctggcctaactggccggtacTCTACATATGTGTT ATGTATGCATACATACTATCTTT ACTATTGC-3″;

*EFNB2*_reg. 4-rev: 5′-atcggcgttcaccgccccatccTGTCAGGGTTGCG TTATCACTGAG-3′;

*EPHB4*_reg. 1-fwd: 5′-tctggcctaactggccggtacCCTGGGTCAGAGAG AGGCTG-3′;

*EPHB4*_reg. 1-rev: 5′-atcggcgttcaccgccccatccAGAAGGGCCCCAGA GGAG-3′;

*EPHB4*_reg. 2-fwd: 5′-tctggcctaactggccggtacTCCCGCCGAGGC CT-3′;

*EPHB4*_reg. 2-rev: 5′-atcggcgttcaccgccccatccCGAGGGGAAGAGG AAGGCC-3′;

Amplified regions were cloned via Hifi into KpnI and XhoI digested pGL4-UBCmin. HUVECs were starved for 4 h and transfected for 45 min with *EFNB2* and *EPHB4* constructs using Lipofectamin 3000 and Plus Reagent (Invitrogen, L3000015). Cells were reseeded in complete medium. Post 24 h cells were lysed and luciferase activity was measured with Bright Glo Assay System (Promega, E2620) as per manufacturers protocol. For normalization, values of firefly luciferase were normalized to basal construct activity.

### Image acquisition and analysis

Overview images of retinas were obtained using a fluorescence microscope (Leica MZ16 F) and confocal image acquisition was performed using a Zeiss confocal microscope LSM780 and LSM880. Processing of microscope images as well as analysis of different parameters of the retinal vasculature was done using Fiji. Analysis of morphological features of the retinal vasculature was done as previously described[114].

For the quantitation of vessel progression, the ratio between the distance from the optic nerve to the sprouting front and the length of the retina leaflet was calculated. The mean of all leaflets was obtained per retina and compared between control and mutant groups.

The quantitation of the EC area refers to the absolute vessel area measured by thresholding IB4 signal of whole retinas. The values were normalized to the control and compared to the mutant groups.

Branchpoints were measured by manually counting all branching points in a field of view (350 × 350 μm) of the capillary plexus between artery and vein. The mean of each retina was obtained and compared between control and mutant.

Sprouts and filopodia were manually counted on high-resolution images. Comparable front areas were imaged in each leaflet. While filopodia were counted per tip cell, sprouts were counted per angiogenic front vessel length. The mean for each retina was calculated and compared between the control and mutant groups.

To quantify artery progression the ratio of the artery length to vessel length from the optic nerve to the vascular front was calculated. The progression of all arteries within a retina was averaged and compared between control and mutant groups.

The number of arterial branches were counted manually and averaged for each retina.

The fraction of the arterial EC area was calculated by the ratio of the arterial EC area (measured by thresholding IB4) to the whole EC area. Values were normalized to the control and compared between control and mutant groups.

Sox17 mean fluorescence intensity (MFI) was calculated by thresholding IB4 signal to outline the vascular area. Specific ROIs were drawn for the artery, as well as for the vascular front and MFI was calculated using ImageJ. To quantify the ratio of Sox17 high expressing

cells and total ERG+ ECs, Sox17 signal was thresholded and nuclei were counted. Similarly, the ERG signal was thresholded and the total EC nuclei number was counted per retina. The ratio of the total number of Sox17 high nuclei and total ERG+ nuclei was calculated and compared between control and mutant groups.

For quantitation of total GFP positive EC area, the GFP area was measured by thresholding and divided by the total EC area, similarly measured by thresholding IB4 signal. Calculations were done for each leaflet and the mean was compared between control and mutant groups. Likewise, for the quantification of GFP positive area per arterial EC area, IB4 and GFP signal of arteries were thresholded to calculate their corresponding area. The ratio was calculated for each artery and the mean of each retina was compared between control and mutant groups.

Additionally, the distribution of GFP+ cells was calculated by quantifying absolute values for total GFP positive area, front GFP positive area, capillary GFP positive area and arterial GFP positive area by thresholding GFP signal. Next, distribution ratios were calculated by dividing front, capillary or arterial GFP values by total GFP values.

The distribution of ephrin-B2- or NICD-GFP-positive cells was counted manually and averaged for each retina.

Endothelial cell proliferation of recombined cells was measured by manually counting the number of EdU/GFP positive endothelial nuclei (ERG positive) and dividing by the number of GFP positive endothelial nuclei. Comparable regions of the front vasculature were acquired of each leaflet. The mean of each retina was calculated and compared between the control and mutant groups.

The number of ephrin-B2 positive and Dach1 positive TCs was counted manually in Fiji using four equal regions of the front vasculature per retina. Averages were calculated per retina.

To quantify the mean fluorescence intensity of immunostainings in the retinal vasculature, a mask of the blood vessels or of the GFP positive area was generated by thresholding IB4 or GFP signal, respectively. These masks were then used to specifically measure immunostaining intensities in the areas of interest. Comparable regions were quantified for each leaflet. The mean for each retina was calculated and compared between the control and mutant groups.

Images of the confrontation assay of HUVECs and HUAECs were acquired using a Zeiss AxioObserver Z1. The relative border length between cell populations was measured in Fiji and normalized to the HUVEC/HUAEC samples.

Confocal images of cultured HUVECs were obtained using a Zeiss confocal microscope LSM780. Counting of p21 or Cyclin D1 positive cells was done manually using at least three representative regions per experiment and condition. The ratio of positive and negative cells was calculated and averaged per experiment and compared between control and knockdown groups.

Quantitation of mean fluorescence intensities of nuclear Dach1 was calculated per HUVEC cell using a nuclear mask generated by thresholding the DAPI signal.

For the quantitation of internalized vs surface VEGFR2 receptor, the ratio of the mean intensities was calculated using Fiji. At least three representative regions were quantified per experiment. The averages of each experiment were calculated and compared between control and knockdown groups.

Live imaging of migrating cells under laminar flow conditions was performed using a Zeiss AxioObserver Z1. The migration of ECs was measured using the Manual Tracking plugin of Fiji. A total of 240 cells (80 cells per experiment) were quantified per experimental condition. The averages of each experiment were calculated and compared between the control and knockdown groups. The position of the flow axis after 24 h corresponds to their location in reference to their starting location. To quantify the nuclear Dach1 signal, live imaging data was combined with subsequent immunostaining. This allowed us to quantify Dach1 signal and correlate it to the timepoint of direction

turning of ECs. The same datasets as mentioned above were used for these quantitations and averaged per experiment. Mean values were compared between the control and knockdown groups.

## Multiplexed scRNA-sequencing

For multiplexed scRNA-sequencing of cultured ECs, HUVECs were labeled with respective sample multiplexing antibodies using the BD Human Sample Multiplexing Kit (BD Bioscience, 633781). Briefly, cells were trypsinized, centrifuged and resuspended in 2% FBS/PBS and incubated with 10 μl sample tag antibody per $1.5 \times 10^5$ cells for 20 min on ice. After several washes with 2% FBS/PBS, equal number of cells per sample were pooled and loaded into a BD Rhapsody Cartridge (BD Bioscience, 633733) and captured on the BD Rhapsody Express Single-Cell Analysis System (BD Bioscience). Whole transcriptome mRNA and Sample Tag libraries were created using the BD Rhapsody Whole Transcriptome Analysis (WTA) Amplification Kit (BD Bioscience, 633801) and DNA sequencing was performed on a NextSeq500 (Illumina) using 2 × 75 bp paired end reads with an 8 bp single index.

## scRNA-seq preprocessing

Raw FASTQ reads were quality and adapter trimmed using TrimGalore! (version 0.6.4 length cutoff 66, quality cutoff 20). The unique molecular identifier (UMI), complex barcode, and sample tags were extracted and demultiplexed using custom scripts (rhapsody-extract-barcode and rhapsody-demultiplex).

Reads were mapped to the human reference genome version GRCh38.p13, with Gencode annotations v35, using STAR version 2.7.3a[118] (--soloType Droplet –soloCellFilter None –soloFeatures Gene –soloCBstart 1 –soloCBlen 27 –soloUMIstart 28 –soloUMIlen 8 –outFilterMultimapNmax 1 –soloCBwhitelist rhapsody_whitelist.txt).

Raw counts were imported as AnnData[119] objects. We removed low-complexity barcodes with the knee plot method, and further filtered out cells with a total contribution above 20% of reads belonging to mitochondrial mRNA. Doublets were predicted with scrublet[120] and –in the absence of any obvious doublet clusters–only cells determined as by the algorithm as doublets have been removed. Finally, each sample's gene expression matrix was normalized using scran[121] (1.22.1) with Leiden clustering[122] input at resolution 0.5.

G2/M and S phase scores were assigned to each cell using gene lists from reference [123] and the scanpy[124] (1.8.2) sc.tl.score_genes_cell_cycle function.

At this stage, samples were merged. For 2D embedding, the expression matrix was subset to the 2,000 most highly variable genes (sc.pp.highly_variable_genes, flavor "seurat"). The top 50 principal components (PCs) were calculated, and batch-corrected using Harmony[125] (0.0.5). The PCs served as basis for k-nearest neighbor calculation (sc.pp.neighbors, n_neighbors=30), which were used as input for UMAP[126] layout (sc.tl.umap, min_dist=0.3).

Leiden clustering was performed at resolution 0.2, and clusters were annotated based on known venous and arterial markers as described in the main text.

## Pseudobulk DE analysis

To compare gene expression between samples, we performed "pseudobulk" DE analysis[127]. First, cells were randomly divided into two pseudoreplicates, and unnormalized expression counts of cells belonging to the same pseudoreplicate were aggregated for each gene. Pseudobulk data was imported into R, and DE analysis was run using the Wald test for GLM coefficients implemented in the DESeq2 package[128] (1.34.7). P values were adjusted for multiple testing using independent hypothesis weighting[129] (IHW, 1.22.0).

## Statistical analysis

Statistical analysis was performed using GraphPad's Prism software. An unpaired two-tailed Student's $t$ test with Welch's correction was used for comparison between two groups. For multiple comparisons, one-way ANOVA with Dunnett's Test for multiple comparisons was used. Data is presented as scatter plots with mean ± standard error of mean (s.e.m). Differences were considered statistically significant at $p < 0.05$.

## Reporting summary

Further information on research design is available in the Nature Portfolio Reporting Summary linked to this article.

## Data availability

The single-cell RNA-seq data generated in this study have been deposited in the Gene Expression Omnibus (GEO) under accession no. GSE223738. All other data supporting the findings of this study are available from the corresponding authors on reasonable request. Source data are provided with this paper.

## Code availability

All code generated for data analysis of this study is deposited online in GitHub and can be accessed at https://github.com/Bioinformatics-Service-MPI-Munster/stewen-2024-nat-comms.

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

## Acknowledgements

The authors thank Dongying Chen and Michael Simons for kindly sharing their pERK1/2 immunostaining protocol. We also thank Stefan Volkery and Malte Stasch from the MPI BioOptic Service for microscopy imaging advice. We thank Mathias Francois for advice regarding SoxF binding studies. The study was supported by the Max Planck Society (R.H.A., M.E.P.), the DFG Collaborative Research Center 1348 Project A10 (J.S., R.H.A., M.E.P.), the Leducq Foundation (R.H.A.) and the Cells in Motion (CiM) graduate school (A.T.G.F., Z., R.H.A.). K.R.-H. is supported by National Institutes of Health (R01-HL128503) and is a Howard Hughes Medical Investigator (HHMI).

## Author contributions

J.S., R.H.A. and M.E.P. designed the study. J.S. performed the vast majority of the experiments. H.-W.J. performed the single-cell sequencing, and K.K. performed the bioinformatic transcriptomic analysis. A.T.G.F. assessed gene deletion in retinal ECs. Z. performed the luciferase assay experiments. S.A. generated the cloning construct for *R26-ephrin-B2*^iOTC^ mouse. F.B. provided general technical assistance. M.S. performed the sorting and FACS analysis of brain and retinal endothelial cells. K.R.-H. generated and provided critical reagents. J.S., R.H.A. and M.E.P. wrote the manuscript.

## Funding

## Competing interests

The authors declare no competing interests.
