## [Peer Review File · Nature Communications]

Eph-ephrin signaling couples endothelial cell sorting and arterial specificationEditorial Note: Parts of this Peer Review File have been redacted as indicated to remove third-party material where no permission to publish could be obtained. The figure on the final page of this Peer Review File has been redacted as indicated to maintain the confidentiality of unpublished data.

REVIEWER COMMENTS

Reviewer #1 (Remarks to the Author):

Stewen et al. analyze how Eph-ephrin signaling controls endothelial cell sorting and arterial specification. The authors use a variety of assays to investigate the effects of loss of EphB4 and ephrin-B2 on arterial specification. Cells lacking EphB4 are more likely to incorporate into arteries in the mouse retina. The authors then analyze downstream and upstream components mediating this cellular behavior. They show that VEGF signaling and downstream pathway activation (ERK, Akt) are affected in addition to arterial gene expression patterns. They also implicate Notch, Sox transcription factors and Dach1 in causing these phenotypes. Despite the wealth of data that are of high quality, it is not clear how all of these things depend on each other and how they regulate each other (Notch signaling for example is upstream and downstream, expression of sox17 is affected, but also affects ephrin-B2 expression, same for Dach1). The epistatic relationships of these signaling pathways and transcription factors need to be better characterized. I am somewhat lacking testable hypothesis that would guide me through the manuscript. I have some more specific comments, as detailed below.

1. I am puzzled by the way the authors cite their own literature. On page 3, the authors state "that endothelial tip cells, ..., give rise to arterial, but not venous endothelial cells" paraphrasing the title of a 2014 paper by Xu et al. "Arteries are formed by vein-derived endothelial tip cells", which the authors co-authored. However, they do not cite this paper here, but cite their follow-up publication, showing that Notch signaling is important for this tip cell behavior (Pitulescu et al., 2017). The authors state that Notch signaling functions during this process in lines 70-72 on page 3, but here they do not cite their paper (and also not a publication from their department that was published back-to-back (Hasan et al.) showing the same function of Notch signaling in zebrafish.) They instead cite two other publications (reviews). This occurs again on page 5, lines 116, 177 where the authors state that "the progeny of Esm1+ tip cells in the postnatal retina is incorporated into growing arteries", again citing their follow up publication instead of the original Xu et al. paper. On page 10, they cite the Xu et al. paper (in addition to their own publication) stating "tip cell-derived ECs,... migrate against the direction of blood flow". However, neither the Xu et al. 2014 publication nor the Pitulescu et al. 2017 publication examined cell migration in relation to blood flow. Blood flow is not mentioned in either publication.

2. In figure 1 the authors show that ephrin-B2/EPHB4 signaling is important for sorting between arterial and venous cultured ECs. This has been appreciated for some time (e.g. Lindskog, ..., Wang, Development 2014, Molecular Identification...) and the cell culture data do not add additional information. Another point is that in Supplementary Fig. 1a the authors show that knockdown of EFNB2 is efficient, even though no data on EFNB2 knockdown in HUAECs are provided (which would be the logical reciprocal experiment to the presented data on knockdown of EPHB4 in the HUVEC population). It is also not clear, how these data necessarily fit in with the rest of the manuscript.

3. Figure 1b, c: It is not clear how the authors arrive at their quantifications. The areas in the boxed regions, at least to my eye do not seem to be significantly different. How did the authors quantify this? Zooming in, sox17 (arterial marker) expression also appears to be present outside of the outlined areas that presumably delineate the arteries. This needs some clarification.

4. Figure 1d, e: The authors show that cells lacking EphB4 contribute more to arteries, while those lacking Efnb2 contribute less to arteries. One caveat here is that initiation of GFP expression and deletion of the gene of interest are two separate recombination events that do not necessarily need to go together in the same cell. Do the authors have a way to control for this? What is also not clear is where the Efnb2 deleted cells end up, or are there just fewer GFP expressing cells after Efnb2 deletion? There are fewer GFP expressing cells at the front and at the center of the retina compared to controls. If this were to be the case, providing the percentage of GFP expressing cells (e.g. those making up the artery area) would not be accurate. The authors would then need to normalize for a

reduction in the number of GFP expressing cells (similar for the EPHB4 deleted cells, where there seem to be slightly more GFP expressing cells). In Figure 4 it seems that the authors now determine total GFP areas that knockout cells occupy (no change, however this is after acute knockout). These data should be mentioned in Figure 1. However, it is also not clear in Figure 4 whether GFP+ area refers to the Front of the retina or to the Total area. The authors should clarify this throughout the manuscript. Please also see comment 8.

5. Some of the graphs contain bars that are close to zero and therefore, no bar is visible (e.g. in Figure 2b (dll4, sox17), Supp. Fig. 2a, Supp. Fig. 3g, Supp. Fig. 5d). The reader can infer which bars (experimental condition) might be zero, but it would be easier to understand if the authors were to label the bars directly instead of relying on a distinct color or pattern for each bar, which is not visible for the very short bars.

6. Figure 2: It is not clear why the authors focus on Sox17 and the influence of Sox transcription factor knockdown on ephrin-B2 and EphB4 expression. As Sox factors are important for arterial differentiation and ephrin-B2 is expressed in arteries, it is not surprising that it is downregulated in Sox knockdown cells. I do not see the hypothesis here. Are Sox factors directly regulating the expression of ephrin-B2 or EphB4, and if so, how? EphB4 goes up, while ephrin-B2 goes down in knockdown cells. Is either EphB4 or ephrin-B2 directly regulated? The authors could have investigated the effect of any other transcription factor/pathway influencing artery/vein differentiation and come to the same conclusion.

7. Figure 3: The authors examine signaling interactions between EphB4 and ephrin-B2. They show that inhibiting ERK phosphorylation after treatment of cells with EphB4/Fc limits increases in artery marker gene expression. Do the authors observe the same effect when they treat EphB4 knockdown cells (that also upregulate arterial gene expression) with an ERK inhibitor?

8. The authors examine arterial contribution of EphB4 knockout cells after short tamoxifen application (36 hrs) to examine cell migration. Here, GFP areas appear to be similar between knockout and control cell populations. Apparently, this is different for longer time-periods after knockout (Figure 1, total GFP areas are different here?). The authors do not explain this sufficiently. What are the differences between short and long periods of knockout? How does the (supposed) increase in total GFP area after long EphB4 knockout periods go together with the apparent reduction of proliferation in EphB4 knockout cells (Figure 5)? Please also refer to point 4, as the quantifications need to be adjusted depending on the total GFP areas.

9. The authors argue that Notch controls not only artery specification, but also cell migration downstream of the Eph-ephrin system (Fig. 4). Is this independent of VEGF signaling and how might Notch interact with ERK here, as active ERK signaling is known to stimulate cell migration? What happens to pERK in retinal ECs when Notch is inhibited? The authors show that EphB4 knockout increases pERK. Is this increase abrogated in dnMam11iOTC or DAPT treatment?

Reviewer #2 (Remarks to the Author):

The manuscript by Stewen et al. reports that Efnb2-Ephb4 signaling is essential in mediating VEGF-Notch signaling and arteriogenesis in the developing vasculature. This study reveals important findings in Efnb2-Ephb4 signaling, with particular mechanistic focus on Ephb4 and its balance with ligand Efnb2. Loss of one of the signaling pair Efnb2 or Ephb4 leads to an upregulation of the other. In particular, ECs without Ephb4 display increased sensitivity to VEGF, resulting in higher level of VEGF-ERK and Notch signaling. The authors link this signaling mechanism to ephb4-mediated cell cycle progression and a higher level of Dach1 expression to explain the arteriogenesis phenotype in Ephb4 iECKO. Despite many important novel findings in this study, there are some conceptual issues and

issues with data interpretation, which will require further evidence to address.

The data interpretation of Fig.1 and 2 in this paper suggests that Ephb4 and Efnb2 play reciprocal roles in regulating the extent of arteriogenesis. It is an intriguing finding that losing one of them leads to increase expression of the other. It looks like pan-endothelial deletion of either Ephb4 or Efnb2 led to obvious angiogenic phenotypes. This suggest that both Efnb2 and Ephb4 play essential roles in (potentially VEGF-mediated) sprouting angiogenesis, consistent with previous publications from the same lab (PMID: 31782728, PMID: 20445537). However, the western blot data in Fig.2a show that silencing Ephb4 or Efnb2, or a combined knockdown of both, result in elevation of NICD and Sox17 expression compared with siControl. Does this suggest that both Ephb4 and Efnb2 play the same role in repressing Sox17 and Notch pathways? If this is true, does it mean that Efnb2, a well known arterial gene, inhibits Notch signaling (which is difficult to understand)? Despite that Dll4 is increased when silencing Ephb4 but not siEfnb2, increased levels of Sox17 and NICD in both siEphb4 and siEfnb2 do explain the phenotype of reduced sprouting activity in both Ephb4iECKO and Efnb2iECKO in Fig.1, as Notch is known to negatively regulate VEGF signaling. Taken together, these findings suggest that both Efnb2 and ephb4 inhibits Notch signaling, which is difficult to understand, given that Efnb2 is an arterial gene and that the quantification data in Fig.1 showing Efnb2 is essential for arteriogenesis. New experiments are probably needed to address this issue.

In contrast to obvious angiogenesis phenotypes, arterial phenotypes actually look subtle in both Ephb4iECKO and Efnb2iECKO, though quantification of arterial area and extension in Fig.1 suggests that Efnb2 and Ephb4 play different roles in arteriogenesis. Despite such quantitative differences, formation of arteries in both models is substantial, suggesting that arterial formation and fate specification can be independent of Ephb4 and Efnb2 and that their roles in arteriogenesis are adjunctive instead of being critical. This needs to be addressed and acknowledged. A minor issues: the drawing lines mask the original signal of Sox17 in Fig1.b/c.

Data in Fig.3 indicates increased VEGF-ERK signaling in siEphb4 cells in response to VEGF stimulation due to increase VEGFR2 stability. This explains the increase of Dll4 in Fig.2a (siEphb4) and Fig.4c (Ephb4iTCKO), given that Dll4 can be induced by VEGF-ERK signaling. This may also explain the slight increase in arteriogenesis in Fig.1, if VEGF signaling is required for arteriogenesis. With a higher level of VEGF-ERK signaling in siEphb4, increased angiogenesis is expected in Ephb4iECKO, but it is not in Fig.1. Therefore, the authors need to explain how increased VEGFR2-ERK signaling in ECs depleted of Ephb4(Fig.3) can be linked to the substantially reduced angiogenesis in EphB4iECKO (Fig.1) and cell cycle arrest (Fig.5). Although it has been reported that extremely high, constitutively sustaining VEGF can lead to cell cycle arrest, the authors need to address whether this is the case in Ephb4iECKO. Also, besides Dll4 expression in the sprouting ECs, is arterial Dll4 also affected by EphB4iECKO?

Furthermore, Fig.1 d/f shows the incorporation of GFP+ tip cell-derived lineage in the arteries of Ephb4iTCKO and Efnb2iTCKO. Would the phenomenon of reduced GFP+ cells in the arteries a result of more tip cells undergoing cell death in Efnb2iTCKO? Also, there is no evidence to show that the GFP+ ECs in the arteries are truly Ephb4-null or Efnb2-null. Given that Esm1cre often results in weak deletion due to its transient expression that ceases after tip-stalk conversion, lighting up GFP is not an evidence of gene deletion.

Fig. 6 show that EphB4 is essential for flow-induced Dach1 expression and cell migration. Does siEphb4 increased flow-induced arterial genes, such as Cx37?

In a developing vasculature, arterial and venous ECs never mixed together, unless there is an A-V shunt. Does an intermingle of AECs and VECs in siEphb4 (Fig.1a) suggest any role in A-V malformation? There does not seem to be such a phenotype in Ephb4iECKO.

Reviewer #3 (Remarks to the Author):

This manuscript studies the role of EPHB4-EFNB2 in cell sorting in the retinal vasculature, using tip cell deletion, and scoring artery development. Interactions between EPH-EFN, Notch and VEGF signaling are characterized along with downstream signaling (Akt, PLCgamma, ERK). Downstream effects on fate (Sox), proliferation (cyclin) and alignment with flow (DACH10) build a model where EFNB2 signaling via NOTCH and VEGF and DACH1 orients artery cells with flow, while EPHB4 suppresses this pathway, and that EPHB4 and EFNB2 reciprocally regulate each other. The story is well executed, and complex, so the conclusions are numerous. None of the molecules and relationships among pathways are entirely novel, but the manuscript builds on many published findings, and links them together in an interesting way. For instance, EPHB4-EFNB2 signaling is long known to regulate artery-vein sorting (some missed citations here would be from the RA Wang lab). The authors claim the novelty is in demonstrating reciprocal balance of the two proteins. Additionally, a strong point of the manuscript is the integration of multiple pathways into a broader understanding of the whole process of artery vein specification. The idea of coupling sorting with specification is novel. The work is carefully done.

- An intriguing finding is that EPHB4 knockdown leads to increased arterial markers, while EFNB2 knockdown leads to decrease in Vegfr2, without changing other markers. These studies were in HUVEC. HUVEC are of large vein origin and are likely fundamentally different from small artery cells in the retina. Do microvascular cells show gene expression changes in artery markers with EFNB2 knockdown? An experiment was done in vivo, but while reciprocal regulation of EPHB4/EFNB2 was shown, the authors did not demonstrate expression of other markers (NOTCH1, DLL4, SOX17 and VEGFR2, Fig 2).

- A similar point, in Fig 2G, also done with HUVEC. There is no quantitation in this figure, but the text 'slight change' implies that EFNB2 knockdown has less of an effect. Is this also a consequence of using HUVEC as a model, or is EFNB2 not as critical for this process?

- The staining in Fig 2D of the EPHB4-AP construct is difficult to figure out. The image appears to be of lower quality.

- the argument from the cell cycle analysis of EPHB4 in vitro and in vivo work is that there is a difference between prolonged inactivation of EPHB4 vs acute inactivation (no cell cycle change in the short inactivation) despite increased arterialization. This suggests that cell cycle arrest is not necessary for arterialization, which contrasts with previous published work by others. Do the 36 hr-treated retinas show less arterialization, than the longer treated (i.e. do they arrest earlier in the artery determination pathway). Can this be demonstrated? Please also comment on the timing and fate relationships.

- Figure 3. The authors show that EPHB4 Fc and U0126 both decrease gene expression of Dll4 and Hey1. As this is pathway analysis, a double treatment (EPHB4 and U0126) should be shown to show whether there is a further decrease.

-Fig 4b. This experiment controls NICD vs NICD/EFNB2, but does not have a wildtype control on the graph for comparison. The labels on the graph are very unclear (NICD/EFNB2 double). Are the graphs from the front or centre?

-Fig S2d uses FACs sorted brain cells. Why brain cells?

-The data showing DACH1 levels changing in conjunction with flow is intriguing as it is a live imaging demonstration of the direct connection between DACH1 and cell behavior. This section is key to the paper but the logic is not well explained and the experiments are described in a condensed, rushed manner.

-The authors conclude that DACH1 promotes an early step of arterial specification as DACH1 is absent in large caliber arteries, but loss and gain of function experiments show a phenotype similar to EPHB4-EFNB2 loss of function. Similar to the comment on short and long-duration ephB4 loss, is there any evidence of an intermediate state (early 'step' vs. late 'step' in arterial specification)? There is no evidence presented here.

Minor points:

-Please add a description of what the GFP staining is in the figure legends.

-Fig 6e: The difference between the two panels is not clear. I assume they are two different examples, but it is hard to see where one panel ends and another starts.

RESPONSE TO REVIEWERS

We are very grateful to all the reviewers for their constructive comments, which helped us improve our manuscript substantially. While a detailed point-by-point response to all questions is provided below, we would like to start by listing the additional experiments and analyses in the revised manuscript:

- 1) AV gene expression changes upon knockdown of EFNB2 in HUAECs and knockdown of EPHB4 or EFNB2 in HRECs.*
- 2) EPHB4 and EFNB2 gene expression changes upon knockdown of CTNNB1 and SMAD4 in HUVECs.*
- 3) analysis of gene deletion efficiency using FACS-isolated GFP+ retinal ECs from *Efnb2^{iΔTC}*, *Ephb4^{iΔTC}* and littermate control mice.*
- 4) analysis of apoptosis and proliferation of *Efnb2^{iΔTC}* GFP+ cells.*
- 5) AV nicking phenotype in *Ephb4^{iΔEC}* retinas.*
- 6) luciferase assays to test putative Sox18 binding to EFNB2 and EPHB4 gene regions.*
- 7) arterial gene expression in EPHB4 KD HUVECs treated with ERK inhibitor U0126.*
- 8) HEY1 and DLL4 expression in HUVECs stimulated with EphB4/Fc and treated with MEK inhibitor U0126.*
- 9) EC proliferation and Sox17 expression in *Esm1-CreERT2 x R26mTmG* mice where recombination was induced by single 4-OHT administration at 24 hours (P5) or 72 hours (P3) prior mouse dissection (P6).*
- 10) migration of NICD+ tip cell-derived progeny into arteries upon anti-VEGF-A treatment.*
- 11) anti-VEGF-A treatment of *Esm1-CreERT2 x R26mTmG* mice.*
- 12) pERK immunostaining in various conditions of Notch inhibition: *dnMaml1^{iOTC}* (P1-P3 Tmx injection); DAPT treatment (1 injection at 14 hours prior analysis) of *Esm1CreERT2 x R26mTmG* mice; DAPT treatment of C57BL6 WT mice (1 injection at 14 hours prior analysis and 2 injections within 24 hours prior to analysis).*
- 13) quantitation of DLL4 expression in *Ephb4^{iΔTC}* and *Efnb2^{iΔEC}* retinal arteries.*
- 14) expression of flow responsive genes in EPHB4 KD HUVECs.*

15) comparison of cell cycle and arterial gene expression upon EPHB4 KD at 24 hours versus 48 hours.

16) DACH1 immunostaining in HUVECs at 24 hours after siEPHB4 transfection.

17) nuclear Dach1 immunostaining in Efnb2-GFP retinal arteries.

REVIEWER COMMENTS

Reviewer #1 (Remarks to the Author):

Stewen et al. analyze how Eph-ephrin signaling controls endothelial cell sorting and arterial specification. The authors use a variety of assays to investigate the effects of loss of EphB4 and ephrin-B2 on arterial specification. Cells lacking EphB4 are more likely to incorporate into arteries in the mouse retina. The authors then analyze downstream and upstream components mediating this cellular behavior. They show that VEGF signaling and downstream pathway activation (ERK, Akt) are affected in addition to arterial gene expression patterns. They also implicate Notch, Sox transcription factors and Dach1 in causing these phenotypes. Despite the wealth of data that are of high quality, it is not clear how all of these things depend on each other and how they regulate each other (Notch signaling for example is upstream and downstream, expression of sox17 is affected, but also affects ephrin-B2 expression, same for Dach1). The epistatic relationships of these signaling pathways and transcription factors need to be better characterized. I am somewhat lacking testable hypothesis that would guide me through the manuscript. I have some more specific comments, as detailed below.

We thank the reviewer for the insightful comments and feedback. Regarding the question of a testable hypothesis, we hope that this aspect is improved in the revised manuscript. In essence, we wanted to investigate how Esm1+ tip cells in the growing vasculature give rise to pre-arterial/arterial cells and how this process is controlled at the molecular level by signaling interactions. Accordingly, the main finding of our study is that ephrin-B2 controls arterial specification of tip cell progeny and guides these cells into growing arteries, whereas the receptor EphB4 opposes this process. Molecularly, ephrin-B2 enhances signaling by Notch, VEGFR2 and the MAP kinases ERK1/2 but also intersects with SoxF family transcription factors and Dach1.

We are aware that these molecular relationships are complicated and a good example is that ephrin-B2 acts both upstream and downstream of Notch and SoxF transcription factors. But it is important to appreciate that ephrin-B2 and its signaling partners generate feedback loops that enhance arterial specification once this process has been initiated in endothelial cells derived from Esm1+ tip cells. Thus, ephrin-B2 acts as a molecular amplifier that directs tip cell progeny into arteries and, at the same time, promotes the acquisition of an arterial fate.

1. I am puzzled by the way the authors cite their own literature. On page 3, the authors state “that endothelial tip cells, ..., give rise to arterial, but not venous endothelial cells” paraphrasing the title of a 2014 paper by Xu et al. “Arteries are formed by vein-derived endothelial tip cells”, which the authors co-authored. However, they do not cite this paper here, but cite their follow-up publication, showing that Notch signaling is important for this tip cell behavior (Pitulescu et al., 2017).

We apologize and want to clarify that we did not intend to leave out any relevant literature. It is correct that our original introduction was mostly focused on studies using mouse as a model system. But we agree with the reviewer and recognize that very important findings have been made in zebrafish. Accordingly, we now cite Xu et al. 2014 in our manuscript (page 3, line 75).

The authors state that Notch signaling functions during this process in lines 70-72 on page 3, but here they do not cite their paper (and also not a publication from their department that was published back-to-back (Hasan et al.) showing the same function of Notch signaling in zebrafish.) They instead cite two other publications (reviews).

Agree. We are now citing the original study in zebrafish (page 3, line 77-78).

This occurs again on page 5, lines 116, 177 where the authors state that “the progeny of Esm1+ tip cells in the postnatal retina is incorporated into growing arteries”, again citing their follow up publication instead of the original Xu et al. paper.

Same as above. We agree and now cite the original study in zebrafish (page 5, line 133).

On page 10, they cite the Xu et al. paper (in addition to their own publication) stating “tip cell-derived ECs,... migrate against the direction of blood flow”. However, neither the Xu et al. 2014 publication nor the Pitulescu et al. 2017 publication examined cell migration in relation to blood flow. Blood flow is not mentioned in either publication.

Agree. We have updated this part of the manuscript and added the relevant citations (page 14, line 423 and page 15, line 429-430).

2. In figure 1 the authors show that ephrin-B2/EPHB4 signaling is important for sorting between arterial and venous cultured ECs. This has been appreciated for some time (e.g. Lindskog,..., Wang, Development 2014, Molecular Identification...) and the cell culture data do not add additional information.

Previous studies both studies in mouse (Lindskog et al., Development, 2014, PMID: 24550118) and in zebrafish (Herbert et al., Science, 2009, PMID: 19815777) have indeed investigated the role of ephrin-B2 and EphB4 in the segregation of venous and arterial progenitors during formation of the dorsal aorta and cardinal vein. This process in the early embryo is obviously very different from our current study, which addresses postnatal angiogenesis. Another big difference is that tip cells and their progeny are not located inside the artery (with high ephrin-B2 expression) or the vein (high EphB4), but in the front region of the growing vasculature where both molecules are co-expressed at lower level.

To gain insight into the function of ephrin-B2 and EphB4, we have used genetic experiments in vivo but also a series of in vitro approaches with cultured cells. In our view, the demonstration that ephrin-B2 and EphB4 prevent the mixing of arterial and venous cells in culture (Fig. 1a) provides an excellent entry point for our study, which later goes substantially beyond mere cell sorting or segregation processes.

Another point is that in Supplementary Fig. 1a the authors show that knockdown of EFNB2 is efficient, even though no data on EFNB2 knockdown in HUAECs are provided (which would be the logical reciprocal experiment to the presented data on knockdown of EPHB4 in the

HUVEC population). It is also not clear, how these data necessarily fit in with the rest of the manuscript.

As suggested by the reviewer, we are now providing data on the knockdown of EFNB2 in HUAECs (Suppl. Fig. 3h), which leads to downregulation of multiple arterial markers but no significant upregulation of EPHB4. Reciprocal regulation of EFNB2 and EPHB4 is, however, seen in knockdown experiments using cultured human retinal ECs (HRECs), which are primary cells isolated from healthy human retina and thereby predominantly of microvascular origin (Suppl. Fig. 3d, e).

3. Figure 1b, c: It is not clear how the authors arrive at their quantifications. The areas in the boxed regions, at least to my eye do not seem to be significantly different. How did the authors quantify this? Zooming in, sox17 (arterial marker) expression also appears to be present outside of the outlined areas that presumably delineate the arteries. This needs some clarification.

*We thank the reviewer for raising this question. We revised the paragraph describing the quantitation of image data (see paragraph on **Image acquisition and analysis** in the Methods on page 39-41) for **Fig. 1c, d** (corresponding to **Fig. 1b, c** in the original submission) and we provide representative lower magnification overview images in **Suppl. Fig. 1d, e**.*

4. Figure 1d, e: The authors show that cells lacking EphB4 contribute more to arteries, while those lacking Efnb2 contribute less to arteries. One caveat here is that initiation of GFP expression and deletion of the gene of interest are two separate recombination events that do not necessarily need to go together in the same cell. Do the authors have a way to control for this?

To address this issue, we sorted GFP+ ECs from Esm1-CreERT2+ mutant and control retinas and investigated gene deletion by qRT-PCR. The resulting data (Suppl. Fig. 2a) indicates an average 50% reduction of Ephb4 transcripts. This may reflect that not all GFP+ cells have fully inactivated the Ephb4 gene but could be also caused by an inevitable lag between Cre-

mediated recombination and loss of Ephb4 transcripts. In any case, there is a statistically significant correlation between GFP expression and reduced Ephb4 expression.

For the assessment of Efnb2 gene deletion, we were only able to compare transcript levels in GFP+ cells from homozygous versus heterozygous mutants because we did not obtain Efnb2 wildtype control pups within these litters. Nevertheless, there is a 50% reduction in GFP+ ECs from Efnb2^{iΔTC} homozygous retinas relative to Efnb2^{iΔTC/+} heterozygous littermates (Suppl Fig. 2b), which is again a strong indication that we have successfully reduced ephrin-B2 expression in tip cell progeny.

In this context, we would also like to emphasize that tip cells and their progeny represent a very small fraction among retinal ECs, which means that the experiments above were very challenging. But we appreciate that it was important to confirm the validity of our genetic approach.

What is also not clear is where the Efnb2 deleted cells end up, or are there just fewer GFP expressing cells after Efnb2 deletion?

Thank you very much for this intriguing and important question. Cleaved caspase 3 staining (Suppl. Fig. 2c) provides no evidence for increased apoptosis in Efnb2^{iΔTC} retinas at the time point of analysis. In contrast, proliferation of Efnb2^{iΔTC} GFP+ cells is significantly reduced relative to control GFP+ ECs (Suppl. Fig. 2d), which is also consistent with the role of ephrin-B2 in VEGF signaling.

There are fewer GFP expressing cells at the front and at the center of the retina compared to controls. If this were to be the case, providing the percentage of GFP expressing cells (e.g. those making up the artery area) would not be accurate. The authors would then need to normalize for a reduction in the number of GFP expressing cells (similar for the EPHB4 deleted cells, where there seem to be slightly more GFP expressing cells).

It is correct that we have quantified the arterial incorporation of GFP+ cells as a readout of ephrin-B2 and EphB4 function in tip cells. It is also true that there are fewer GFP+ cells in Efnb2^{iΔTC} homozygotes, which is likely caused by the role of ephrin-B2 in VEGF signaling and downstream signal transduction. As suggested by the reviewer, we have now also quantified

the relative distribution of GFP+ cells in the arterial, plexus and front area of control and mutant retinas (Fig. 1e, f). This approach clearly confirms that less GFP+ cells are found in the Efnb2^{iΔTC} mutant artery, whereas loss of Ephb4 leads to the opposite result.

In Figure 4 it seems that the authors now determine total GFP areas that knockout cells occupy (no change, however this is after acute knockout). These data should be mentioned in Figure 1. However, it is also not clear in Figure 4 whether GFP+ area refers to the Front of the retina or to the Total area. The authors should clarify this throughout the manuscript. Please also see comment 8.

Thank you very much for this feedback. It is correct that acute inactivation of Ephb4 in Esm1+ cells leads to increased arterial incorporation of recombined cells without altering the total number of GFP+ cells relative to control littermates. After careful consideration of your suggestion, we still feel that this data is best shown later in the manuscript (Fig. 6 in the revised manuscript) when we discuss the role of migration and the interplay with other signaling pathways.

Following the reviewer's advice, we have improved the labeling of graphs and images in all figures so that it is easier to understand what is shown or has been quantified.

5. Some of the graphs contain bars that are close to zero and therefore, no bar is visible (e.g. in Figure 2b (dll4, sox17), Supp. Fig. 2a, Supp. Fig. 3g, Supp. Fig. 5d). The reader can infer which bars (experimental condition) might be zero, but it would be easier to understand if the authors were to label the bars directly instead of relying on a distinct color or pattern for each bar, which is not visible for the very short bars.

We are grateful for these comments. Direct labeling of bars might indeed be beneficial but would also substantially increase the space required for the display of these graphs. It is correct that the color/pattern of a specific bar is not visible when values are close to zero, but all these instances occur within a group of graphs where the order of bars is evident from the adjacent histograms.

6. Figure 2: It is not clear why the authors focus on Sox17 and the influence of Sox

transcription factor knockdown on ephrin-B2 and EphB4 expression. As Sox factors are important for arterial differentiation and ephrin-B2 is expressed in arteries, it is not surprising that it is downregulated in Sox knockdown cells. I do not see the hypothesis here.

Thank you for your comment. We were interested in the interdependent balance between ephrin-B2 and EphB4 expression, which is detectable at the transcript level and therefore likely to be controlled by one or several transcription factors. To identify transcriptional regulators that can control the balance between EFNB2 and EPHB4 expression and thereby promote arterial specification of tip cells, we analyzed scRNA-seq data from HUVECs in a bioinformatics approach. Selected candidate transcription factors were tested in a small-scale siRNA screen and, among all our potential candidates, only SoxF transcription factors fulfilled the criteria described. We think that including all this information would require too much space and would also detract from the main story. However, we present now two additional examples from our siRNA screen in the revised manuscript (Suppl. Fig. 4c, d).

Are Sox factors directly regulating the expression of ephrin-B2 or EphB4, and if so, how? EphB4 goes up, while ephrin-B2 goes down in knockdown cells. Is either EphB4 or ephrin-B2 directly regulated?

Similar to our finding that all three SoxF family members act redundantly in vitro, it also has been previously shown that Sox7, Sox17, and Sox18 cooperatively regulate retinal angiogenesis in mice (Zhou et al. PLoS One 2015, PMID: 26630461). For various technical and scientific reasons (e.g. strong upregulation after EPHB4 knockdown), we focused on Sox18 and selected several putative binding motifs in the EPHB4 and EFNB2 genes for further investigation (Fig. 3i). Luciferase reporter constructs containing approximately 1kb of genomic region surrounding each putative binding site were generated and tested in HUVECs (Fig. 3i, j, k). Whereas the two putative regions from EPHB4 generated no luciferase activity (Fig. 3j), two regions from EFNB2 strongly increased luciferase expression, which was further augmented by EPHB4 silencing (Fig. 3j, k). These findings argue for direct binding of Sox18 and potentially other SoxF proteins to the EFNB2 gene but not to EPHB4.

The authors could have investigated the effect of any other transcription factor/pathway influencing artery/vein differentiation and come to the same conclusion.

We appreciate this comment, but it was actually an unbiased approach based on scRNA-seq data and bioinformatics (regulon analysis) that led us to SoxF proteins (see previous question). The revised supplementary data now shows two other examples of transcription factors (β -catenin and Smad4), which lead to downregulation of either EFNB2 or EPHB4 after knockdown without simultaneous upregulation of receptor or the ligand expression, respectively (Suppl. Fig. 4c, d). We report all the relevant results on page 9 of the revised manuscript.

7. Figure 3: The authors examine signaling interactions between EphB4 and ephrin-B2. They show that inhibiting ERK phosphorylation after treatment of cells with EphB4/Fc limits increases in artery marker gene expression. Do the authors observe the same effect when they treat EphB4 knockdown cells (that also upregulate arterial gene expression) with an ERK inhibitor?

Yes, there is a reduction in arterial gene expression in EPHB4 KD cells treated with ERK inhibitor. These data are now included in Suppl. Fig. 5f.

8. The authors examine arterial contribution of EphB4 knockout cells after short tamoxifen application (36 hrs) to examine cell migration. Here, GFP areas appear to be similar between knockout and control cell populations. Apparently, this is different for longer time-periods after knockout (Figure 1, total GFP areas are different here?). The authors do not explain this sufficiently. What are the differences between short and long periods of knockout?

It is correct that prolonged inactivation of Ephb4 (tamoxifen at P1-P3 followed by analysis at P6) results in a slight but not statistically significant increase of recombined GFP+ cells, (Fig. 1c), which is not seen after short (acute) gene inactivation (Fig. 7b). The rationale for the latter series of experiments was to uncouple the possible alterations in cell cycle from roles on arterial specification and cell migration. Indeed, acute gene deletion experiments show that

GFP+ EC proliferation (EdU+/GFP+ cells) is unchanged in Ephb4^{ΔTC} retinas (Fig. 7b), whereas arterial incorporation is already increased (Fig. 6g). Together, these results indicate that the earliest steps of arterial specification and migration towards the artery precede cell cycle changes in tip cell progeny. Despite this surprising finding, we believe that full arterial differentiation requires cell cycle changes and, accordingly, fate tracking experiments show that reduced EdU incorporation in GFP+ cells after 3 days coincides with increased Sox17 expression (Fig. 7c, d). We trust that revisions have allowed us to clarify these points.

How does the (supposed) increase in total GFP area after long EphB4 knockout periods go together with the apparent reduction of proliferation in EphB4 knockout cells (Figure 5)?

We show in Fig 1c, e, that the total GFP area increases slightly but insignificantly after loss of Ephb4. As already discussed above, Ephb4-deficient tip cell progeny acquire first pre-arterial features before changes in cell cycle are detectable.

Please also refer to point 4, as the quantifications need to be adjusted depending on the total GFP areas.

As discussed under point 4, we have now also quantified the distribution of GFP+ cells in the arterial, plexus and front area of control and mutant retinas (Fig. 1e, f).

9. The authors argue that Notch controls not only artery specification, but also cell migration downstream of the Eph-ephrin system (Fig. 4). Is this independent of VEGF signaling and how might Notch interact with ERK here, as active ERK signaling is known to stimulate cell migration?

We thank the reviewer for this interesting question. To address whether migration of NICD+ cells into artery is VEGF-A dependent, we treated pups with anti-VEGF-A antibody (P5-P6) two days after induction of Notch activation in tip cells (P3-P4). Analysis of tip cell progeny at P7 shows that VEGF-A inhibition reduces the migration of NICD+/GFP+ ECs into developing arteries (Fig. 5c) relative to vehicle-treated animals.

Regarding the role of ERK1/2 activation, Luo et al. (Nature, 2021; PMC7116692) have reported that “ECs with very high VEGFR2–ERK signalling are more likely to form arterial vessels”. This is also consistent with other published research showing that augmented ERK signaling via suppression of the PI3K pathway promotes arteriogenesis (Ren et al., J Clin Invest 2010, PMID: 20237411). This is in agreement with our own findings, which show that ephrin-B2 and its elevated expression after loss EphB4, enhances VEGF-mediated activation of ERK1/2 in ECs. High pERK levels in tip cells might indeed facilitate cell migration, either outward into the peripheral retina during sprouting or during reverse migration towards arteries.

Regarding the regulation of ERK1/2 activity by Notch, we show that acute Notch inhibition (DAPT treatment for a short period) reduces pERK1/2 signal in sprouts and abrogates the increased migration of Ephb4^{iΔTC} tip cell progeny into arteries (Fig. 6f, h). Likewise, Notch inactivation using dnMaml1 expression in Esm1+ cells reduces pERK1/2 signal and impairs arterial specification and migration of these cells (Fig. 6e). However, pERK1/2 is not detected when NICD-overexpressing tip cell progeny enters the artery (see images below). This suggests that ERK1/2 activation plays a transient role early in arterial specification, which is relevant at the angiogenic front but not inside arteries.

What happens to pERK in retinal ECs when Notch is inhibited?

Here, one needs to distinguish the effect of moderate vs. robust Notch inhibition. Moderate Notch inhibition (tamoxifen at P1-P3 followed by analysis at P6) by overexpression of dominant-negative Mastermind Like Transcriptional Coactivator 1 (dnMaml1) specifically in ESM1+ tip cells leads to a considerable reduction of pERK in GFP+ (recombined) tip cells (Fig. 6e). A similar reduction in pERK is seen in retinas from mice treated with DAPT (14 hours) but not with vehicle (Fig. 6f). These experiments indicate that both VEGF and Notch activity are required for downstream ERK activation in sprouting endothelial cells.

We also tested the effect of robust Notch inhibition (2 DAPT injections within 24 hours, at 14- and 24-hours prior analysis) because Pontes-Quero et al. (Nat Commun, 2019; PMC6079472) have shown that anti-Dll4 antibody treatment for 24 hours (2 anti-Dll4 antibody injections within 24 hours, at 8- and 24-hours prior analysis) leads to upregulation of pERK throughout the angiogenic front in the postnatal retina. Indeed, we see a similar upregulation of pERK immunosignals after 24 hours of sustained Notch inhibition (Suppl. Fig. 7g). The broad upregulation of pERK after prolonged Notch inhibition is likely to reflect the strong increase in VEGF-A expression both in ECs but also in the avascular (and therefore hypoxic) peripheral retina (see Fig. 1c in Pitulescu et al. Nat Cell Biol. 2017, PMID: 28714968).

The authors show that EphB4 knockout increases pERK. Is this increase abrogated in dnMaml1^{iOTC} or DAPT treatment?

Unfortunately, it turned out to be impossible to generate a sufficient number of Ephb4^{iΔTC} dnMaml1^{iOTC} animals to address this question. One reason is that the mating strategy for the generation of these mutants is complicated and very inefficient (low number of offspring with the right genotype). In addition, we had to prioritize other genetic experiments (e.g. gene inactivation efficiency in tip cell progeny), which were also very demanding because they required a large number of animals with a certain genotype.

In our view, a direct investigation of ERK1/2 function in tip cell progeny would be required to gain conclusive insight into the role of MAPK signaling during the specification of pre-arterial cells. As other relevant downstream signaling pathways antagonize or intersect with ERK1/2

signaling, it would be relevant to look at these other pathways in the same biological context. However, these questions go beyond the scope of the current manuscript and deserve a separate study with the appropriate approaches and genetic experiments.

Reviewer #2 (Remarks to the Author):

The manuscript by Stewen et al. reports that Efnb2-Ephb4 signaling is essential in mediating VEGF-Notch signaling and arteriogenesis in the developing vasculature. This study reveals important findings in Efnb2-Ephb4 signaling, with particular mechanistic focus on Ephb4 and its balance with ligand Efnb2. Loss of one of the signaling pair Efnb2 or Ephb4 leads to an upregulation of the other. In particular, ECs without Ephb4 display increased sensitivity to VEGF, resulting in higher level of VEGF-ERK and Notch signaling. The authors link this signaling mechanism to ephb4-mediated cell cycle progression and a higher level of Dach1 expression to explain the arteriogenesis phenotype in Ephb4 iECKO. Despite many important novel findings in this study, there are some conceptual issues and issues with data interpretation, which will require further evidence to address.

We would like to thank the reviewer for summarizing our findings and for the helpful comments.

The data interpretation of Fig.1 and 2 in this paper suggests that Ephb4 and Efnb2 play reciprocal roles in regulating the extent of arteriogenesis. It is an intriguing finding that losing one of them leads to increase expression of the other. It looks like pan-endothelial deletion of either Ephb4 or Efnb2 led to obvious angiogenic phenotypes. This suggest that both Efnb2 and Ephb4 play essential roles in (potentially VEGF-mediated) sprouting angiogenesis, consistent with previous publications from the same lab (PMID: 31782728, PMID: 20445537). However, the western blot data in Fig.2a show that silencing Ephb4 or Efnb2, or a combined knockdown of both, result in elevation of NICD and Sox17 expression compared with siControl.

We are grateful for the reviewer's comments and agree that both EphB4 and ephrin-B2 control sprouting angiogenesis, which is consistent with previous work. This is also the reason for using a tip cell-specific gene inactivation strategy, which has very limited impact on general angiogenesis, for the major part of our study.

It is also correct that combined silencing of EPHB4 and EFNB2 in vitro do not completely normalize NICD and Sox17 levels (Fig. 2a, b). This might reflect that the combined elimination of ephrin-B2 and EphB4 as regulators leads to the rebalancing of certain signaling pathways to levels that are different from control cells. However, technical reasons, such as differences in the kinetics of protein loss after knockdown or the upregulation of other Eph/ephrin molecules (see Suppl. Fig. 3g), also need to be considered. It is for this reason that we do not want to overinterpret this particular finding.

Does this suggest that both Ephb4 and Efnb2 play the same role in repressing Sox17 and Notch pathways?

Our data show that EPHB4 KD HUVECs increase SOX17 and Notch pathway components (Fig. 2a, b, n, o), which indicates that EphB4 limits arterial fate induction. Conversely, EFNB2 downregulation reduces SOX7 expression without significantly changing SOX17 and SOX18 (Fig. 2a, b, o, p). Western Blot analysis of protein lysates shows that EFNB2 silencing in HUVECs does not significantly change DLL4, Notch1TM and active Notch (N1ICD) (Fig. 2a, b). However, in human retinal ECs (HRECs), which probably represent mostly microvascular cells, knockdown of EPHB4 and of EFNB2 leads to opposite effects regarding SOX17 and DLL4 expression, respectively (Suppl. Fig. 3d, e). The latter highlights the importance of balanced ephrin-B2 and EphB4 levels.

If this is true, does it mean that Efnb2, a well known arterial gene, inhibits Notch signaling (which is difficult to understand)?

Overall, our data show that ephrin-B2 promotes Notch signaling, which is consistent with the known important role of Notch in arterial specification. For example, stimulation of ephrin-B2 in cultured HUVECs by treatment with recombinant EphB4/Fc fusion protein induces the expression of DLL1, DLL4 and HEY1 (Fig. 4e). As mentioned above, knockdown of EFNB2 in

*HUAECs or HREC*s reduces *DLL4* gene expression (Suppl. Fig. 3e, h). *In vivo*, pan-endothelial deletion of *Efnb2* leads to reduced *Dll4* expression in the capillary plexus (Suppl. Fig. 7d) and in arteries (see below) without altering expression in sprouting tips (Suppl. Fig. 7d).

Despite that *Dll4* is increased when silencing *Ephb4* but not *siEfnb2*, increased levels of *Sox17* and *NICD* in both *siEphb4* and *siEfnb2* do explain the phenotype of reduced sprouting activity in both *Ephb4*^{iECKO} and *Efnb2*^{iECKO} in Fig.1, as Notch is known to negatively regulate VEGF signaling.

We agree that alterations in Notch and VEGF signaling are likely to contribute to the reduced angiogenesis in pan-endothelial Efnb2 and Ephb4 mutants. We have only included this model to show the differential effects on artery formation, namely a reduction after loss of ephrin-B2 and an increase after Ephb4 inactivation. Most of our in vivo data relies on Esm1-CreERT2 for functional experiments and general angiogenesis is not impaired by this approach, whereas we gain insight into the specification of tip cell-derived arterial ECs.

Regarding the regulation of retinal angiogenesis by Notch and VEGF, we would like to refer to the work Rui Benedito (e.g. Pontes-Quero et al., Nat Commun, 2019, PMID: 31043605) or previous work linking ephrin-B2 and VEGF signaling (Sawamiphak et al., Nature, 2010, PMID: 20445540, Wang et al., Nature, PMID: 20445537).

Taken together, these findings suggest that both *Efnb2* and *ephb4* inhibits Notch signaling, which is difficult to understand, given that *Efnb2* is an arterial gene and that the

quantification data in Fig.1 showing Efnb2 is essential for arteriogenesis. New experiments are probably needed to address this issue.

We hope that this misunderstanding is resolved by our answers provided above. Ephrin-B2 enhances Notch signaling but also VEGF signaling, whereas EphB4 suppresses Notch activity and other drivers of arterial specification.

In contrast to obvious angiogenesis phenotypes, arterial phenotypes actually look subtle in both Ephb4^{iECKO} and Efnb2^{iECKO}, though quantification of arterial area and extension in Fig.1 suggests that Efnb2 and Ephb4 play different roles in arteriogenesis.

As mentioned above, we have only included pan-endothelial Efnb2 and Ephb4 mutants to show the differential effects on artery formation, namely a reduction after loss of ephrin-B2 and an increase after Ephb4 inactivation. Most of our in vivo data relies on Esm1-CreERT2 for functional experiments and general angiogenesis is not impaired by this approach, whereas we gain insight into the specification of tip cell-derived arterial ECs.

Despite such quantitative differences, formation of arteries in both models is substantial, suggesting that arterial formation and fate specification can be independent of Ephb4 and Efnb2 and that their roles in arteriogenesis are adjunctive instead of being critical. This needs to be addressed and acknowledged.

It is correct that artery formation is not completely abolished in pan-endothelial Efnb2 mutants, which, however, is likely to reflect that inducible gene inactivation does not lead to an immediate depletion of gene transcript and protein. This is particularly relevant for arteries in the central part of the retina, which form first during postnatal development. Accordingly, arteries in the peripheral retinal vasculature, which have formed later, are strongly affected by loss of Efnb2 or Ephb4. Moreover, full knockouts for Efnb2, which do not rely on Cre/CreERT2-mediated recombination, die around midgestation and display dramatic angiogenesis defects together with an almost complete loss of arteries (Wang et al., Cell, 1999; PMID: 9630219; Adams et al., Genes Dev, 1999, PMID: 999085; Gerety and Anderson., Development, 2002; PMID: 11880349)

A minor issues: the drawing lines mask the original signal of Sox17 in Fig1.b/c.

Thank you for this comment. We agree and have modified the thickness of the line marking the artery. In addition, we display the original image without the marked artery (Suppl. Fig. 1d, e).

Data in Fig.3 indicates increased VEGF-ERK signaling in siEphb4 cells in response to VEGF stimulation due to increase VEGFR2 stability. This explains the increase of Dll4 in Fig.2a (siEphb4) and Fig.4c (Ephb4iTCKO), given that Dll4 can be induced by VEGF-ERK signaling. This may also explain the slight increase in arteriogenesis in Fig.1, if VEGF signaling is required for arteriogenesis. With a higher level of VEGF-ERK signaling in siEphb4, increased angiogenesis is expected in Ephb4iECKO, but it is not in Fig.1. Therefore, the authors need to explain how increased VEGFR2-ERK signaling in ECs depleted of Ephb4(Fig.3) can be linked to the substantially reduced angiogenesis in EphB4iECKO (Fig.1) and cell cycle arrest (Fig.5).

Thank you for this comment. The strength of the effect on retinal arteriogenesis, which has been already discussed above, strongly depends on the kinetics and the efficiency of inducible gene inactivation. Recombination of a loxP-flanked sequence will not immediately affect transcript and protein levels so that this approach has some inherent delay.

Regarding the interaction between different signaling pathways and their effect on endothelial behavior, it is clear that signaling levels and context play an important role. In the retina, VEGF is an important driver of EC proliferation in the vein and the capillary plexus near the vascular growth front, but the combination high VEGF, active ERK1/2 and high Notch signaling promotes arterial fate induction, which involves cell cycle arrest and therefore reduced proliferation. Luo and colleagues (Luo et al., Nature, 2021, PMID: 33299176) have recently concluded:" Our results also explain how VEGF, a potent inducer of ERK and mitogenic activity, can induce arterial differentiation. A relatively mild increase in ERK activity is known to induce MYC activity; however, strong VEGF-ERK stimulation induces high DLL4-Notch signalling, and this reduces MYC levels and EC proliferation, thereby promoting arterIALIZATION". Our findings regarding signaling changes after loss of EphB4 are therefore consistent with previously published work. Increased arterial specification goes along with

increased VEGFR2 and ERK1/2 activation as well as elevated Notch activation and decreased cell proliferation.

Although it has been reported that extremely high, constitutively sustaining VEGF can lead to cell cycle arrest, the authors need to address whether this is the case in Ephb4iECKO.

We think this question has been already addressed above. ECs with high VEGF and increased Notch activity arrest proliferation, as shown by the work of Rui Benedito and others (Luo et al., Nature, 2021, PMID: 33299176).

Also, besides Dll4 expression in the sprouting ECs, is arterial Dll4 also affected by EphB4iECKO?

Indeed, expression of Dll4 is increased not only in the Ephb4^{iΔTC} angiogenic front area but also in the artery and 1st order arterial branches (Suppl. Fig. 7c).

Furthermore, Fig.1 d/f shows the incorporation of GFP+ tip cell-derived lineage in the arteries of Ephb4iTCKO and Efnb2iTCKO. Would the phenomenon of reduced GFP+ cells in the arteries a result of more tip cells undergoing cell death in Efnb2iTCKO?

This is an indeed a possibility, also considering that VEGFR2 is downregulated in EFNB2 KD cells. However, evaluating apoptosis by active caspase 3 immunostaining we could not detect differences in apoptotic GFP+ cells (Suppl Fig. 2c). We know that apoptosis is a rarely observed event in the angiogenic front area and mostly observed around artery in P6 mouse retinas (Franco et al., PLOS Biology, 2015; Watson et al., Development, 2016). Yet, strong deletion of Flk1 was shown to lead to increased apoptosis in retinal vasculature (Pitulescu et al., Nature Cell Biol, Suppl. Fig. 5I, PMID: 28714968).

Regardless of no obvious increased apoptosis, we found decreased proliferation (about 30%) of GFP+ cells in the angiogenic front area in Efnb2iDTC arteries compared to control littermates, which may account for the defect in GFP+ ECs arterial incorporation (Suppl. Fig. 2d).

Also, there is no evidence to show that the GFP+ ECs in the arteries are truly Ephb4-null or Efnb2-null. Given that Esm1cre often results in weak deletion due to its transient expression that ceases after tip-stalk conversion, lighting up GFP is not an evidence of gene deletion.

To address this issue, we sorted GFP+ ECs from Esm1-CreERT2+ mutant and control retinas and investigated gene deletion by qRT-PCR. The resulting data (Suppl. Fig. 2a) indicates an average 50% reduction of Ephb4 transcripts. This may reflect that not all GFP+ cells have fully inactivated the Ephb4 gene but could be also caused by an inevitable lag between Cre-mediated recombination and loss of Ephb4 transcripts. In any case, there is a statistically significant correlation between GFP expression and reduced Ephb4 expression.

For the assessment of Efnb2 gene deletion, we were only able to compare transcript levels in GFP+ cells from homozygous versus heterozygous mutants because we did not obtain Efnb2 wildtype control pups within these litters. Nevertheless, there is a 50% reduction in GFP+ ECs from Efnb2^{iΔTC} homozygous retinas relative to Efnb2^{ΔTC/+} heterozygous littermates (Suppl Fig. 2b), which is again a strong indication that we have successfully reduced ephrin-B2 expression in tip cell progeny.

In this context, we would also like to emphasize that tip cells and their progeny represent a very small fraction among retinal ECs, which means that the experiments above were very challenging. But we appreciate that it was important to confirm the validity of our genetic approach.

Fig. 6 show that EphB4 is essential for flow-induced Dach1 expression and cell migration. Does siEphb4 increased flow-induced arterial genes, such as Cx37?

We thank the reviewer for this question. Even under static conditions, siEPHB4 knockdown cells show increased transcript levels for flow-responsive gene, including KLF2, KLF4, GJA4, GJA5, ICAM1 and ANGPT (Fig. 8b).

In a developing vasculature, arterial and venous ECs never mixed together, unless there is an A-V shunt. Does an intermingle of AECs and VECs in siEphb4 (Fig.1a) suggest any role in A-V malformation? There does not seem to be such a phenotype in Ephb4iECKO.

In Fig. 1f (Fig. 1i of the revised manuscript), we report that about 25% from Ephb4^{iΔTC/+} mutants show an AV nicking phenotype, whereas the fraction of mice with arterial overextension (i.e., longer than normal arteries) is doubled relative to control. In pan-endothelial Ephb4 and Efnb2 mutants, reduced angiogenesis is the predominant phenotype, which means that extension of the vasculature (including arteries) into the peripheral retina and thereby the chance to develop a nicking defect is profoundly reduced. In our view, these results highlight the importance of our experiments with the Esm1-CreERT2 transgenic line, which allow insights into artery formation without disruption of general angiogenesis.

Reviewer #3 (Remarks to the Author):

This manuscript studies the role of EPHB4-EFNB2 in cell sorting in the retinal vasculature, using tip cell deletion, and scoring artery development. Interactions between EPH-EFN, Notch and VEGF signaling are characterized along with downstream signaling (Akt, PLCgamma, ERK). Downstream effects on fate (Sox), proliferation (cyclin) and alignment with flow (DACH10) build a model where EFNB2 signaling via NOTCH and VEGF and DACH1 orients artery cells with flow, while EPHB4 suppresses this pathway, and that EPHB4 and EFNB2 reciprocally regulate each other. The story is well executed, and complex, so the conclusions are numerous. None of the molecules and relationships among pathways are entirely novel, but the manuscript builds on many published findings, and links them together in an interesting way.

We thank the reviewer for the kind assessment of our findings.

For instance, EPHB4-EFNB2 signaling is long known to regulate artery-vein sorting (some missed citations here would be from the RA Wang lab). The authors claim the novelty is in demonstrating reciprocal balance of the two proteins.

We have introduced several additional relevant citations, which includes Kim et al., Development, 2008; PMID: 18952909; Herbert et al., Science, 2009; PMCID: PMC2865998; Lindskog et al., Development, 2014. PMCID: PMC3929407.

Additionally, a strong point of the manuscript is the integration of multiple pathways into a broader understanding of the whole process of artery vein specification. The idea of coupling sorting with specification is novel. The work is carefully done.

Thank you very much for your kind words.

- An intriguing finding is that EPHB4 knockdown leads to increased arterial markers, while EFNB2 knockdown leads to decrease in Vegfr2, without changing other markers. These studies were in HUVEC. HUVEC are of large vein origin and are likely fundamentally different from small artery cells in the retina. Do microvascular cells show gene expression changes in artery markers with EFNB2 knockdown?

*As suggested by the reviewer, we have investigated gene expression changes upon EPHB4 knockdown and EFNB2 knockdown in human retinal endothelial cells (HRECs). Data are now presented in **Suppl. Fig. 3d, e**. The new results indicate a similar reciprocal regulation of EphB4 and ephrin-B2 at the transcriptional level.*

An experiment was done in vivo, but while reciprocal regulation of EPHB4/EFNB2 was shown, the authors did not demonstrate expression of other markers (NOTCH1, DLL4, SOX17 and VEGFR2, Fig 2).

Thank you for this comment. We show increased pERK1/2 (a readout of increased VEGF signaling) in the angiogenic front of Ephb4^{iΔEC} (Fig. 4c) and Ephb4^{iΔTC} (Suppl. Fig. 5a) retinas, whereas Efnb2^{iΔTC} mutants show reduced pERK1/2 (Suppl. Fig. 5b) compared to control. In addition, Dll4 expression is increased in Ephb4^{iΔEC} (Suppl. Fig. 7b, c) and Ephb4^{iΔTC} retinas (Fig. 6b), but decreased in Efnb2^{iΔTC} mutants (Suppl. Fig. 7d). Sox17 is upregulated in Ephb4^{iΔEC} arteries and within the sprouting area (Fig. 3a, b). While Sox17 expression is not appreciably

altered in Efnb2^{iΔEC} retinal arteries (Fig. 3c, d), the total number of Sox17^{high} ECs per total ERG+ ECs is decreased (Fig. 3d).

- A similar point, in Fig 2G, also done with HUVEC. There is no quantitation in this figure, but the text 'slight change' implies that EFNB2 knockdown has less of an effect.

It is correct that Fig. 2m (Fig. 2g in the original submission), together with many other results in the same figure, is based on scRNA-seq and therefore does not have p-values. While the changes after knockdown of EFNB2 are less profound than those seen with siEPHB4, there is a visible increase in the fraction of venous-like cells.

Is this also a consequence of using HUVEC as a model, or is EFNB2 not as critical for this process?

We thank the reviewer for this question. As shown in Fig. 2i, k, scRNA-seq data from HUVECs indicates that transcripts encoding ephrin-B2 are enriched in the tip cell-like cluster relative to the other subpopulations. It is this tip cell-like population that responds strongest to EFNB2 knockdown, leading to the downregulation of many markers associated with angiogenesis (Suppl. Fig. 3i) and cell migration (Suppl. Fig. 3j).

- The staining in Fig 2D of the EPHB4-AP construct is difficult to figure out. The image appears to be of lower quality.

We agree. This is caused by technical limitations of the AP fusion protein approach (low resolution) but also by the expression of ephrin-B2 by non-vascular cells in the retina. In fact, these cells are also abundantly visible in Efnb2-GFP knock-in animals (Fig. 9c).

- the argument from the cell cycle analysis of EPHB4 in vitro and in vivo work is that there is a difference between prolonged inactivation of EPHB4 vs acute inactivation (no cell cycle change in the short inactivation) despite increased arterialization. This suggests that cell cycle arrest is not necessary for arterialization, which contrasts with previous published work by others.

Thank you very much for this interesting question. We argue that first steps of pre-arterial cell specification occur before cell cycle changes are detectable. Nevertheless, proliferation is changed in Esm1-CreERT2-labelled tip cell progeny at 3 days after recombination, which is accompanied by increased Sox17 immunostaining, indicating progressing arterial differentiation (Fig. 7c, d).

Similarly, knockdown of EPHB4 in cultured HUVECs does not affect proliferation after 24 hours, whereas arterial markers are already upregulated. However, a much stronger increase of arterial markers and reduced EdU incorporation are seen in knockdown cells after 48 hours (Fig. 7e, g).

Taken together, these results are fully consistent with existing literature reporting that arterial specification involves cell cycle arrest. Nevertheless, our study indicates that arterial differentiation is a multi-step process where pre-arterial cells show first changes in gene expression and migratory behavior prior to arresting proliferation. Further steps of arterial differentiation and maturation are likely to follow after tip cell-derived ECs have been incorporated into arteries.

Do the 36 hr-treated retinas show less arterialization, than the longer treated (i.e. do they arrest earlier in the artery determination pathway). Can this be demonstrated? Please also comment on the timing and fate relationships.

We have compared EdU incorporation and Sox17 expression in Esm1-CreERT2 retinal endothelial cells at 1 day versus 3 days after recombination (administration of 4-OHT) (Fig. 7c and d). Our data show that acutely recombined cells (at 1 day) are more proliferative and have lower Sox17 expression compared to tip cell-derived ECs at day 3. This is consistent with the notion that first steps of pre-arterial specification and migration occur before cell cycle changes are detectable. At the same time, we do not question that cell cycle arrest is a requirement for the progression of arterial differentiation.

- Figure 3. The authors show that EPHB4 Fc and U0126 both decrease gene expression of Dll4 and Hey1. As this is pathway analysis, a double treatment (EPHB4 and U0126) should be shown to show whether there is a further decrease.

Thank you for this question. **Fig. 3g (Suppl. Fig. 5e** in the revised manuscript) shows that increased *HEY1* and *DLL4* expression in cells treated with *EphB4/Fc* is suppressed by the MEK inhibitor *U0126*. In addition, we have included new results showing that the upregulation of *HEY1* and *DLL4* after *EPHB4* knockdown is reduced by ERK pathway inhibition (*U0126* treatment for 16h) (**Suppl. Fig. 5f**).

-Fig 4b. This experiment controls *NICD* vs *NICD/EFNB2*, but does not have a wildtype control on the graph for comparison. The labels on the graph are very unclear (*NICD/EFNB2* double). Are the graphs from the front or centre?

This experiment is based on our previous finding that Esm1-CreERT2-controlled expression of NICD directs tip cell progeny into growing retinal arteries (Pitulescu et al., Nat Cell Biol. 2017, PMID: 28714968). Over time, NICD^{iOTC} disappear from the front and capillary plexus, whereas they increase inside arteries.

We now show that this effect of NICD requires ephrin-B2 (Fig. 5b). As such, this experiment does not include a wildtype control but NICD overexpression is an obvious requirement for this approach.

Fig. 8a from Pitulescu et al., Nat Cell Biol. 2017:

[redacted]

-Fig S2d uses FACs sorted brain cells. Why brain cells?

*The number of ECs per postnatal wild-type retina is very small (around 2000) and this number is even smaller in mutants with angiogenic defects. We therefore used brain-derived ECs in **Suppl. Fig. 3c (Fig. S2d in the original submission)** because it is much easier to isolate cells in sufficient number. Nevertheless, we have included data with retinal ECs in the revised manuscript (**Suppl. Fig. 2a, b**), which confirm successful gene inactivation in *Esm1-CreERT2*-derived GFP+ cells (labeled by expression of a Cre reporter).*

-The data showing DACH1 levels changing in conjunction with flow is intriguing as it is a live imaging demonstration of the direct connection between DACH1 and cell behavior. This section is key to the paper but the logic is not well explained and the experiments are described in a condensed, rushed manner.

*Thank you for this feedback. We agree that this data is very interesting and relevant. We hope that the text and data presentation are improved in the revised version (**Fig. 8, 9 and Suppl. Fig. 9**).*

-The authors conclude that DACH1 promotes an early step of arterial specification as DACH1 is absent in large caliber arteries, but loss and gain of function experiments show a phenotype similar to *EPHB4-EFNB2* loss of function. Similar to the comment on short and long-duration *ephB4* loss, is there any evidence of an intermediate state (early 'step' vs. late 'step' in arterial specification)? There is no evidence presented here.

*Our in vivo data show expression of *Dach1* in tip cells, where the transcription factor is co-expressed with *ephrin-B2* (**Fig. 9c, d**). It is also clear that *Dach1* supports pre-arterial cell specification and promotes reverse migration of tip cell-derived progeny (**Fig. 9i, j**). We have also included new data showing that nuclear *Dach1* immunostaining is absent in retinal arteries (**Suppl. Fig. 9c**), which is consistent with previous work reporting the same in maturing coronary arteries and retinal vasculature (Chang et al., *Genes Dev*, 2017; PMID: 28779009). Thus, *Dach1* and its interplay with the *Eph-ephrin* system control an early step of pre-arterial cell specification, which occurs in endothelial sprouts and precedes cell cycle changes. *Ephrin-B2*, *Notch*, *Sox17* and other regulators remain expressed in the endothelium*

of arterioles and bigger arteries, which indicates roles at later steps of arterial specification, namely proliferation arrest and upregulated expression of marker genes.

Minor points:

-Please add a description of what the GFP staining is in the figure legends.

Thank you for your feedback. We have updated the figure legends and also show additional information about genetic experiments in the figures.

-Fig 6e: The difference between the two panels is not clear. I assume they are two different examples, but it is hard to see where one panel ends and another starts.

We thank the reviewer for pointing this out. We have introduced now a heatmap representing the time course and labeled cell trajectories in the heatmap colors (Fig. 8i). The panels should be now self-explanatory.

REVIEWER COMMENTS

Reviewer #1 (Remarks to the Author):

The authors have addressed all of my concerns and greatly improved the manuscript. Congratulations on this very interesting and exciting piece of work!

Reviewer #2 (Remarks to the Author):

I appreciate new data and detailed replies to my initial review comments. However, key issues, to my mind, remain unresolved.

1. Fig.2a Data from HUVECs is a major molecular phenotype in this paper. While the role of EPHB4 is quite clearly demonstrated by siEPHB4 KD, the role of EFNB2 here remains questionable as it is in the in vivo study. The data in Fig.2a clearly show that silencing EFNB2 leads to a remarkable increase in Sox17 expression and a minor increase of NICD compared to siCtrl. The authors did not address or explain these findings. Tshi is important as these data are directly relevant to the role of EFNB2 in arterial specification. This also shows that the roles of EFNB2/EPHB4 in arterial specification is not completely reciprocal although the discovery of mutual repression of these two is interesting. Furthermore, from these data it seems that both EPHB4 and EFNB2 inhibit Notch and Sox17. This is difficult to understand as the author intended to show that losing EFNB2 leads to reduced arterialization in vivo.
2. Another issue comes with the choice of arterial markers in WB and scRNAseq data in Fig.2. Among the "arterial markers" in this figure, only Sox17 is a definitive arterial marker. Though CXCR4, Dll4 and Notch can be arterial, they are also highly expressed in the sprouting ECs and therefore not arterial-specific. The other 2 SoxF factors, Sox7 and Sox18 are also not highly expressed in arteries (see staining results in PMID: 26630461). Besides Sox17, the authors need to show data of definitive arterial markers such as Cx37, Cx40, Sema3G, Nrp1, Jag1. As it is siRNA treatments in HUVECs, these can be easily done by RT-PCR or bulk RNAseq, not necessarily scRNAseq.
3. The issue of arterial phenotypes in pan-EC or tip-EC deletion of EFNB2 or Ephb4 has not been sufficiently addressed. These phenotypes are important as they are the basis of this paper. (Line 116-117/ Suppl Fig.1d-e) The authors measured the absolute arterial length in control and EFNB2 iECKO samples and conclude that it is decreased in the KO. Given that the radial growth in the KO is also reduced, absolute arterial length is not a fair measurement. In this case, it is necessary to normalize arterial extension for the total vascular diameter. This will completely change the conclusion in line 116-117, as arterial extension in the total vascular diameter in the EFNB2 KO is actually comparable to the control, suggesting that EFNB2 is required for angiogenesis but NOT for arterial formation.
4. With tamoxifen deletion at P1-3 and dissection at P6, even though the deletion doesn't happen at P1, it does not explain the formation of arteries independent of EFNB2, as arterial formation at p1-2 is very minimal. Embryonic lethality in global EFNB2-null is completely irrelevant to the current study.
5. Arterialization in pan-EC KO of EphB4 looks increased. However, it does not look like there is any A-V "nicking" phenotype in pan-ECKO mice as it is shown in TCKO. Since this is a major phenotype of A-V patterning in addition to arterial formation, the authors need to address this question.
6. Fig.1 The definition of arterial extension in these data is inaccurate as no arterial marker (e.g. Sox17) labeling was done. Although statistical differences are shown, the overall phenotypes of arterial formation in EPHB4-iTCKO and EFNB2-iTCKO (Fig.1g/h) are actually mild. In summary, the data provided show that a loss of EFNB4 results in mild increase in arterial formation in pan EC KO, and an "A-V nick" phenotype in TC KO. This is in line with the increased contribution of tip cell lineage in the arteries as shown in the main figures. However, despite obvious angiogenesis phenotype, the arterial phenotype in EFNB2 pan-ECKO is questionable.

7. The role of EPHB4: I don't think the authors answered my questions regarding the role of EPHB4 in angiogenesis. Instead, the authors provided answers regarding its role in arterial differentiation, which is a different subject. To simplify the issue: if there is increased VEGF-ERK signaling in siEPHB4, it doesn't seem to explain the phenotype of reduced vascular growth in EPHB4 pan-ECKO. The loss of EPHB4 leads to an in vitro phenotype showing increased VEGF-ERK signaling that doesn't match the in vivo angiogenesis phenotype.

8. VEGF levels and cell cycle arrest: The authors didn't answer this question. If the authors believe there is very high VEGF-ERK signaling in the EPHB4 pan-ECKO, please provide data supporting this. Such evidence, which is not shown in the literature cited. Although the pERK staining in Fig.4c does show an increase in EPHB4 pan ECKO, it does not demonstrate a "very high" level that can arrest cell cycle, neither does this explain the reduced growth phenotype.

9. The role of EphB4 in flow-induced Dach1 expression and cell migration: This issue is only partially resolved. It would be easy to show RT-PCR data of a panel of arterial genes from siEPHB4 HUVECs under static and flow condition, including GJA4, GJA5, Sox17, Sema3G, NRP1 and Jag1.

Reviewer #3 (Remarks to the Author):

The authors have provided a detailed reply to my queries and have added new data. I have no additional concerns.

RESPONSE TO REVIEWERS

We thank Reviewer 1 and 3 for their positive response to our revised manuscript. We are also grateful to Reviewer 2 for raising useful questions, which led to inclusion of additional figure panels. Please find below a detailed point-by-point response to the questions and the additional experiments and analyses that have been included in the revised manuscript:

- 1) Quantitation and representative image of AV crossing phenotype in Ephb4^{ΔEC} retinas.*
- 2) RT-qPCR of selected arterial genes (JAG1, SEMA3G, NRP1, GJA4, GJA5) upon knockdown of EFNB2 and EPHB4 in HUVECs and HRECs.*
- 3) RT-qPCR of selected arterial genes (JAG1, SEMA3G, NRP1, GJA4, GJA5) upon knockdown of EFNB2 in HUAECs.*
- 4) Comparative expression analysis of top arterial marker genes from P6 postnatal retina (Chavkin et al. Nat Comm, 2022) and adult mouse brain (Kalucka et al., Cell, 2020) dataset, respectively, in siControl, siEPHB4 and siEFNB2 HUVEC scRNAseq own dataset.*
- 5) Confrontation assay for heterotypic co-cultures of siControl HUVECs or siEFNB2 HUVECs with siControl HUAECs.*

Reviewer #2 (Remarks to the Author):

I appreciate new data and detailed replies to my initial review comments. However, key issues, to my mind, remain unresolved.

1. **Fig.2a** Data from HUVECs is a major molecular phenotype in this paper. While the role of EPHB4 is quite clearly demonstrated by siEPHB4 KD, the role of EFNB2 here remains questionable as it is in the in vivo study. The data in Fig.2a clearly show that silencing EFNB2 leads to a remarkable increase in Sox17 expression and a minor increase of NICD compared to siCtrl. The authors did not address or explain these findings. Tshi is important as these data are directly relevant to the role of EFNB2 in arterial specification.

In Fig. 2a we show one of the three biological replicate experiments for the Western blot analyses quantified in Fig. 2b. Given the potential discrepancy between the blot and the quantitation, we provide the individual blots detecting NICD and Sox17 proteins obtained using the lysates from the three biological replicate knockdown experiments done in HUVECs for Fig. 2a. These blots were used to produce the quantitation graphs from Fig. 2b.

All three replicates clearly show the upregulation of NICD and Sox17 in the EPHB4 KD condition, whereas siEFNB2 is comparable to control.

This also shows that the roles of EFNB2/EPHB4 in arterial specification is not completely reciprocal although the discovery of mutual repression of these two is interesting.

Our data show that combined knockdown of ephrin-B2 and EphB4 does not completely restore the levels of high N1ICD or Sox17 proteins obtained upon the knockdown of EphB4. This might indicate additional changes apart from ephrin-B2-EphB4 cross-regulation but could also reflect the upregulation of additional Eph/ephrin family members in the absence of ephrin-B2 or EphB4, respectively (Supplementary Fig. 3p).

Furthermore, from these data it seems that both EPHB4 and EFNB2 inhibit Notch and Sox17. This is difficult to understand as the author intended to show that losing EFNB2 leads to reduced arterialization in vivo.

Besides the single cell transcriptomics of siControl, siEFNB2 and siEPHB4 HUVECs, we have also performed bulk RT-qPCRs to analyze SOX17, DLL4 and JAG1 gene expression in HUVECs, HRECs and HUAECs (Suppl. Fig. 3g, h, i). Whereas knockdown of EPHB4 increases SOX17 and DLL4 expression levels in HUVECs and HRECs, knockdown of EFNB2 results in a significant decrease of SOX17 and DLL4 expression without significantly altering JAG1 expression when compared to control cells. Therefore, these results along the data from Fig. 2a, b indicate that EphB4 and ephrin-B2 control Notch and Sox17 expression in opposite directions and not together.

2. Another issue comes with the choice of arterial markers in WB and scRNAseq data in Fig.2. Among the “arterial markers” in this figure, only Sox17 is a definitive arterial marker. Though CXCR4, Dll4 and Notch can be arterial, they are also highly expressed in the sprouting ECs and therefore not arterial-specific. The other 2 SoxF factors, Sox7 and Sox18 are also not highly expressed in arteries (see staining results in PMID: 26630461).

It is correct that many arterial markers, especially those expressed by more immature arteries, are also found in tip cells (DLL4, CXCR4, EFNB2, SOX17). In fact, this feature is likely to reflect the prearterial nature of tip cells. Nevertheless, these markers are used for the identification of arterial endothelium by many studies including the following recent examples: Chavkin et al., Nature Comm PMID: 36202789; Teng Ang et al., Cell, 2022, PMID: 3573828; Su et al., Nature, 2018, PMID: 29973725; Ye et al., Cells, 2022; PMID: 35406678; Genet et al., Circulation, 2023, PMID: 38126211; Grant et al., Development, 2021; PMID: 34927679.

Besides Sox17, the authors need to show data of definitive arterial markers such as Cx37, Cx40, Sema3G, Nrp1, Jag1. As it is siRNA treatments in HUVECs, these can be easily done by RT-PCR or bulk RNAseq, not necessarily scRNAseq.

In Suppl. Fig. 3 we have included RT-qPCR results for gene expression changes of additional arterial markers suggested above, from EPHB4 and EFNB2 knockdown in HUVECs (Suppl. Fig. 3g) and HRECs (Suppl. Fig. 3h), as well as from EFNB2-treated HUAECs (Suppl. Fig. 3i). The analysis shows that EPHB4 knockdown leads to the upregulation of most of these markers (excluding NRP1) in both HUVECs and HRECs. Among these markers, knockdown of EFNB2

significantly downregulates SEMA3G (50%) and upregulates GJA5 (200%) (Suppl. Fig. 3g, h). Knockdown of EFNB2 in HUAECs results in further increase in GJA4 and GJA5 connexin genes expression (Suppl. Fig. 3i).

However, functionally, EFNB2 KD HUVECs are not more arterial. In a confrontation assay, these cells segregate from siControl HUAECs as efficient as siControl HUVECs (Suppl. Fig. 3m).

Besides adding the analysis of other arterial markers suggested, we have investigated expression of arterial markers identified in P6 ECs mouse retina (Chavkin et al., PMID: 36202789) and adult mouse brain (Kalucka et al., PMID: 32059779) in the tip cell-like cluster of our scRNAseq KD samples. As shown in Suppl. Fig. 3j, k, l, transcriptional changes observed upon knockdown of EPHB4 indicate that these cells are more similar to the arterial endothelium of the angiogenic (and therefore immature) vasculature in the postnatal murine retina than to the mature and quiescent vasculature of adult mouse brain. In addition, these arterial markers show lower expression within EFNB2 KD HUVECs, which is consistent with the notion that these cells exhibit more venous-like features.

3. The issue of arterial phenotypes in pan-EC or tip-EC deletion of EFNB2 or Ephb4 has not been sufficiently addressed. These phenotypes are important as they are the basis of this paper. (Line 116-117/ Suppl Fig.1d-e) The authors measured the absolute arterial length in control and EFNB2 iECKO samples and conclude that it is decreased in the KO. Given that the radial growth in the KO is also reduced, absolute arterial length is not a fair measurement.

In this case, it is necessary to normalize arterial extension for the total vascular diameter. This will completely change the conclusion in line 116-117, as arterial extension in the total vascular diameter in the EFNB2 KO is actually comparable to the control, suggesting that EFNB2 is required for angiogenesis but NOT for arterial formation.

We have done all our quantitations as relative measurements, as indicated in the figures, legends and methods. To interpret the arterial phenotype, we have quantified several parameters: artery area (Sox17 high EC area)/ EC area; length of artery relative to the length of the vascular progression; number of arterial branches/ main artery; number of Sox17 high expression + cells/ total number of ERG+ cells.

4. With tamoxifen deletion at P1-3 and dissection at P6, even though the deletion doesn't happen at P1, it does not explain the formation of arteries independent of EFNB2, as arterial formation at p1-2 is very minimal. Embryonic lethality in global EFNB2-null is completely irrelevant to the current study.

We politely but firmly disagree. While artery formation is indeed not far advanced in the P2 retina, sprouting and therefore first steps of arterial EC specification are initiated before CreERT2-mediated gene inactivation and loss of the corresponding gene product are taking effect. The rate of phenotype development depends also on the turnover of the gene product, which varies greatly between proteins.

5. Arterialization in pan-EC KO of EphB4 looks increased. However, it does not look like there is any A-V “nicking” phenotype in pan-ECKO mice as it is shown in TCKO.

Since this is a major phenotype of A-V patterning in addition to arterial formation, the authors need to address this question.

*We now refer to this defect as AV crossover because it is open whether venous flow or diameter are altered. Representative examples of arterial alterations and AV crossovers in pan-endothelial Ephb4 mutants are shown in **Supplementary Fig. 1k**. The abundance of this phenotypic alteration is also quantified.*

We have added a paragraph (lines 532-540 on page 18) discussing this vascular abnormality and the relevance of nicking as an indicator of cardiovascular diseases such as hypertension or stroke.

6. Fig.1 The definition of arterial extension in these data is inaccurate as no arterial marker (e.g. Sox17) labeling was done.

Although statistical differences are shown, the overall phenotypes of arterial formation in EPHB4-iTCKO and EFNB2-iTCKO (Fig.1g/h) are actually mild.

In summary, the data provided show that a loss of EFNB4 results in mild increase in arterial formation in pan EC KO, and an “A-V nick” phenotype in TC KO. This is in line with the increased contribution of tip cell lineage in the arteries as shown in the main figures.

However, despite obvious angiogenesis phenotype, the arterial phenotype in EFNB2 pan-ECKO is questionable.

Regarding the first part of this question (arterial extension), please see above our answer to question 3.

With regard to the severity of the reported defects, we have already discussed the inherent delay in tamoxifen-induced gene inactivation and loss of the gene product (see our answer to question 4). These limitations are even more relevant for the targeting of tip cells because these cells are constantly formed during the angiogenic growth of the vasculature in the superficial layer of the postnatal retina. Activation of Esm1-CreERT2 may therefore not capture every single tip cell, which is particularly relevant for tip ECs emerging after P3. Despite these technical limitations and the relatively short time window of postnatal retinal angiogenesis, we report striking alterations in arterial morphology, which support opposing roles of ephrin-B2 and EphB4 in arterial development.

Regarding the pan-endothelial Efnb2 and Ephb4 mutants, these models were included to justify the use of the Esm1-CreERT2 approach, which avoids major alterations in vascular outgrowth. Moreover, it is noteworthy that artery size is differentially altered in pan-endothelial Efnb2 and Ephb4 mutants despite the fact that both models display strong reductions in vascular density and outgrowth.

7. The role of EPHB4: I don't think the authors answered my questions regarding the role of EPHB4 in angiogenesis. Instead, the authors provided answers regarding its role in arterial differentiation, which is a different subject. To simplify the issue: if there is increased VEGF-ERK signaling in siEPHB4, it doesn't seem to explain the phenotype of reduced vascular growth in EPHB4 pan-ECKO. The loss of EPHB4 leads to an in vitro phenotype showing increased VEGF-ERK signaling that doesn't match the in vivo angiogenesis phenotype.

This question is somewhat puzzling, as our manuscript primarily addresses the process of arterial specification, which also means that experiments have been designed for this specific purpose. It is appreciated that some findings, especially the effect on downstream signaling, bear some relevance for angiogenesis. With regard to the function of VEGF signaling in arterial specification, it has been convincingly shown that high levels of VEGFR2 signaling (VEGFR2^{high}) induces a cell cycle arrest and directs cells preferentially into the artery (Luo et al., Nature 2021, PMID: 33299176). In contrast, veins in this model contain a much higher fraction of VEGFR2^{low} cells despite the high proliferation in this vessel bed. These findings have established that high VEGF signaling can impair vascular growth.

8. VEGF levels and cell cycle arrest: The authors didn't answer this question. If the authors believe there is very high VEGF-ERK signaling in the EPHB4 pan-ECKO, please provide data supporting this. Such evidence, which is not shown in the literature cited.

We are referring to Chen et al. (JCI Insight, 2022; PMID: 35015735) and this study shows strong ERK activation upon EphB4 deletion in Cdh5+ endothelium (please see below Fig. 6a-c from this study with the corresponding figure legend). In our study, data showing increased phospho-ERK signal after loss of Ephb4 is shown in Fig. 4a-c.

Fig. 6. Activation of MAPK in EPHB4-deficient ECs during developmental angiogenesis.

(A) TM was administered to *Ephb4*^{fl/fl} and *Ephb4*^{fl/fl} *Cdh5*^{ert2cre} embryos at E13.5 and embryos were harvested at E18.5. (A) Skin sections were stained with anti-CD31 and anti-phospho-ERK MAPK antibodies (pERK) and Hoechst.

Note strong activation of MAPK in ECs of *Ephb4^{fl/fl} Cdh5^{ert2cre}* embryos. (B) Plot shows the percentage of pERK⁺ ECs per BV identified in randomly selected areas of skin. Bars show the mean \pm 1 SEM of percentage pERK⁺ ECs per vessel (*Ephb4^{fl/fl}*, n = 11; *Ephb4^{fl/fl} Cdh5^{ert2cre}*, n = 20). ***, P < 0.0001, Mann-Whitney test. (C) Liver tissue from individual embryos was analyzed by Western blotting using pERK antibodies. Note constitutive activation of MAPK in *Ephb4^{fl/fl} Cdh5^{ert2cre}* liver samples. See complete unedited blots in the supplemental material.

Although the pERK staining in Fig.4c does show an increase in EPHB4 pan ECKO, it does not demonstrate a “very high” level that can arrest cell cycle, neither does this explain the reduced growth phenotype.

Given that there are limits in the quantitation of immunofluorescence signals, we have rephrased the relevant sentences and no longer state that the pERK signal is “high” or “very high”.

9. The role of EphB4 in flow-induced Dach1 expression and cell migration: This issue is only partially resolved. It would be is easy to show RT-PCR data of a panel of arterial genes from siEPHB4 HUVECs under static and flow condition, including GJA4, GJA5, Sox17, Sema3G, NRP1 and Jag1.

[Redacted]